EMBO
Molecular Medicine

# APOE from astrocytes restores Alzheimer's Aβ-pathology and DAM-like responses in APOE deficient microglia

Pranav Preman [ID] [1,2], Daan Moechars [ID] [1,2], Emre Fertan [ID] [3,4], Leen Wolfs [1,2], Lutgarde Serneels [1,2], Disha Shah [1,2], Jochen Lamote [5], Suresh Poovathingal [1], An Snellinx [1,2], Renzo Mancuso [6,7], Sriram Balusu [ID] [1,2], David Klenerman [3,4], Amaia M Arranz [ID] [8,9], Mark Fiers [ID] [1,2 ✉] & Bart De Strooper [ID] [1,2,10 ✉]

## Abstract

**The major genetic risk factor for Alzheimer's disease (AD), *APOE4*, accelerates beta-amyloid (Aβ) plaque formation, but whether this is caused by APOE expressed in microglia or astrocytes is debated. We express here the human APOE isoforms in astrocytes in an *Apoe*-deficient AD mouse model. This is not only sufficient to restore the amyloid plaque pathology but also induces the characteristic transcriptional pathological responses in *Apoe*-deficient microglia surrounding the plaques. We find that both APOE4 and the protective APOE2 from astrocytes increase fibrillar plaque deposition, but differentially affect soluble Aβ aggregates. Microglia and astrocytes show specific alterations in function of *APOE* genotype expressed in astrocytes. Our experiments indicate a central role of the astrocytes in APOE mediated amyloid plaque pathology and in the induction of associated microglia responses.**

**Keywords** Alzheimer's Disease; Astrocytes; Microglia; APOE; β-amyloid Pathology
**Subject Category** Neuroscience

## Introduction

The *APOE4* variant of apolipoprotein E was identified 30 years ago as the major genetic risk factor for Alzheimer's Disease (AD) at the population level (Bennett et al, 2009; Corder et al, 1993; Strittmatter et al, 1993) while the *APOE2* variant was the first identified protective allele against AD (Corder et al, 1994; Nagy et al, 1995). Apolipoprotein E is a lipid carrier and as such involved directly or indirectly in a spectrum of pathophysiological processes. In AD, its specific role in promoting β-amyloid (Aβ) plaque pathology is, however, well established (Fleisher et al, 2013; Kok et al, 2009; Murphy et al, 2013; Schmechel et al, 1993). While effects on clearance of Aβ (Castellano et al, 2011; Deane et al, 2008; Verghese et al, 2013) have been described, the major role is direct promotion of Aβ fibrillization (Castano et al, 1995; Ma et al, 1994; Wisniewski et al, 1994) as also supported by in vivo experiments (Huynh et al, 2017; Liu et al, 2017b). This interpretation is supported by genetic studies in humans. The *APOE4* allele is associated with increased amyloid deposition (Bennett et al, 2009; Gonneaud et al, 2016; Hanson et al, 2013; Polvikoski et al, 1995), possibly explaining the early onset of disease in homozygous *APOE4* carriers (Blacker et al, 1997; Martins et al, 2005; Powell et al, 2021; Sando et al, 2008). On the other hand, AD mouse models show limited amyloid plaque pathology in the absence of *Apoe* (Bales et al, 1999; Holtzman et al, 2000b; Ulrich et al, 2018). APOE4-targeted replacement mice, expressing the human *APOE4* allele instead of the mouse *Apoe*, crossed with different AD mouse models such as PDAPP, APP/ PS1ΔE9 and 5xFAD, accelerated plaque formation compared to APOE3-targeted replacement mice (Bales et al, 2009; Castellano et al, 2011; Fagan et al, 2002; Holtzman et al, 2000a; Rodriguez et al, 2014; Youmans et al, 2012). This has also been observed with viral-vector mediated overexpression of APOE4 isoforms in AD mouse brains (Dodart et al, 2005; Hudry et al, 2013; Zhao et al, 2016).

Logically, it is assumed that the protective effect of *APOE2* is the opposite of *APOE4*. The protective effect is reflected in a low prevalence of *APOE2* alleles in AD patient cohorts (Talbot et al, 1994; West et al, 1994; Reiman et al, 2020) and reduction of Aβ plaque load in *APOE2* carriers (Chiang et al, 2010; Grothe et al, 2017; Kantarci et al, 2012; Nagy et al, 1995; Polvikoski et al, 1995; Serrano-Pozo et al, 2015). Other studies suggest that APOE2 protection may be at the level of tau pathology or even at the level of delay in cognitive deficits, without reduction in Aβ plaque formation (Berlau et al, 2007, 2009). Similarly, in mouse models,

[1]VIB Center for Brain & Disease Research, Leuven, Belgium. [2]Laboratory for the Research of Neurodegenerative Diseases, Department of Neurosciences, Leuven Brain Institute (LBI), KU Leuven (University of Leuven), Leuven, Belgium. [3]Yusuf Hamied Department of Chemistry, University of Cambridge, Cambridge, UK. [4]UK Dementia Research Institute, University of Cambridge, Cambridge, UK. [5]VIB FACS Expertise Center, Center for Cancer Biology, Leuven, Belgium. [6]Microglia and Inflammation in Neurological Disorders (MIND) Lab, VIB-UAntwerp, Centre for Molecular Neurology, Antwerp, Belgium. [7]Department of Biomedical Sciences, University of Antwerp, Antwerp, Belgium. [8]Laboratory of Humanized Models of Disease, Achucarro Basque Center for Neuroscience, Leioa, Spain. [9]Ikerbasque Basque Foundation for Science, Bilbao, Spain. [10]UK Dementia Research Institute, University College London, London, UK. ✉E-mail: mark.fiers@kuleuven.be; b.strooper@ucl.ac.uk

the effect of APOE2 on Aβ plaques remains controversial with some studies confirming decreased amyloid deposition compared to APOE3 (Castellano et al, 2011; Fagan et al, 2002; Hudry et al, 2013; Zhao et al, 2016), while others show similar or even increased deposits (Bales et al, 2009; Dodart et al, 2005; Rodriguez et al, 2014; Youmans et al, 2012). Besides, in many studies the APOE2 genotype was not included when reporting APOE4 isoform-specific effects on amyloid plaque formation (Bien-Ly et al, 2012; Fryer et al, 2005; Kim et al, 2011; Liu et al, 2017b; Mahan et al, 2022).

The cellular source that produces the pathologically relevant APOE is another point of debate. The protein is expressed mainly in the liver. However, APOE is not transferred over the blood brain barrier (Huynh et al, 2019; Lane-Donovan et al, 2016) suggesting that astrocytes are the main contributor of APOE in the brain (Boyles et al, 1985; Zhang et al, 2014). Under stress conditions APOE might be expressed by neurons (Xu et al, 2006), and under these conditions APOE4 impairs synaptic function and aggravates tau pathology (Andrews-Zwilling et al, 2010; Buttini et al, 2010; Knoferle et al, 2014; Koutsodendris et al, 2023). Neuronal APOE, however, does not seem to be associated specifically with amyloid pathology. In contrast, microglia can significantly upregulate APOE expression as part of their DAM (Disease Associated Microglia) response to the amyloid plaques (Chen et al, 2020; Keren-Shaul et al, 2017; Krasemann et al, 2017; Sala Frigerio et al, 2019). However, reconstitution of APOE expression in microglia only in an *Apoe* knockout background (Mancuso et al, 2024) or microglia-specific knockout of *APOE* had no effect on amyloid plaque formation or microglial association with plaques (Henningfield et al, 2022). This suggests that APOE upregulation in microglia, while an important characteristic of DAM cell state induction, is not directly involved in the initial plaque deposition, which as we discussed, seems to be the main AD causing pathological effect of APOE4.

Thus, it appears that in the very early phases of AD, when Aβ is believed to seed small soluble aggregates, and no DAM cell state is yet induced in the microglia, astrocytes serve as the main source of APOE in the brain. APOE expression specifically under the GFAP-promoter modified plaque deposition in an isoform-specific manner (Fagan et al, 2002; Holtzman et al, 2000a; Liu et al, 2017b), while astrocyte specific knockout of *APOE3/APOE4* led to reduction of Aβ plaque formation (Mahan et al, 2022), in line with this hypothesis. Therefore, we investigated whether it is indeed sufficient to express APOE in astrocytes to induce Aβ plaque formation, and more importantly, whether that is sufficient to induce the DAM microglia response. Interestingly, in our experimental paradigm, APOE2 expression only in astrocytes did not reduce the amyloid plaque formation compared to APOE3, even when APOE4 had significantly higher levels than both. We present transcriptomic changes in astrocytes expressing the different APOE isoforms, that suggest possible impairment of proteostasis and autophagy in APOE4 astrocytes. We provide evidence that microglia lacking endogenous APOE expression, can be stirred in their response to amyloid pathology by APOE secreted by astrocytes. Moreover, we show that the APOE effect of astrocytes on amyloid plaques is largely cell autonomous, as in the absence of microglia, amyloid plaques are still generated.

# Results

## Expression of APOE in astrocytes is sufficient to restore fibrillar amyloid plaque pathology

In order to study the role of astrocyte-derived APOE on amyloid plaque pathology, we expressed the different human APOE isoforms (further referred to as APOE2, APOE3 and APOE4), along with reporter mCherry protein, specifically in astrocytes in an *Apoe*-deficient AD mouse model ($App^{NL-G-F}$ x $Apoe^{-/-}$), using recombinant AAV2/8 particles (injected ICV at postnatal day 2/3, see Methods for details) and the astrocyte specific promoter ALDH1L1 (Fig. 1A). AAV particles expressing only mCherry protein were used as controls (further referred to as APOEKO). At 2 months of age, we observed mCherry expression in the cortical regions, along with limited spread in the subcortical area (Appendix Fig. S1A). At 6 months of age, we used immuno-fluorescence to confirm mCherry expression in ALDH1L1+ astrocytes (Appendix Fig. S1B). The ALDH1L1 promoter is specifically expressed in astrocytes in mouse brain (Cahoy et al, 2008; Zhang et al, 2014) and has been extensively used to target gene expression in an astrocyte-specific setting (Clarke et al, 2018; Endo et al, 2022; Hasel et al, 2021; Mahan et al, 2022; Michalovicz et al, 2019). Using flow cytometry, we found that around 14% of ACSA2+ astrocytes expressed mCherry (Appendix Fig. S1Cii) and around 90% of the mCherry-positive cells were positive for astrocyte marker ACSA2 (Appendix Fig. S1Ciii). It is likely that remaining mCherry-positive cells are a subpopulation of astrocytes that are ACSA2-negative. We do not exclude that a small number of non-astrocytic cells are transduced by the viral particles. However, we confirmed by immunofluorescence the lack of mCherry expression in NeuN+ neurons, Apc+ oligodendrocytes and Iba1+ microglia (Appendix Fig. S1D). We also confirmed APOE expression in mCherry+ cells (Appendix Fig. S1E), demonstrating that we reconstituted human APOE expression in astrocytes.

We characterized the amyloid pathology in the transduced mice at 6 months of age. $App^{NL-G-F}$ mice exhibited a high load of amyloid plaque deposits in cortical regions as observed by X-34 staining (Appendix Fig. S2B). This was largely absent when they were crossed with $Apoe^{-/-}$ mice (Fig. 1B; Appendix S2A,B). Expression of human APOE isoforms in astrocytes restored the fibrillar Aβ plaque depositions in the AAV-transduced cortical regions (Fig. 1B; Appendix Fig. S2A). APOE4 and, remarkably, also APOE2, showed significantly increased load of fibrillar plaques compared to APOE3 reconstituted animals (Fig. 1C). We further quantified morphological differences, namely size and compactness (Fig. EV1A), of individual Aβ plaques between the groups (see Methods for details). The APOE3 group showed a significant increase in the proportion of medium sized plaques in the range of 20–40 μm diameter, while APOE2 and APOE4 both showed increased abundance of larger plaques (above 40 μm diameter) (Fig. EV1B). Plaques in the APOE3 group showed significantly increased compactness across all size ranges (Fig. EVCi-iii). Therefore, fibrillar plaques formed in the presence of APOE3 tend to be smaller and more compact morphologically compared to plaques formed in the presence of APOE2 and APOE4.

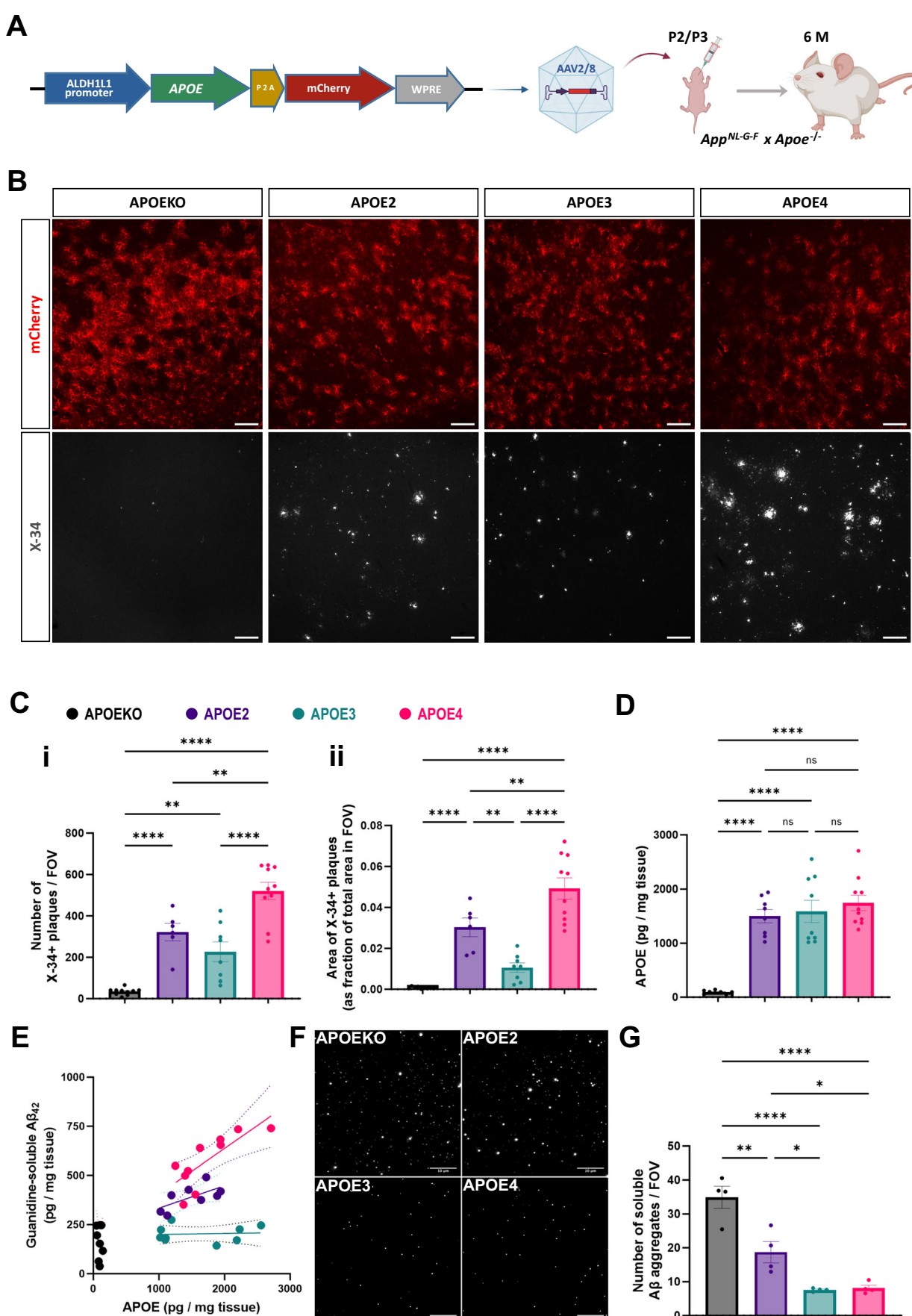

**Figure 1. Expression of APOE in astrocytes is sufficient to restore fibrillar Aβ plaque pathology.**

(A) AAV2/8 vector was used to express human APOE under the ALDH1L1 promoter. AAV particles were injected via ICV in $App^{NL-G-F}$ x $Apoe^{-/-}$ neonatal mice on postnatal days two or three (P2/P3). Samples were collected for analysis at 6 months of age. (B) IF images of AAV-transduced cortical regions (see Appendix Fig. S2A) at 6 months of age. mCherry (red) shows the transduced astrocytes. X-34 (white) shows the fibrillar plaque deposits. Scale bar: 100 μm. (C) Bar plots showing (i) number of X-34+ plaques per field of view (FOV, 20× magnification) and (ii) area of X-34+ plaques (as fraction of total area in FOV, 20× magnification), in the AAV-transduced cortical region. Data points show mean value of 3 fields of view per mouse ($n = 6–10$ mice per group). In C(i), for APOEKO vs APOE2 ****$p < 0.0001$; APOEKO vs APOE3 **$p = 0.0036$; APOEKO vs APOE4 ****$p < 0.0001$; APOE2 vs APOE3 $p = 0.375$; APOE2 vs APOE4 **$p = 0.0061$; APOE3 vs APOE4 ****$p < 0.0001$. In C(ii), for APOEKO vs APOE2 ****$p < 0.0001$; APOEKO vs APOE3 $p = 0.2237$; APOEKO vs APOE4 ****$p < 0.0001$; APOE2 vs APOE3 **$p = 0.0078$; APOE2 vs APOE4 **$p = 0.0071$; APOE3 vs APOE4 ****$p < 0.0001$. (D) Bar plot showing total APOE levels in brain homogenates measured by MSD-ELISA. Data points show mean value for 2 technical replicates per mouse ($n = 8–10$ mice per group). ns = non-significant. For APOEKO vs APOE2 ****$p < 0.0001$; APOEKO vs APOE3 ****$p < 0.0001$; APOEKO vs APOE4 ****$p < 0.0001$; APOE2 vs APOE3 $p = 0.9731$; APOE2 vs APOE4 $p = 0.6311$; APOE3 vs APOE4 $p = 0.8603$. (E) Scatter plot showing linear regression between total APOE levels (x-axis) and corresponding guanidine-soluble Aβ₄₂ levels (y-axis), measured by MSD-ELISA. Data points show mean value of 2 technical replicates per mouse ($n = 8–10$ mice per group). (F) TIRF images of mab 6E10-positive Aβ aggregates (white) captured from soaked brain fraction using SimPull technique. Scale bar: 10 μm. (G) Quantification of mAb 6E10-positive Aβ aggregates. Data points show mean value for 3 technical replicates per mouse ($n = 4$ mice per group, 9 field of views per technical replicate). For APOEKO vs APOE2 **$p = 0.0016$; APOEKO vs APOE3 ****$p < 0.0001$; APOEKO vs APOE4 ****$p < 0.0001$; APOE2 vs APOE3 *$p = 0.0223$; APOE2 vs APOE4 *$p = 0.0296$; APOE3 vs APOE4 $p = 0.9984$. Colour legend for experimental groups in (D), (E) and (G) are indicated in (C). Statistical tests: Data presented as mean ± SEM (C), (D), and (G). One-way ANOVA and Tukey's multiple comparison test (C), (D), and (G). Source data are available online for this figure.

We quantified human APOE expression in homogenized brain tissues using a human APOE specific ELISA (Fig. 1D) and qPCR (Appendix Fig. S1F) and found no significant overall differences in levels of expression between APOE2, APOE3 and APOE4 groups (Fig. 1D; Appendix Fig. S1F). Interestingly, we observed that APOE secreted from astrocytes colocalized with X-34 positive fibrillar amyloid plaques (Appendix Fig. S2C). Therefore, we wondered whether each APOE isoform could differentially affect plaque formation within each experimental group. We used ELISA to quantify the plaque associated guanidine soluble Aβ₄₂ levels in the different samples as a quantitative read out for amyloid deposition. Linear regression analysis to examine the relationship between APOE levels and guanidine soluble Aβ₄₂ levels (Fig. 1E), showed a relatively strong positive association between APOE4 and Aβ₄₂ ($R^2 = 0.61$, slope $= 0.23$, $p = 0.007$). APOE2 follows a similar trend ($R^2 = 0.43$, slope $= 0.11$, $p = 0.077$), while APOE3 levels showed no association ($R^2 = 0.006$, slope $= 0.005$, $p = 0.843$). Thus, APOE4 has a strong dose-dependent effect on amyloid fibrillization in this model of plaque deposition.

We wondered whether the absence of fibrillar amyloid plaques in the APOEKO group led to an increase in soluble Aβ species in the mouse brains. Using the single molecule pulldown (SiMPull) technique (Sideris et al, 2021), we quantified the number of soluble Aβ aggregates (Fig. 1F,G) eluted by soaking the brain tissue in artificial CSF. APOEKO mouse brains showed a significant increase in the number of soluble Aβ aggregates compared to other experimental groups (Fig. 1G), confirming a shift towards soluble species in the brain parenchyma in the absence of plaque deposition. Interestingly, the APOE4 group which showed a high level of fibrillar plaque deposition, showed low levels of soluble Aβ aggregates (Fig. 1G). APOE2 was intermediary between APOEKO and APOE4. Most remarkably, APOE3, which showed lower amounts of fibrillar plaque accumulation than APOE4 (Fig. 1C), displayed low levels of soluble aggregates similar to APOE4 in this assay (Fig. 1G).

Overall, our data confirm and extend the notion that APOE secreted from astrocytes is central in the formation of amyloid plaques in mice. While APOE4 accelerates this process, APOE2 also shows a similar trend, especially when compared to APOE3, indicating that the APOE2 protective effect is not simply the opposite of APOE4's accelerating effect on plaque formation in AD, at least in the AD mouse model used here.

## APOE expression modulates astrocyte cell-states in $App^{NL-G-F}$ mouse brain

Our data points to an important role for astrocytes in amyloid plaque formation and a distinct role of the different APOE isoforms in this process at 6 months of age. There are only limited transcriptomic studies available that investigate the effect of the human APOE variants on astrocyte phenotypes in vivo (Lee et al, 2023; Serrano-Pozo et al, 2021; Tcw et al, 2022). We therefore decided to analyse the transcriptome profiles of mCherry+/ACSA2+ astrocytes from mouse brains at 6 months of age using droplet-based scRNAseq. After filtering out non-astrocyte cells and low-quality cells, we retained 18,667 astrocytes across the 4 experimental groups. Clustering analysis revealed 12 astrocyte sub-populations (Fig. 2A) with unique gene signatures (Fig. 2B, Dataset EV1–EV2), agreeing with the high level of regional and functional heterogeneity of this cell type in the brain. The clusters divided broadly into two groups based on telencephalon and non-telencephalon identity, as previously described (Zeisel et al, 2018) (Fig. 2A,B). Clusters a1, a3 and a4 show a high expression of *Agt*, *Itih3* and *Slc6a11* which are markers for non-telencephalon astrocytes. Clusters a0, a2, a5, a6, a7, a8 and a9 show increased expression of *Mfge8*, *Lhx2* and *Ppp1r3g* which are markers for telencephalon astrocytes (Appendix Fig. S3A). Since we observed changes in plaque pathology according to APOE isoforms in the cortical areas, we focused on only the telencephalon astrocytes for further analysis. Out of these, Cluster a2 was enriched for homeostatic genes of cortical astrocytes such as *Epha5*, *Id3*, *Car2* (Batiuk et al, 2020; Murai and Pasquale, 2011; Theparambil et al, 2020). Subventricular zone astrocytes were enriched in Cluster a5 (*Slc38a1*, *Hopx*, *Zic1* and *Zic4*) (Hasel et al, 2021; Mizrak et al, 2019), hippocampal astrocytes in Cluster a7 (*Hopx*, *Hes5* and *Nr2f1*) (Li et al, 2015; Zweifel et al, 2018; Hatakeyama et al, 2004; Bertacchi et al, 2020) and Cluster a9 (*Hopx*, *Gfap* and *Igfbp5*) (Ye and D'Ercole, 1998; Bushong et al, 2002; Li et al, 2015), and striatal astrocytes in Cluster a8 (*Crym*) (Chai et al, 2017; Ollivier et al, 2024). We also identified two reactive populations of astrocytes. Cluster a10 was mainly characterized by downregulation of homeostatic astrocyte genes such as *Slc1a2*, *Glul*, *Aqp4*, and *Gja1* (Appendix Fig. S3B), and upregulated *Gfap*, *Vim*, *S100a6* and *Meg3* (Fig. 2B; Appendix Fig. S3B). Cluster a10 showed, in addition, increased expression of plaque induced genes (PIGs) such as *Cyba*, *Gpx4*, *Igfbp5*, *S100a6*, *Cd9*, and *Gfap*,

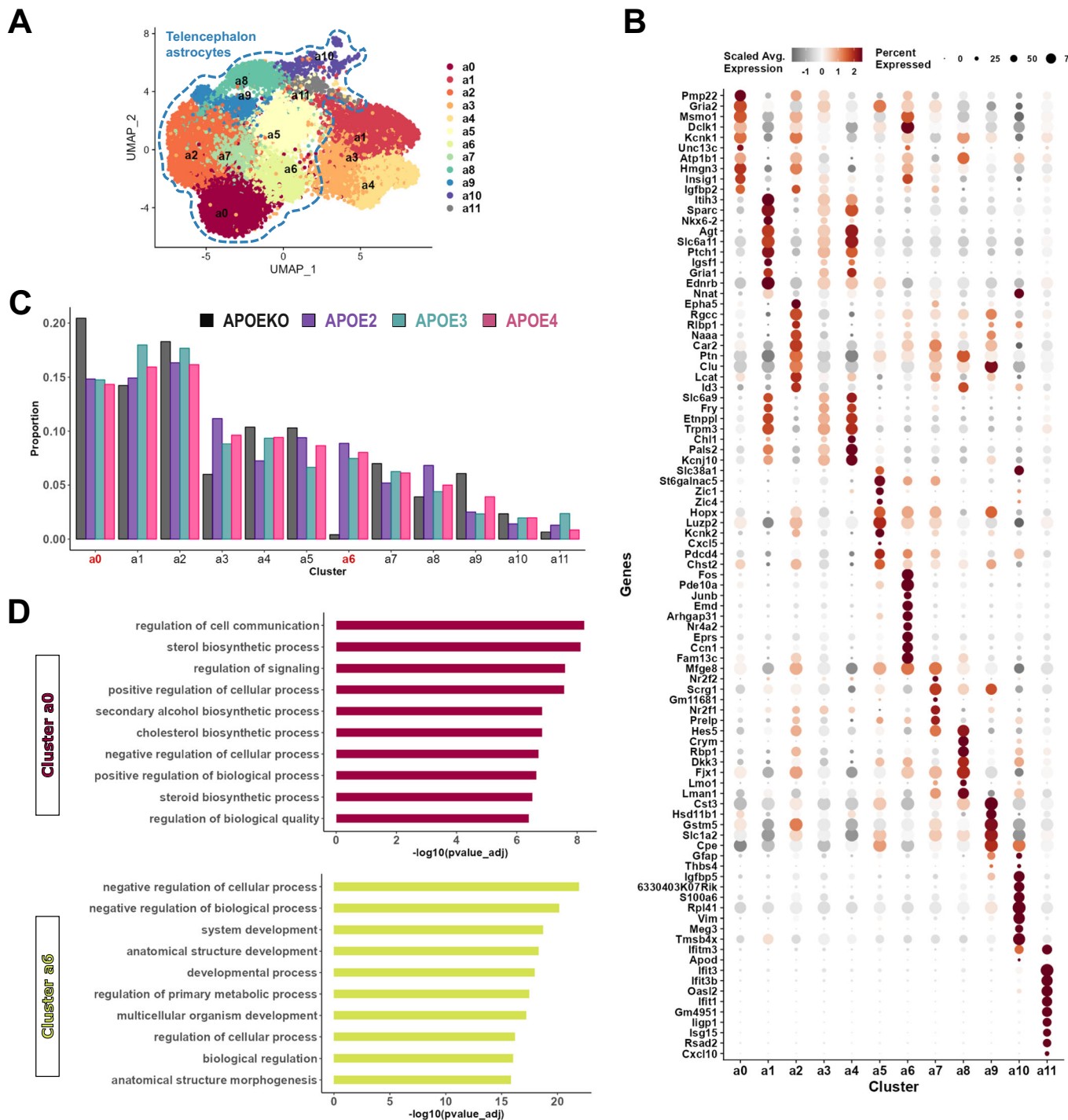

**Figure 2. APOE expression and astrocyte cell-states in *App*^NL-G-F mouse brain.**

(A) UMAP plot showing 18,667 astrocytes (mCherry+/Acsa2+) sorted from 6 months old mouse brains from the four experimental groups (*n* = 2–3 mice per group). Different sub-populations identified have been assigned Cluster numbers. Dotted lines separate the clusters with telencephalon astrocyte signatures. (B) Dot plot showing the top 10 differentially expressed genes in each cluster. Colour scale indicates normalized expression level, scaled per gene (z-score). Dot size indicates percentage of cells, in each cluster, expressing the gene. (C) Bar plot showing the proportion of astrocytes from each experimental group present in the different clusters. (D) Bar plot showing top 10 significantly enriched GO Biological process terms for Clusters a0 and a6, ordered according to −log₁₀(adjusted *p*-value). Statistical tests: g:SCS multiple testing correction method (Kolberg et al, 2023) in (D) applying significance threshold of 0.05.

previously proposed to be part of the combined microglia-astrocyte response to amyloid plaques (Chen et al, 2020) (Appendix Fig. S3C). Besides, Cluster a11 is characterized by upregulation of interferon genes such as *Cxcl10*, *Ifit3*, *Iigp1* and *Ifitm3*, indicating a cell-state of astrocyte associated to inflammation (Hasel et al, 2021) (Appendix Fig. S3D).

Thus, our APOE reconstitution experiment covered a wide range of different astrocyte subtypes and cell states. Overall, different APOE isoforms do not seem to have a major effect on the clusters apart from the APOEKO condition where Cluster a0 was strongly enriched and Cluster a6 completely absent (Fig. 2C). Functional enrichment (Gene Ontology: Biological Process database) analysis of Cluster a6 showed enrichment for terms related to negative regulation of cellular (GO:0048523) and biological (GO:0048519) processes (Fig. 2D), driven by genes such as *Pde10a*, *Emd*, *Nr4a2*, and *Eprs* (Dataset EV2). Among the top markers of Cluster a6 are genes involved in inflammatory pathways (*Fos*, *Junb*, *Nr4a2*) (Estrada et al, 2020; Zenz et al, 2008) and cell proliferation and communication (*Btg2*, *Ccn1*) (Jun and Lau, 2011; Yuniati et al, 2019; Suzuki et al, 2021) (Fig. 2B, Dataset EV1). Cluster a0 showed enrichment for terms related to sterol (GO:0016126) and cholesterol (GO:0006695) biosynthetic processes (Fig. 2D), driven by genes such as *Msmo1*, *Insig1* and *Hmgcr* (Dataset EV2). To explore the effect of APOE deficiency on astrocyte transcriptomes further, we performed differential gene expression analysis to compare each experimental group against the APOE3 group (Fig. 3A, Dataset EV3). We found that the highest number of DEGs were in the APOEKO group- 375 UP and 321 DOWN genes, compared to number of DEGs in APOE2 ($p = 4.4\mathrm{e}{-}64$) or APOE4 ($p = 4\mathrm{e}{-}85$). Top upregulated genes in the APOEKO astrocytes such as *Ubb*, *Ubc* and *Ctsl* (Fig. 3B, Dataset EV3), are involved in protein degradation, while *Ptgds* (Kanekiyo et al, 2007; Kannaian et al, 2019) and *Cst3* (Mi et al, 2007; Sastre et al, 2004), are associated with inhibition of Aβ fibrillization, in line with absence of amyloid plaques in APOEKO mice. Other upregulated genes such as *Ndrg2* (Feng et al, 2022; Zhang et al, 2023), *Gpx4* (Chen et al, 2020) and *Hmgb1* (Fan et al, 2016) are known modulators of astrocyte reactivity.

We compared next the differential effects of APOE4 vs. APOE3 and APOE2 vs. APOE3 (Fig. 3D). Compared to APOE3, both APOE4 and APOE2 significantly upregulated *Pde10a*, which was also highly upregulated in Cluster a6 (Fig. 2B) and showed minimal expression in APOEKO astrocytes (Fig. 3B,C). *Pde10a* belongs to a family of phosphodiesterases and their inhibition has been suggested to modulate AD and other neurodegenerative disorders, potentially activating CREB (cAMP response element-binding)-dependent synaptic plasticity and memory formation (García-Osta et al, 2012; O'Connor et al, 2004; Puzzo et al, 2009), but these effects were related to expression of *Pde10a* in neurons, not astrocytes. Both APOE4 and APOE2 astrocytes also upregulated reactive markers such as *Neat1* and *Cd9* (Chen et al, 2020; Habib et al, 2020; Irwin et al, 2023). APOE4 astrocytes specifically upregulated several mitochondrial genes (*mt-Nd1*, *mt-Cytb*, *mt-Rnr2*). Notably, APOE4 also downregulated several genes involved in autophagy (*Actg1*, *Tax1bp1*, *Sqstm1*, *Lamp1*, *Lamp2*) (Fig. 3D) and lipid metabolism (*Msmo1*, *Hmgcs1*). On the other hand, APOE2 astrocytes, similar to APOEKO, upregulated genes involved in protein degradation such as *Ubb* and *Uba52*, with down-regulation of lipid metabolism related genes (*Etnppl*, *Insig1*)

(Fig. 3D). Both APOE4 and APOE2 differ from APOE3 in the sense that APOE3 is characterized by upregulated interferon response genes such as *Ifi27*, *Lgals3bp*, *Ifit3* and *Ifitm3* (Fig. 3D). APOE3 astrocytes also upregulated genes involved in proteostasis (*Hspa8*, *Hsp90ab1*, *Hspa5*), *Sesn3* that reduces reactive oxygen species (ROS) in stress induced state, genes regulating phospholipase enzymes/PI signalling (*Acsl3*, *Pla2g7*, *Plcd4*, *Prex2*), and synaptic plasticity (*Nnat*). Thus, different APOE isoforms induce different transcriptomic responses in astrocytes in the *App*[NL-G-F] mouse model, with potential impairment of proteostasis and autophagy in APOE4-expressing astrocytes.

We also explored the astrocyte transcriptome for expression level and possible dysregulation of previously identified AD risk genes (Bellenguez et al, 2022) (Appendix Fig. S3E). Overall, astrocytes expressed only a small number of these risk genes. Out of these, *Clu* and *Cox7c* become highly upregulated in the absence of APOE, while *Abca1* and *Sort1* get downregulated (Fig. 3B; Appendix Fig. S3E). *Clu* has been reported to have the opposite effect of *Apoe* on amyloid pathology, with overexpression of *Clu* in AD mice showing decreased plaque deposition (Chen et al, 2021). Notably, we saw decreased expression of *Fermt2* and *Ctsb* in APOE4 astrocytes (Fig. S3E), but the difference was not significant compared to APOE3 group (Dataset EV3). A recent study has shown that downregulation of *Fermt2* in astrocytes can lead to reduction of territory size and associated cognitive deficits (Endo et al, 2022). This suggests that APOE expression in an amyloid environment is linked to the modulation of AD risk genes in astrocytes.

Overall, our analysis indicates that absence or expression of APOE has profound effects on astrocyte transcriptional features, indicating the multifunctionality of the protein. The differences between APOE2, APOE3 and APOE4 are less outspoken, but cover important functions such as interferon response genes, proteostasis genes, PI signalling, lipid metabolism, and as salient observation the upregulation of *Pde10a*.

## Fibrillar plaques induced by astrocyte-derived APOE elicit a reactive response in *Apoe*-deficient microglia

Reactive microglia states associated to amyloid plaques strongly upregulate *Apoe* expression (Keren-Shaul et al, 2017; Krasemann et al, 2017; Sala Frigerio et al, 2019). We therefore questioned whether APOE from astrocytes is necessary and sufficient to restore the response of *Apoe*-deficient microglia to amyloid plaques in our model.

We used immunofluorescence to analyse the morphological changes in microglia near the astrocyte-induced amyloid plaques. We indeed observed increased microglial clustering around the fibrillar plaque deposits (Fig. 4A). To confirm this, we segmented individual microglia (Appendix Fig. S4A,B) and classified them based on their proximity to amyloid plaques as "plaque-associated" (within X34+ thresholded plaque area and up to 5 μm outside a plaque's edge) or "non-plaque-associated" (Appendix Fig. S4C,D). Plaque-associated microglia clustered together with a significantly lower distance to neighbouring microglia, compared to the well-tiled microglia present away from plaques (Appendix Fig. S4E). Plaque-associated microglia also had significantly lower territory size (convex area) (Appendix Fig. S4Fi) and total process length (Appendix Fig. S4Fii), indicating a less

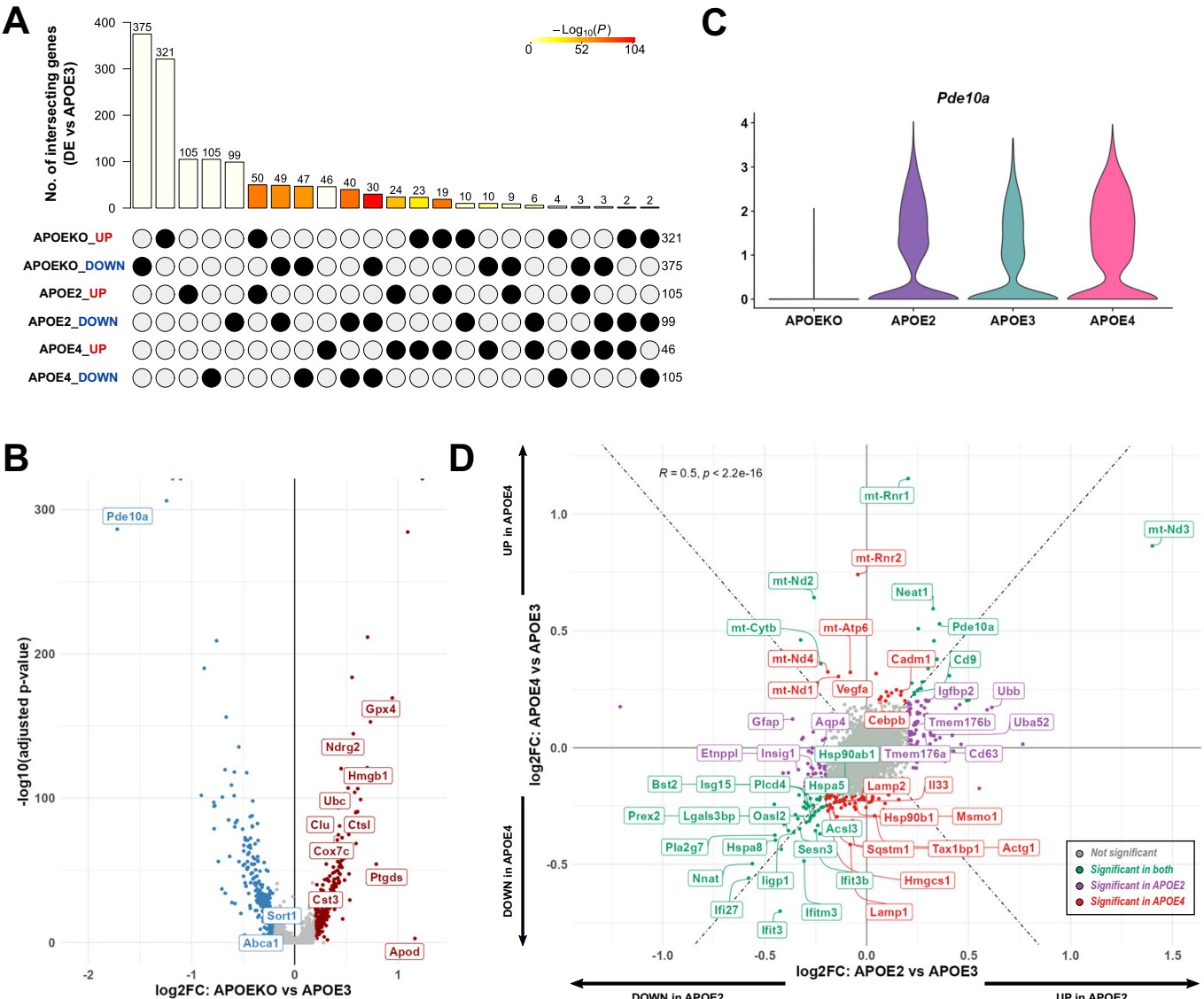

**Figure 3. APOE isoforms modulate astrocyte transcriptome in *App*^NL-G-F mouse brain.**

(A) Upset plot showing number of differentially expressed genes (UP or DOWN) in telencephalon astrocytes from each experimental group, compared to APOE3 group. Bar plot shows the number of differentially expressed genes (DEGs) common between different DE analyses (overlapping sets indicated by black dots in column below the bars). Bars are coloured according to statistical significance of intersection of DEGs. (B) Volcano plot showing differentially expressed genes between telencephalon astrocytes and APOE3 expressing telencephalon astrocytes. Data points for significant genes are coloured (Red for UP and blue for DOWN). Significance assigned based on |Log2(Fold Change)| > 0.2 and adjusted $p$-value < 0.05. (C) Violin plot showing normalized gene expression for *Pde10a* in telencephalon astrocytes from the experimental groups. ($n = 2$–3 mice per group) (D) Quadrant plot comparing differential expression of genes in telencephalon astrocytes in APOE2 vs APOE3 (along x-axis) and in APOE4 vs APOE3 (along y-axis). Colours in legend key indicate significance of genes up- or downregulated in APOE2 or APOE4 or both. Significance of differentially expressed genes based on |Log2(Fold Change)| > 0.2 and adjusted $p$-value < 0.05. Pearson's correlation, R = 0.5. Statistical tests: MAST differential expression test in (A), (B) and (D), $p$-values were adjusted with Bonferroni correction based on the total number of genes in the dataset. SuperExactTest (Wang et al, 2015) in (A). Source data are available online for this figure.

branched, amoeboid morphology, characteristic of reactive microglia. We confirmed that plaque-associated microglia stained positive for phagocytic marker Cd68 (Fig. 4B; Appendix Fig. S4G) and lysosomal protease Ctsd (Fig. 4C; Appendix Fig. S4H), characteristic of the DAM cell state (Keren-Shaul et al, 2017). Thus, *Apoe*-deficient microglia are still able to adopt a reactive state in response to fibrillar plaques induced by astrocyte-derived APOE.

Next, we wondered whether the APOE isoforms influence microglial responses, in terms of number and morphological changes. We observed a significantly higher number of microglia in the AAV-transduced region for APOE4 compared to other groups (Fig. 4Di), but regression analysis demonstrated this was due to the strong positive association ($R^2 = 0.86$, slope = 731.09, $p = 3.7e-12$) of total microglia numbers with the level of plaque load, independent of APOE isoform (Fig. 4Dii). Interestingly, this

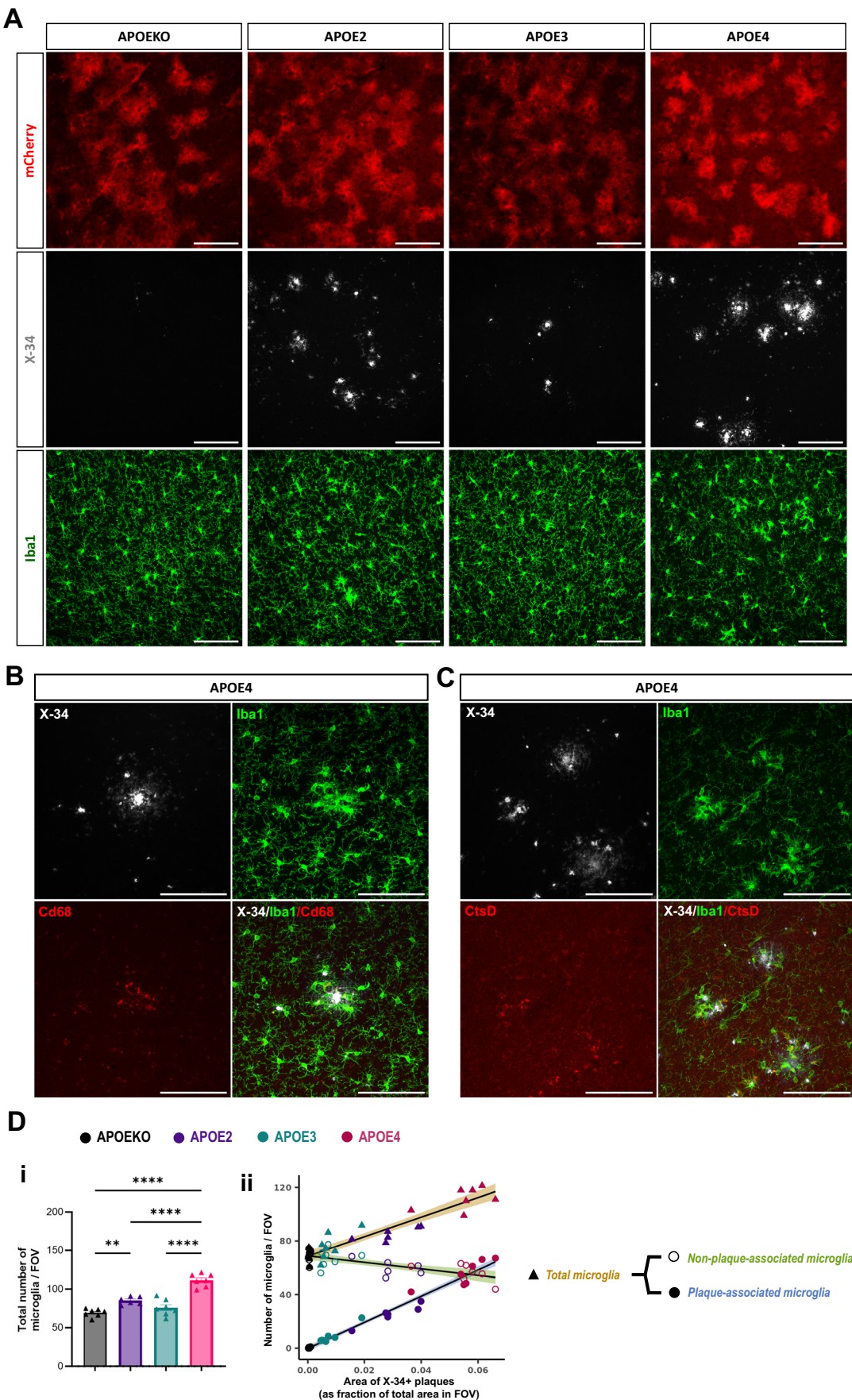

**Figure 4. *Apoe*-deficient microglia mount reactive responses to fibrillar amyloid plaques.**

(A) IF images of AAV-transduced cortical regions (see Appendix Fig. S2A), at 6 months of age. mCherry (red) shows the transduced astrocytes. X-34 staining (white) shows the fibrillar plaque deposits. Co-staining with anti-Iba1 antibody shows microglial cells (green). Scale bar: 100 μm. (B) IF images of AAV-APOE4 transduced cortical regions (see Appendix Fig. S2A) at 6 months of age. X-34 staining (white) shows the fibrillar plaque deposits. Co-staining with anti-Iba1 antibody shows microglial cells (green) and anti-Cd68 antibody shows phagocytic structures (red) inside microglia surrounding plaques. Scale bar: 100 μm. (C) IF images of AAV-APOE4 transduced cortical regions at 6 months of age. X-34 staining (white) shows the fibrillar plaque deposits. Co-staining with anti-Iba1 antibody shows microglia (green) and anti-CtsD antibody shows lysosomal structures (red) inside clustered microglia surrounding plaques. Scale bar: 100 μm. (D) (i) Bar plot showing total number of microglia per field of view (FOV, 40× magnification) in AAV-transduced regions. Data points show mean value of 3 fields of view per mouse ($n = 6–7$ mice per group). For APOEKO vs APOE2 **$p = 0.0056$; APOEKO vs APOE3 $p = 0.3957$; APOEKO vs APOE4 ****$p < 0.0001$; APOE2 vs APOE3 $p = 0.1562$; APOE2 vs APOE4 ****$p < 0.0001$; APOE3 vs APOE4 ****$p < 0.0001$. (ii) Scatter plot showing linear regression between area of X-34+ staining (as fraction of total area in FOV) and total number of microglia (solid triangle), or number of plaque-associated microglia (solid circle), or number of non-plaque-associated microglia (hollow circle), for each experimental group. Data points show mean value 3 fields of view per mouse ($n = 6–7$ mice per group). Shape legends for different categories of microglia in (i) and (ii) are indicated. Statistical tests: Data presented as mean ± SEM (Di). One-way ANOVA and Tukey's multiple comparison test in (Di). Source data are available online for this figure.

increase in microglia population with plaque load was driven by increased number of plaque-associated microglia ($R^2 = 0.97$, slope = 973.13, $p = 2e−16$) while the non-plaque-associated population remained relatively stable despite increase in plaque load ($R^2 = 0.5$, slope = −242.04, $p = 3.5e−05$) (Fig. 4Dii). Additionally, there were no significant changes in microglial morphology (territory size and total process length), for either non-plaque-associated (Appendix Fig. S4I) or plaque-associated microglia (Appendix Fig. S4J), between the different experimental groups. Therefore, at morphological level, microglial responses to fibrillar plaques do not seem to depend on the isoform of APOE produced by astrocytes but are likely mediated directly by the plaque load.

In order to explore the changes in microglial reactivity at the transcriptomic level, we performed droplet-based scRNAseq on CD11b + /CD45-low microglia isolated at 6 months of age from the mouse brains expressing different APOE isoforms in astrocytes. After filtering out non-microglial cells and low-quality cells, we retained 18,569 *Apoe*-deficient microglia over the four experimental groups (APOEKO, APOE2, APOE3 and APOE4). Unbiased clustering analysis led to the identification of 7 sub-populations of microglia (Figs. EV2A,B, Dataset EV4–EV5). We did not find major shifts in cell states as observed in other experimental paradigms ((Keren-Shaul et al, 2017; Sala Frigerio et al, 2019) probably because the AAV vector restores amyloid plaque formation only in the area of transduction (Appendix Fig. S2A). It is indeed remarkable that the effects of APOE expression in astrocytes on amyloid plaque formation is limited to those brain area where the APOE is expressed, despite APOE being a secreted protein. As we isolated microglia from the whole brain, including the non-transduced regions, only a fraction of the total pool of microglia is exposed to the amyloid plaques. However, we observed interesting shifts in gene signatures by performing differential gene expression analysis (Fig. EV3A, Dataset EV6). Among the upregulated genes in the APOEKO group were several genes involved in the cytokine response state (*Ccl3, Ccl4, Ccl12, Il1a, Gpr84, Jun*) (Fig. EV3B). Thus, APOEKO, which shows absence of fibrillar plaques and increased soluble Aβ aggregates, is associated with microglia tending to adopt a cytokine response state. Interestingly, among the different sets of DEGs, the highest intersection was observed between APOEKO and APOE4 ($p = 4.2e−78$), where 52 genes were commonly upregulated in the two groups (Fig. EV3A, EV3C, Dataset EV6) including significant overlap ($p = 2.8e−11$) of a small subset of plaque induced genes (PIGs) (*C1qb, C1qc, Cd63, Cd9, Cst3, Fcer1g, Hexb, Ly86*) (Chen

et al, 2020) (Fig. EV3C, Dataset EV6). Their upregulation in the APOEKO group which lack fibrillar plaques, suggests that induction of these plaque-associated microglial markers possibly precedes fibrillar plaque formation in amyloid models. We compared also the differentially expressed genes in APOE4 vs. APOE3 and APOE2 vs. APOE3 to understand the differential effect of the disease promoting and the disease protective allele versus APOE3 that is considered neutral (Fig. EV3D). We observed that compared to APOE2, APOE4 had a significant upregulation of several DAM marker genes such as *Cd74, Cst7, Cd63, Cd9, Ccl6*, complement cascade genes (*C1qa, C1qb, C1qc*) and the pro-inflammatory gene *Nfkbia* (Sousa et al, 2018). This was accompanied by downregulation of homeostatic genes (*Cx3cr1, P2ry13, Ccr5, Siglech*). Also, *Mertk* which is necessary for detection and engulfment of amyloid plaques by microglia (Huang et al, 2021), and *Inpp5d*, a known AD risk factor which has a role in limiting plaque formation (Castranio et al, 2023), are downregulated in APOE4. Interestingly, compared to APOE2 and APOE4, APOE3 upregulated several markers of interferon response in microglia (*Ifitm3, Ifit3, Ifi27l2a, Lgals3bp, Stat1*) (Fig. EV3D). APOE3 also showed an upregulation of antigen-presentation genes (*H2-K1, H2-D1, H2-Q7*).

Overall, our data show that *Apoe*-deficient microglia are able to react to fibrillar plaques mounting a DAM-like response when APOE is provided via astrocytes.

## Astrocyte-derived APOE is sufficient for fibrillar amyloid plaque pathology

Our data suggest two possibilities: either APOE from the astrocytes acts directly on amyloid plaque formation and then microglia respond to the plaques formed, or APOE directly modifies microglia responses and thus they sculpt the amyloid plaques with the help of the APOE delivered by the astrocytes.

To address this, we depleted microglia using PLX3397 treatment at 2 months of age before the fibrillar plaques become visible and continued treatment until the mice were 6 months of age (Fig. 5A). Despite strong depletion of microglia (percentage reductions for APOE2: 83%, APOE3: 77%, APOE4: 87%) under those conditions (Fig. 5B; Appendix Fig. S5A), fibrillar plaque deposition in APOE expressing cortical regions remained substantial, with only APOE4 group showing statistically significant decrease in plaque load upon depletion (Fig. 5C). As depletion of microglia was performed at 2 months and initial amyloid plaque seeding might have occurred

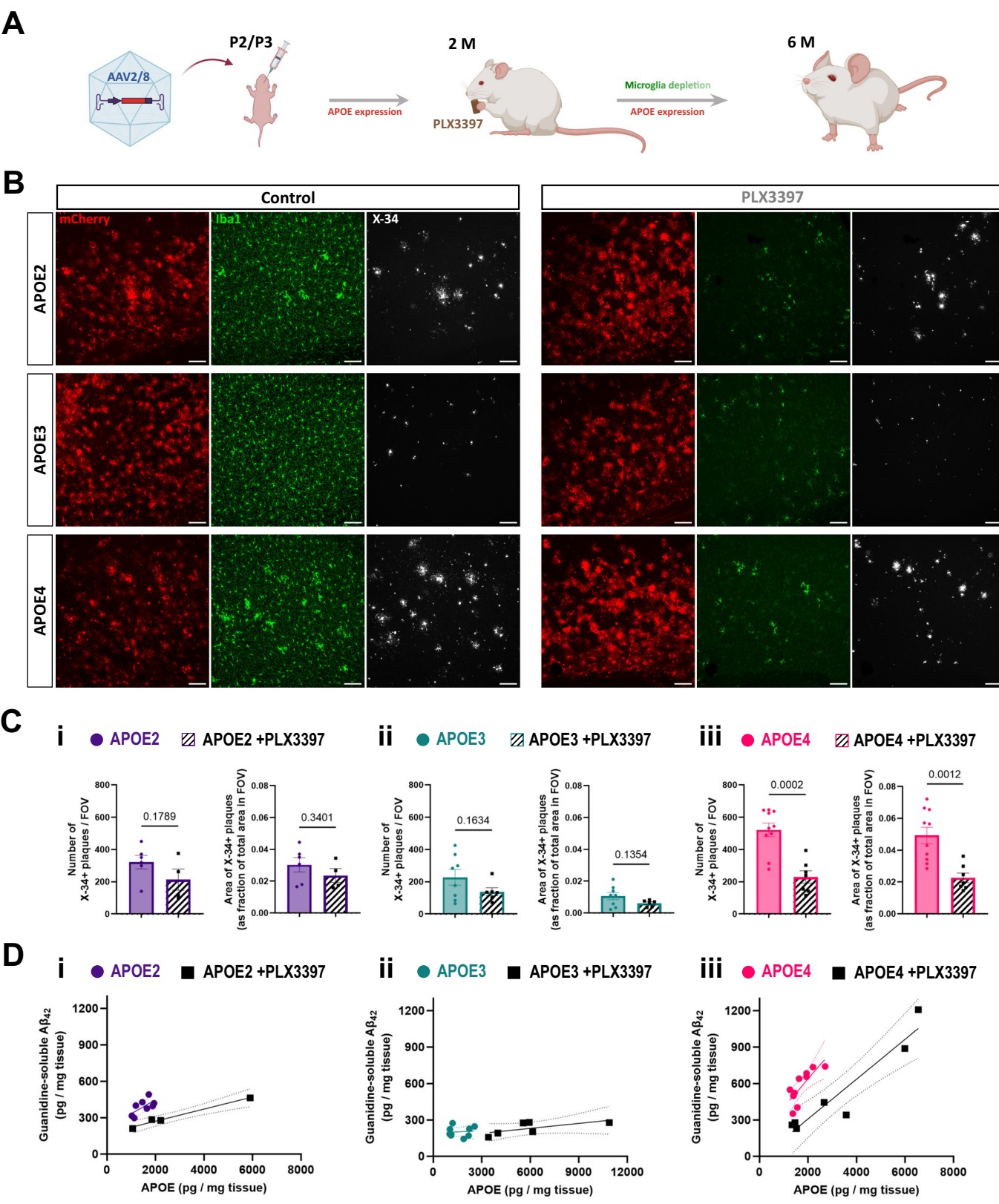

◄　**Figure 5.　Astrocyte-derived APOE initiates, and microglia modify, fibrillar Aβ plaque pathology.**

(A) Experimental design illustrating PLX3397 treatment for microglial depletion in *App*[NL-G-F] x *Apoe*[−/−] mice. Treatment started at 2 months of age by mixing PLX3397 (600 mg/kg) in chow and samples were collected for analysis at 6 months of age. (B) IF images of AAV-transduced cortical regions (see Appendix Fig. S2A) at 6 months of age from control and PLX3397-treated groups. Please note that the control group is the same cohort as used in Fig. 1. mCherry (red) shows the transduced astrocytes. X-34 staining (white) shows the fibrillar plaque deposits in mouse brain cortex. Co-staining with anti-Iba1 antibody shows microglial cells (green). Scale bar: 100 μm. The images of the control experimental group were re-captured from the same mouse cohort samples used for experimental groups in Fig. 1B which is the reason why panel 2 in column 1 partially overlaps with panel 3 in column 3 of Fig. 1. (C) Bar plots showing number of X-34+ plaques per field of view (FOV, 20× magnification) and area of X-34+ plaques (as fraction of total area in FOV, 20× magnification), in the AAV-transduced cortical region for (i) APOE2, (ii) APOE3 and (iii) APOE4 groups and their respective PLX3397-treated groups. Data points show mean value of 3 fields of view per mouse (*n* = 4–10 mice per group). Data points for control non-treated groups are the same as shown in Fig. 1C. (D) Scatter plot showing linear regression between total APOE levels (x-axis) and corresponding guanidine-soluble Aβ$_{42}$ levels (y-axis), measured by MSD-ELISA for PLX3397-treated and control mice from (i) APOE2, (ii) APOE3 and (iii) APOE4 groups. Data points show mean value for each mouse (*n* = 4–10 mice per group. 2 technical replicates per mouse). Data points for control non-treated groups are the same as shown in Fig. 1E. Statistical tests: Data represented as mean ± SEM in (C). Unpaired t-test in (C). *p*-values are indicated. Source data are available online for this figure.

already before that time point, we cannot rule out the possibility that microglia played a role in the APOE induced initial seeding of the plaques or that the remaining microglia are sufficient to induce plaque seeds.

Since we have earlier seen that APOE levels also influence Aβ plaque deposition (Fig. 1E), we used multiple regression analysis to account for the confounding effect of APOE levels while comparing guanidine-soluble Aβ$_{42}$ levels between PLX3397-treated and untreated animals in each experimental group (Fig. 5D). It is notable that especially in the APOE3 mice, the APOE levels are increased in the microglia-depleted brains (Appendix Fig. S5Bii). This increase in APOE levels was present in APOE4 mice as well (Appendix Fig. S5Biii), with APOE2 mice showing a similar non-significant trend (Appendix Fig. S5Bi). Interestingly, for the APOE3 group, like above (Fig. 1E), we found no significant association between Aβ$_{42}$ levels and APOE levels and there was no significant shift in mean Aβ$_{42}$ levels (β = −28.53, SE = 38.31, *p* = 0.4708) with PLX-treatment. On the other hand, both the APOE2 group (β = −148.2, SE = 32.22, *p* = 0.0013) and the APOE4 group (β = −357.1, SE = 60.64, *p* = 0.0002) showed a statistically significant decrease in mean Aβ$_{42}$ levels with PLX-treatment compared to the non-treated group (Fig. 5D). Therefore, this suggests that while APOE from astrocytes is sufficient for plaque deposition at 6 months of age in this model, microglia play a role in modulating Aβ pathology in APOE2 and APOE4 expressing mice.

## Discussion

We investigated the role of APOE in the interplay of astrocytes, microglia and amyloid plaques. The study confirms that APOE derived from astrocytes is sufficient to induce fibrillar Aβ plaques. It was unexpected that the protective APOE2 isoform had similar, albeit less pronounced, effects as pathogenic APOE4 on amyloid plaque formation in our experiments. Thus, its protective role in AD is likely downstream of the amyloid plaque formation, affecting the cellular phase of the disease, i.e. the response of the cells in the brain to amyloid plaques (De Strooper and Karran, 2016).

Previous studies have shown that knocking out mouse *Apoe* leads to a significant reduction of fibrillar plaque formation in different AD mouse models (Bales et al, 1999; Holtzman et al, 2000b; Ulrich et al, 2018). We confirm here in the *App*[NL-G-F] mouse that despite the presence of strong aggregation promoting mutations in this particular Aβ sequence, *Apoe* deletion almost

annihilates the generation of fibrillar amyloid plaques, while increasing the soluble Aβ aggregate species (Fig. 1). These species induce a cytokine response (Mancuso et al, 2019) in the *Apoe*-deficient microglia. Interestingly, *Apoe*-deficient astrocytes upregulate genes associated with inhibition of Aβ fibrillization (*Ptgds* and *Cst3*) and regulation of protein degradation (*Ubb, Ubc* and *Ctsl*) (Apelt et al, 2003; Davis et al, 2021; Liu et al, 2017a; Wegiel et al, 2000). Overall, loss of APOE expression has clearly multiple effects on the astrocyte-amyloid plaque-microglia interactions, in accordance with the central role of APOE in this important aspect of AD pathogenesis. It has been reported that reduction of fibrillar amyloid plaques via astrocytic-APOE knockout (Xiong et al, 2023), but not microglial-APOE knockout (Henningfield et al, 2022), results in a shift towards formation of cerebral amyloid angiopathy (CAA). However, we did not investigate the formation of CAA in our model and future studies will be needed to understand this aspect of cell-specific APOE on plaque pathology.

Restoration of APOE expression in astrocytes alone is sufficient to induce formation of fibrillar plaques and microglia reactivity, extending findings from previous studies (Holtzman et al, 2000a; Liu et al, 2017b; Mahan et al, 2022). In general, experiments suggest the order of APOE4 > APOE3 > APOE2 for effects on amyloid deposition (Fagan et al, 2002; Castellano et al, 2011; Hudry et al, 2013; Zhao et al, 2016), and therefore our observation that APOE2 is more similar to APOE4 than APOE3 in that regard in our model, is somewhat puzzling. Nevertheless other studies have also reported that APOE2 expression is associated with similar or even higher plaque load than APOE3 (Rodriguez et al, 2014; Youmans et al, 2012; Bales et al, 2009; Dodart et al, 2005). Interestingly, having an APOE2 allele (Berlau et al, 2007, 2009) or the rare isoform APOE3Ch (Christchurch variant) (Arboleda-Velasquez et al, 2019), shields patients against dementia even in the presence of high amyloid plaque load. The protective effects of APOE3Ch downstream of the amyloid plaque formation (on which a relative modest effect was observed) were also seen in an AD amyloid mouse model where overexpression had stronger effect in reducing tau seeding and propagation along with neuronal dystrophy around plaques upon injection of AD-tau brain extract (Chen et al, 2023).

No consensus exists whether and how additional pathophysiological functions of APOE2 and APOE4 contribute to the specific risk of AD. A major problem to clarify the role of APOE in Alzheimer's disease specifically and to define a therapeutic hypothesis to support a translational approach, is the pleiotropic function of APOE. Indeed, APOE4 has been implicated in the pro-

inflammatory state in microglia (Brown et al, 2002; Colton et al, 2005; Lanfranco et al, 2021; Lin et al, 2018; Serrano-Pozo et al, 2021; Vitek et al, 2009) and astrocytes (Arnaud et al, 2022); endolysosomal dysfunction (Ji et al, 2002; Nuriel et al, 2017; Pohlkamp et al, 2021; Prasad and Rao, 2018; Xian et al, 2018); impaired autophagy (Parcon et al, 2018; Simonovitch et al, 2016, 2019); inducing and promoting spreading of tau pathology, likely mediated by APOE receptors (Brecht et al, 2004; Rauch et al, 2020; Shi et al, 2017, 2021; Wang et al, 2021); neurodegeneration (Agosta et al, 2009; Lupton et al, 2016; Susanto et al, 2015); maintaining integrity of pericytes and blood brain barrier (Bell et al, 2012; Montagne et al, 2020, 2021); and impaired myelination in oligodendrocytes (Blanchard et al, 2022). We suggest that the APOE2 protective effect is largely driven by effects on the cellular reactions downstream of the amyloid plaque formation. We cannot exclude however that the APOE2 promoting effect on amyloid plaques as seen in the current work is model specific, and that expressing APOE2 also in other cell types might induce amyloid plaque reducing effects. It has also been shown that APOE2 expression leads to increased APOE levels in mouse models and human brain (Riddell et al, 2008; Bales et al, 2009; Conejero-Goldberg et al, 2014), which can influence amyloid pathology. This is not recapitulated in our model due to inherent limitation of AAV-mediated overexpression.

APOE3 expression in astrocytes in our experiments results in two phenomena: (1) the restoration of amyloid plaque formation at 6 months and (2) the decrease in soluble Aβ aggregates in these animals. At first glance, increased fibrilization might explain decreased levels of soluble Aβ aggregates, but the APOE2 allele causes more amyloid plaques than APOE3 in our model, while it also leaves more soluble Aβ aggregates in the extracellular milieu (Fig. 1C,G). In addition, APOE2, but not APOE3, shows a dose-dependent relationship between level of expression and amyloid aggregation (Fig. 1E), as APOE4 does. The results thus indicate that the relationship between APOE genotypes and Aβ plaque formation is not simple. For the interpretation of the data, it is important to take into account that APOE is secreted here only by astrocytes but is taken up via LDL receptors into the endosomal compartments of all cell types in the brain, including the microglia and astrocytes. Aβ, when endocytosed, becomes concentrated in the acidic endosomal compartments, which promotes amyloid fibril formation and then secretion (Chung et al, 1999; Hu et al, 2009; Knauer et al, 1992). Therefore, uptake of Aβ bound to APOE might cause increased concentrations of Aβ in these acidic compartments. Thus, in the absence of APOE, clearance of Aβ is lowered, resulting in increased soluble Aβ aggregates in the medium as demonstrated with the APOEKO. When APOE3 is present, Aβ peptides are taken up by astrocytes and microglia, resulting in a decrease of soluble Aβ (Fig. 1F,G). This process is receptor-mediated and therefore likely saturable, hence there was no concentration-dependent effect of APOE3 on uptake of Aβ in the overexpression condition of our experiment. When a critical concentration of Aβ is reached in the endosomes, Aβ fibrilization is induced, leading to the release of Aβ fibrils and the generation of Aβ plaques extracellularly as seen in Fig. 1B,C and measured as an increase in guanidine extractable Aβ42 in Fig. 1E. The APOE4 dose-dependent effect on the formation of amyloid plaques is not at the level of uptake, as the effect remains dose-dependent at high concentrations. APOE4 affects, however, the endosomal-lysosomal pathway (Nuriel et al,

2017) and the APOE4 induced recycling block can be rescued by decreasing pH (Xian et al, 2018). These effects on the recycling pathway are, in contrast to the saturable uptake, potentially dose-dependent and we speculate that they result in less degradation and more release of fibrillized Aβ. Less is known about the effect of APOE2 in these mechanisms. From our data it seems that APOE2 is less effective in clearing soluble Aβ from the extracellular milieu (hence increased soluble Aβ aggregates compared to APOE3) but because it also increases deposition of amyloid fibrils, APOE2 must have a similar effect as APOE4 on the functioning of endosomes. Kinetic studies should help to understand how precisely different APOE genotypes modulate these different aspects of amyloid plaque biology.

Astrocytes are a highly heterogenous group of brain cells and future studies need to explore the differential contribution of these subtypes in pathological conditions. In our model, APOE expression with the AAV was induced in different subtypes of astrocytes in the mouse brain (Fig. 2A). We identified cell states (Clusters a0 and a6) which are clearly affected by APOE expression. Enrichment of APOEKO group in Cluster a0, which is marked by processes related to lipid metabolism is expected in the absence of a lipoprotein like APOE. Cluster a6 which is exclusively present in the APOE-expressing groups, and thus also the plaque-bearing brains, is enriched for genes modulating inflammatory pathways which might play a role in astrocyte reactivity towards amyloid pathology. Differential gene expression analysis of APOE2 and APOE4 versus APOE3 showed downregulation of several autophagy-related genes specifically in APOE4 expressing astrocytes, and not in APOE2 astrocytes. This is in line with evidence from brain samples of APOE4 patients showing downregulation of autophagy components (Parcon et al, 2018) and supports the role of astrocyte-mediated autophagy in degradation of Aβ (Simonovitch et al, 2016, 2019).

Microglia are the main responders to amyloid plaques, mounting a DAM response (disease associated microglial cell-state response) (Keren-Shaul et al, 2017), which also causes the induction of APOE expression in these cells. The DAM response is dependent on APOE expression (Krasemann et al, 2017; Sala Frigerio et al, 2019) but it was thought that APOE expression in microglia was essential. Recent data also suggests a cell autonomous role of microglia-specific APOE4 expression in blocking the DAM response, compared to APOE3 (Yin et al, 2023b; Liu et al, 2023). In our model, we have presented non-cell autonomous effect of different astrocyte-specific APOE isoforms on microglial response to amyloid pathology. At morphological level, we demonstrated increased microglial clustering (Fig. 4A), and specific staining of DAM markers (Fig. 4B) in *Apoe*⁻/⁻ microglia surrounding the amyloid plaques. At transcriptional level, microglia from the APOE4 group upregulated several DAM markers (*Cd74, Cst7, Cd63, Cd9, Ccl6*), complement cascade genes (*C1qa, C1qb, C1qc*) and the pro-inflammatory gene *Nfkbia* (Sousa et al, 2018), similar to transcriptomic changes in microglia from *APOE4* AD individuals (Serrano-Pozo et al, 2021). In both astrocytes and microglia, there was an enrichment of interferon response and antigen-presentation genes in the APOE3 dataset. This was accompanied with limited plaque formation, compact plaques and low levels of oligomeric Aβ in the mice brain. Recently it was reported that conditional knockout of microRNA miR155 in microglia in APP/PS1 mice leads to activation of interferon-mediated signalling,

which was associated with increased phagocytosis and plaque compaction (Yin et al, 2023a). APOE expression and Aβ pathology are closely associated in our model which makes it difficult to determine the factors contributing directly or indirectly to the DEG changes seen in microglia and astrocytes. Future studies comparing the effect of APOE deletion in AD versus wild-type mouse models should help to confirm these findings. Moreover, studying the effect of APOE isoforms in wild-type mice on neuronal development and synapse formation, will be of interest as APOE is known to influence synaptic density and to cause behavioural defects in AD mouse (Dumanis et al, 2009; Klein et al, 2010; Koffie et al, 2012).

Depletion of microglia in our model, demonstrated that even though astrocyte-derived APOE is sufficient to induce plaque formation by 6 months of age, there was still a significant decrease in amyloid level in the absence of microglia, suggesting that they modulate amyloid plaque formation. We also observed increased levels of APOE in the microglia-depleted mouse brains, irrespective of the isoform. APOE co-aggregates with soluble Aβ species in vitro (Xia et al, 2024). Thus, it is possible that APOE-Aβ co-aggregates are taken up by microglia for clearance, and when microglia is depleted, this leads to increased levels of APOE in the mouse brain.

Limitations of the study include the AAV-mediated transgene expression, which results in overexpression of APOE under a non-native promoter. However, this approach allowed to establish the dose-dependent effects of different APOE isoforms on amyloid deposition, helping to understand the effect of dose versus isoform. Additionally, we did not explore the lipidation status of different APOE isoforms, which influences Aβ pathology (Koldamova et al, 2005; Wahrle et al, 2005). We also did not investigate possible gender- and ethnicity-specific effects of the different APOE isoforms. Previous reports have demonstrated that gender can affect AD pathology, microglia response and APOE expression (Stephen et al, 2019; Thomas et al, 2016; Sala Frigerio et al, 2019), while the protective and disease causing effects of APOE2 and APOE4 are variable in different ethnic populations (Farrer et al, 1997; Naslavsky et al, 2022). Finally, it was not possible to isolate microglia from the AAV-transduced regions only. This has likely diluted the effect size of changes we see in the microglia at transcriptomic level. This problem was not present for astrocytes where we could use the reporter to specifically enrich for the transduced cells.

In conclusion, we have investigated the cell autonomous and non-cell autonomous effects of the different APOE isoforms on astrocytes and microglia, respectively. It is clear that APOE plays an important role in the interaction between these cell types, with astrocytic APOE being a major driver of the amyloid plaque pathology which in its turn modifies the microglia response. Therefore, we propose that *APOE4* is a "core" gene (Liu et al, 2019) of AD pathogenesis in the sense that the strong effect on amyloid pathology, a core pathway of AD, explains the strong specific association with AD (odds ratio ~3.7–14.5). The other mechanisms of APOE4 dysfunction probably explain the association with neurodegenerative diseases in general, such as Parkinson's disease (odds ratio ~1.6–4.19) and Lewy body dementia (odds ratio ~2.4–6.1) (Bras et al, 2014; Guerreiro et al, 2018; Huang et al, 2006; Irwin et al, 2012; Tropea et al, 2018; Tsuang et al, 2013); TDP-43 proteinopathy (odds ratio ~2) (Josephs et al, 2014; Wennberg et al, 2018; Yang et al, 2018); and vascular dementia (odds ratio ~3.13) (Davidson et al, 2006; Sun et al, 2015). Several groups have suggested lowering of APOE expression with APOE-antibodies (Gratuze et al, 2022; Liao et al, 2018;

Xiong et al, 2021) or using antisense therapy (Huynh et al, 2017) as a therapeutic option for AD. Our work shows that even in a relative circumscribed and limited part of Alzheimer's disease pathogenesis, i.e., the formation of amyloid plaques and the microglia response to it, the different isoforms of APOE act in diverse and unanticipated ways, affecting amyloid plaques, Aβ soluble aggregates, but also the cellular reactions of microglia and astrocytes. We summarized above also the effects of APOE on neurons and blood vessels. A downregulation of APOE expression is not mimicking the protective effects of APOE2, and APOE2 protective action is complex and remains poorly understood. Even with increasing understanding of the pleiotropic role of APOE in the pathogenesis of Alzheimer's disease it remains difficult to define a precise therapeutic hypothesis based on the deep mechanistic understanding of the role of the different APOE isoforms in the disease.

# Methods

**Reagents and tools table**

| Reagent/Resource | Reference or Source | Identifier or Catalog Number |
|---|---|---|
| **Experimental Models** | | |
| C57Bl6/J (M. Musculus) | Jackson Lab | |
| **Recombinant DNA** | | |
| pAAV-hALDH1L1-mCherry | VectorBuilder | VB181123-1160msp |
| pAAV-hALDH1L1-APOE2-mCherry | VectorBuilder | VB200713-1259snq |
| pAAV-hALDH1L1-APOE3-mCherry | VectorBuilder | VB200316-1163eeg |
| pAAV-hALDH1L1-APOE4-mCherry | VectorBuilder | VB200713-1260frb |
| **Antibodies** | | |
| mouse biotin anti-β-Amyloid, 6E10 | Biolegend | 803007 |
| mouse Alexa 647 anti-β-Amyloid, 6E10 | Biolegend | 803020 |
| rabbit anti-Iba1 (1:500) | Wako | 019-19741 |
| guinea pig anti-NeuN (1:300) | Synaptic Systems | 266-004 |
| mouse anti-Apc (1:200) | Millipore | OP80 |
| guinea pig anti-GFAP (1:1000) | Synaptic Systems | 173-004 |
| guinea pig anti-Iba1 (1:500) | Synaptic Systems | 234-004 |
| rabbit anti-Apoe (1:500) | Atlas Antibodies | HPA065539 |
| rabbit anti-Ctsd (1:200) | Santa-cruz | sc10725 |
| rat anti-Cd68 (1:100) | Biolegend | 137001 |
| donkey anti-rabbit Alexa 488 (1:500) | Invitrogen | A21206 |

| Reagent/Resource | Reference or Source | Identifier or Catalog Number |
|---|---|---|
| donkey anti-rat Alexa 488 (1:500) | Invitrogen | A21208 |
| donkey anti-mouse Alexa 488 (1:500) | Invitrogen | A21202 |
| donkey anti-rabbit Alexa 594 (1:500) | Invitrogen | A21207 |
| donkey anti-guinea pig Cy™5 (1:500) | Jackson Immuno | 706-175-148 |
| PE anti-CD11b (1:50) | Miltenyi | 130-113-806 |
| BV421 anti-CD45 (1:500) | BD Biosciences | 563890 |
| APC anti-ACSA2 (1:200), | Miltenyi | 130-116-142, |
| BV421 anti-CD11b (1:500) | BD Biosciences | 562605 |
| PE anti-O4 (1:50) | Miltenyi | 130-117-357 |
| TotalSeq™-A cell hashing antibodies (1:500) | Biolegend | |
| **Oligonucleotides and other sequence-based reagents** | | |
| Gentoyping PCR primers | This study | Methods and Protocols (Mice) |
| qPCR primers | This study | Methods and Protocols (RNA extraction and qPCR) |
| **Chemicals, Enzymes and other reagents** | | |
| Cas9 HiFi protein | Integrated DNA Technologies | 1081061 |
| CRISPR crRNA | Integrated DNA Technologies | 1072533 |
| PLX3397/Pexidartinib | MedChemExpress | HY-16749 |
| cOmplete™ protease inhibitor cocktail | Roche | 5056489001 |
| TRI reagent | Sigma | T9424 |
| Superscript II reverse transcriptase | Thermo Fisher Scientific | 18064071 |
| SensiFast Sybr No-Rox | Bioline | BIO-98020 |
| Read Buffer T (4x) | MSD | R92TD-1 |
| Leupeptin | Cayman | 14716 |
| Aprotinin | Sigma | L5793-5MG |
| Pepstatin | Sigma | P5318-5MG |
| Neutravidin | Thermo Fisher | 31000 |
| Pefabloc SC protease inhibitor | Merck | 1429868001 |
| PhosSTOP™ phosphatase inhibitor | Merck | 4906845001 |
| X-34 dye | Sigma | SML1954 |
| DAPI dye | Sigma | D9542 |
| Sodium Citrate Dihydrate | VWR | 470302-530 |
| Donkey serum | Jackson Immunolabs | 017-000-121 |

| Reagent/Resource | Reference or Source | Identifier or Catalog Number |
|---|---|---|
| Glycergel®, Aqueous Mounting Medium | Agilent | C0563 |
| Actinomycin D | Sigma | A1410-5MG |
| Percoll | GE Healthcare | 17-5445-02 |
| FcR blocking solution | Miltenyi | 130-092-575 |
| eFluor 780 dye | Thermo Fisher Scientific | 65-0865-14 |
| **Software** | | |
| CRISPOR web tool | http://crispor.tefor.net/ | |
| ImageJ v1.53t | https://imagej.net/software/fiji/ | |
| Bedtools v2.27.1 | https://github.com/arq5x/bedtools2 | |
| Cellranger v6.1.2 | https://www.10xgenomics.com/support/software/cell-ranger/latest | |
| R v4.3.3 | https://www.r-project.org/ | |
| g:Profiler web tool | https://biit.cs.ut.ee/gprofiler/gost | |
| Graphpad Prism v10.0.2 | https://www.graphpad.com/ | |
| **Other** | | |
| NEPA21 Electroporator | Nepagene | |
| FastPrep-24 Homogeniser | MP Biomedicals | |
| Beckman Optima TLX centrifuge | Beckman Coulter | |
| 96-well SECTOR plates | MSD | L15XA-3 |
| R-PLEX Human APOE Assay kit | MSD | K1512IR-2 |
| VT1000S vibratome | Leica | |
| Nikon AX microscope | Nikon | |
| Miltenyi neural tissue dissociation kit P | Miltenyi | 130-092-628 |
| MACSQuant Tyto Cell Sorter | Miltenyi | |
| FACSAria Fusion Cell Sorter | BD Biosciences | |
| Chromium Next GEM Single Cell 3′ Reagent Kits v3.1 | 10X Genomics | CG000204 |

## Methods and protocols

### Mice

The mouse *Apoe* gene is localized on chromosome 7 and consists of four exons and three introns, exon 2–4 contains the coding sequences. *Apoe* knock-out mice were generated using two guides to target the second intron (5′-GUCUCGGCUCUGAACUACAU-3′)

and the fourth exon (5′-ACUCGGGCAGUACCGCAACG-3′). mRNA guides were selected using the CRISPOR web tool (http://crispor.tefor.net/). Ribonucleoproteins (RNPs) containing 3 µM purified Cas9 HiFi protein and 4 µM CRISPR RNA crRNA and trans activating crRNA were electroporated (Nepa21) into Rag2tm1.1Cgn zygotes by the CBD Mouse Expertise Unit of KU Leuven. Out of 113 embryo's electroporated, 91 embryos were transplanted to foster mothers, 10 pups were born and 3 were positive for the deletion. To establish the *Apoe* knock-out allele (*Apoe*em1Bdes), one founder mouse carrying an 846 bp deletion was selected and crossed with the original strain twice in order to cross out potential off targets events while keeping the background and genetic mutations of the strain. For standard PCR genotyping of the *Apoe* allele, primers 5′-CTCACGGATGGGCACTCAC-3′ and 5′-GCTCCCAAGTCACACAAGAA-3′ were used, resulting in an amplicon of 331 bp for the knock-out allele and in an amplicon of 1117 bp for the wild-type allele.

To generate *App*tm3.1Tcs *Apoe*em1Bdes *Rag2*tm1.1Cgn mice (in this study named as *App*NL-G-F x *Apoe*−/− x *Rag2*−/−) and *App*tm3.1Tcs *Apoe*em1Bdes mice (named as *App*NL-G-F x *Apoe*−/−), the mice described above were crossed with *App*tm3.1Tcs knock-in mice harbouring the Swedish, Iberian and Arctic FAD mutation as well as the humanized Abeta sequence (Saito et al, 2014). All the data presented in this study are from *App*NL-G-F x *Apoe*−/− mice. For a pilot study to check viral transduction efficiency and specificity, *App*NL-G-F x *Apoe*−/− x *Rag2*−/− mice were used, and data from these are presented in Appendix Figures: S1A, S1E, S1F, S2A and S2B.

Mice were kept on a C57Bl6J background and both females and males were included in the study. Mice were housed in cages enriched with wood wool and shavings as bedding and given access to water and food ad libitum with a circadian rhythm of 7 am–9 pm light and 9 pm–7 am dark. Mice were randomly assigned to experimental groups (unblinded). All experiments were conducted in accordance with institutional and national guidelines and regulations, following approval of Ethical Committee for Animal Experimentation at the University of Leuven (KU Leuven) (permit number P007/2017 and P072/2019).

### Intracerebroventricular viral vector injections

Adeno-associated viruses (AAVs) were used to express different human APOE isoforms in mouse astrocytes. Plasmid constructs were designed to express different human APOE alleles under astrocyte specific promoter Aldh1l1. Plasmids were created and packaged into AAVs by VectorBuilder (www.vectorbuilder.com). We used AAV2/8 serotype (>$10^{13}$ GC/ml) and the following plasmid constructs: pAAV-hALDH1L1-mCherry (Vectorbuilder ID: VB181123-1160msp), pAAV-hALDH1L1-APOE2-mCherry (Vectorbuilder ID: VB200713-1259snq), pAAV-hALDH1L1-APOE3-mCherry (Vectorbuilder ID: VB200316-1163eeg), pAAV-hALDH1L1-APOE4-mCherry (Vectorbuilder ID: VB200713-1260frb). For intracerebroventricular (ICV) injections, on postnatal day 2/3, the mouse pups were anaesthetized by hypothermia and approximately $2 \times 10^{10}$ infectious particles of AAV (1 µl per site) were injected into the ventricles with Hamilton syringes bilaterally: halfway between lambda and bregma, 1 mm bilaterally from the midline and 3 mm from the pial surface (Kim et al, 2014). Transduced pups were returned to their home cages until weaning age.

### PLX3397 treatment for microglia depletion

For depletion of microglia from mouse brain (Elmore et al, 2014; Spangenberg et al, 2016), PLX3397/Pexidartinib was formulated in standard chow (Bio Services NL). Modified chow containing PLX3397 (600 mg/kg of chow) was administered to the mice for 4 months, starting at 2 months of age before onset of plaque deposition.

### RNA extraction and qPCR

For sample collection, mice were euthanized using an overdose of carbon dioxide and perfused with ice-cold 1x DPBS. Brains were surgically removed and snap-frozen in liquid nitrogen. Frozen brain tissue was homogenized in PBS (supplemented with cOmplete™ protease inhibitor cocktail), in a beadmill using FastPrep (MP Biomedicals). TRI reagent was added to the supernatant to lyse any remaining cells and 1-bromo-3-chloropropane was used for phase separation. The solution was centrifuged at 12,000 rcf for 15 min at 4 °C. RNA in the aqueous phase was precipitated in 2-propanol. After centrifugation at 12,000 rcf for 10 min at 4 °C, the RNA pellet was washed with 75% ethanol and solubilized using RNAse-free water.

Reverse transcription of mRNA was done using the Superscript II reverse transcriptase. Real-time semi-quantitative PCR was performed using the SensiFast Sybr No-Rox kit for cDNA. Following forward and reverse primers were used for human *APOE*: 5′-GTTGCTGGTCACATTCCTGG-3′ and 3′-GCAGG-TAATCCCAAAAGCGAC-5′. Following forward and reverse primers were used for housekeeping genes mouse *Gapdh*: 5′-TTGATGGCAACAATCTCCAC-3′ and 3′-CGTCCCGTAGA-CAAAATGGT-5′, and *Actin*: 5′- GTGACGTTGACATCCG-TAAAGA-3′ and 3′-GCCGGACTCATCGTACTCC-5′.

### Protein extraction and MSD-ELISA

For sample collection, mice were euthanized using an overdose of carbon dioxide and perfused with ice-cold 1x DPBS. Brains were surgically removed and snap-frozen in liquid nitrogen. Frozen brain tissue was homogenized in PBS (supplemented with cOmplete™ protease inhibitor cocktail), in a beadmill using FastPrep-24. The homogenate was centrifuged at 5000 rcf for 5 min at 4 °C. The supernatant was ultra-centrifuged at 55,000 rpm (Rotor TLA-110, Beckman Optima TLX) for 60 min at 4 °C and the supernatant was used as PBS-soluble fraction. For guanidinium chloride (GuHCl) based extraction, the pellet was resuspended in 6 M GuHCl, 50 mM Tris-HCl (pH 7.6), (supplemented with cOmplete™ protease inhibitor cocktail) and sonicated using a micro-tip for 30 s at 10% amplitude (Branson). After incubation for 1 h at 25 °C, the sample was ultracentrifuged at 70,000 rpm (Rotor TLA-110, Beckman Optima TLX). The supernatant was then diluted to 0.1 M GuHCl and used as guanidinium-soluble fraction.

Aβ$_{42}$ and hAPOE levels were quantified on MSD 96-well plates (Meso Scale Discovery) using MSD-ELISA. For Aβ$_{42}$, standard 96-well SECTOR plates (MSD) were coated with LTDA_Aβ42 capture antibody (in house generated mouse monoclonal antibody against Aβ$_{42}$ in collaboration with Maarten Dewilde). After overnight incubation at 4 °C, the plates were rinsed with PBS 0.05% Tween 20 and blocked with casein buffer (PBS with 1% casein, pH 7.4) for 4 h

at room temperature. Synthetic human $A\beta_{42}$ dilutions were used as standards. Standards and samples were preincubated with LTDA_-hA$\beta$N labelled with a sulfo-TAG detection antibody (in house generated mouse monoclonal antibody against the N-terminal sequence of human A$\beta$, in collaboration with Maarten Dewilde), in casein buffer for 5 min at room temperature. The blocked assay plate was rinsed five times with PBS 0.05% Tween 20, and the sample and detection antibody mix were added and incubated overnight at 4 °C. For hAPOE, we used R-PLEX Human APOE Assay kit and total APOE level in each sample was measured as sum of APOE level in PBS-soluble and Guanidine-soluble fractions. Briefly, GOLD 96-well Small Spot Streptavidin SECTOR plates (MSD) were coated with biotinylated capture antibody, for 1 h at room temperature. The plates were rinsed with PBS 0.05% Tween 20, followed by addition of standards and samples diluted in assay buffer and incubated for 1 h at room temperature. After rinsing, detection antibody was added and incubated for 1 h at room temperature. For measurement, the plates were rinsed, and 2x Read T buffer (MSD) (for A$\beta$42) or GOLD Read Buffer B (MSD) (for hAPOE) was added, and plates were read on an MSD Sector Imager 2400 A.

## SiMPull assay for soluble A$\beta$ aggregate quantification

Soluble oligomeric aggregates of A$\beta$ were extracted from frozen brain tissue. Post thawing, soaking extraction was performed with a protocol modified from a previously published method (Hong et al, 2018). Briefly, brain samples were added to 1.5 mL Eppendorf Protein LoBind® tubes and soaking buffer containing 124 mM NaCl, 2.8 mM KCl, 1.25 mM $NaH_2PO_4$, 26 mM $NaHCO_3$, 5 mM EDTA, 1 mM EGTA, 5 μg/mL leupeptin, 5 μg/mL aprotinin, 2 μg/mL pepstatin, 5 mM NaF, 20 μg/mL Pefabloc® SC protease inhibitor and PhosSTOPTM phosphatase inhibitor tablet was added to each sample. Samples were placed on a rotary shaker for 4.5 h at 4 °C and then centrifuged at 2000 rcf for 10 min at 4 °C. Following the initial centrifugation, the supernatant was centrifuged again for 2 h at 4 °C, at 17000 rcf. Then the supernatant was aliquoted, snap-frozen in liquid nitrogen, and stored at −80 °C.

Single-molecule pull-down coverslip preparation was performed as previously described (Emin et al, 2022). Briefly, neutravidin (0.2 mg/ml) in PBS with 0.05% Tween 20 (PBST) was added to glass coverslips covalently mounted with polyethylene glycol (PEG) and biotin for 10 min, followed by two wash steps with PBST and once with PBS with 1% Tween 20 (1%T). Afterwards, biotinylated 6E10 (10 nM) was added for 15 min, followed by two wash steps with PBST and once with 1%T. The soaked brain samples were added and incubated overnight at 4 °C followed by two wash steps with PBST and once with 1%T. The coverslips were then incubated with labelled Alexa Fluoro 647 conjugated 6E10 (500 pM) for 45 min, followed by two wash steps with PBST and once with 1%T. A second PDMS gasket was added to the slide to increase the well volume and 8 μl of PBS was added to prevent drying of the wells.

Then the coverslips were imaged on a purpose built (Emin et al, 2022) TIRF microscope using a 638 nm laser, with 50 ms of exposure for 50 frames for each field of view. The images were then reconstructed using the ComDet plugin in Fiji (https://github.com/UU-cellbiology/ComDet) and the number of spots were counted for each field of view with the same size and brightness thresholds

applied to all images. A minimum of 9 field of views were acquired from each well.

## Immunohistochemistry (IHC)

For sample collection, mice were euthanized using an overdose of carbon dioxide and perfused with ice-cold 1x DPBS. Brains were surgically removed, post-fixed in 4% paraformaldehyde (PFA) overnight and cut into 40-μm-thick sections on vibratome. A$\beta$ plaques were detected by staining with X-34. Nuclei staining was performed using DAPI. The antibodies (with dilutions) used for immunostaining are indicated in the Reagents and Tools table above. Antigen retrieval was performed by microwave boiling the sections in 10 mM tri-Sodium Citrate buffer pH 6.0. After rinsing with PBS, the free-floating sections were blocked and permeabilized using 10% donkey serum and 0.3% Triton X-100. Primary antibodies, diluted to required concentrations, were then added to the sections and incubated overnight at 4 °C. The sections were washed using PBS and secondary antibodies conjugated with fluorophores were added for 2 h at room temperature. Finally, after washing with PBS, sections were mounted on the slides with Glycergel. Confocal images with z-stacks (1.5 μm/1 μm step size) were obtained using a Nikon AX inverted confocal microscope driven by NIS (4.30) software. All images were acquired with 20x (field of view: 880.42 × 880.42 μm) or 40x (field of view: 438.88 × 438.88 μm) or 60x objectives (field of view: 292.3 × 292.3 μm). For excitation 405 nm, 488 nm, 561 nm, 640 nm laser lines were used. Raw images were processed in ImageJ (1.53t) software. Z-stacks were converted to Maximum Intensity Projections for quantifications, unless otherwise specified. For representative images provided in Figures, brightness was adjusted, if necessary (by equal measure across all experimental groups), for better visualization.

## Plaque IHC quantification

Flattened tiff files from 20x and 40x images were used for quantifying X-34 positive amyloid plaques. Individual plaques larger than ~4 micron were segmented from 20x and 40x IHC image using an interactive machine learning approach, implemented through ImageJ and the Labkit plugin (Arzt et al, 2022; Schindelin et al, 2012). In this method a random forest pixel classifier was trained, with $\sum_{n=0}^{3} 2^n$ values for σ and the original image, gaussian blur, difference of gaussians, gaussian gradient magnitude, Laplacian of gaussian, hessian eigenvalues filters of the image as inputs. After segmenting, image features were extracted using a python pipeline. Briefly, the size, location and grey-scale co-occurrence matrix of the segmented plaques were calculated using the scikit-learn imaging library (Van Der Walt et al, 2014). Next, Angular Second Moment (ASM) of the grey-scale-co-occurrence matrix was calculated as an estimate of amyloid plaque compactness (Haralick et al, 1973). ASM is a measure of local image homogeneity, a higher value being consistent with a more homogenous plaque with fewer transitions between light and dark zones. Haralick's features are a well-established set of rotation invariant image statistics that have been used in medical classification issues across many fields and imaging modalities (Doyle et al, 2008; Gnep et al, 2017; Hapsari et al, 2022). Among the different Haralick's features, ASM was found to be the statistic most

congruent with the visual differences between more diffuse and more dense plaques.

## Microglia IHC quantification

3D z-stacks from raw 40x images were used for segmenting Iba1-positive microglia. Individual microglia were segmented using a custom pipeline combining a state-of-the-art machine learning model with established image processing methods. This more tailored solution was chosen over more prevalent methods like Imaris as it was unable to accurately segment physically touching and overlapping cell bodies with intertwining branches, in the absence of nuclei staining. Instead, the established method Labkit was found to be particularly good at identifying which pixels belonged to microglia but failed to distinguish individual microglia from each-other, especially around plaques, where microglia tend to cluster densely. On the other hand, we found that state-of-the-art Cellpose machine learning β was unable to segment the full cell including its processes (Stringer et al, 2021). To achieve the highest accuracy, a pipeline was built to combine the strengths of both methods. First, the pre-trained 'cyto' Cellpose model was manually fine-tuned to segment the cell bodies and find their centroids. Secondly, all pixels belonging to microglia were identified using Labkit. A classifier was trained with $\sum_{n=1}^{6} 2^n$ for σ and the original image, gaussian blur, difference of gaussians, gaussian gradient magnitude, Laplacian of gaussian, hessian eigenvalues, mean and max filters of the image as inputs. The resulting semantic segmentation mask was tentatively split into cell-bodies and processes. We then performed minor closing operations on the process mask to fix small continuity errors which would otherwise cause interruptions in the microglial processes. These two masks were then recombined.

Both the Labkit segmentation mask and the Cellpose segmentation mask were then subjected to the Exact Euclidean distance transformation which resulted in masks where the value of each pixel equals its distance to the edge of the binary mask. These two distance masks were combined in a weighted sum so the Cellpose masked areas were of greater value then the Labkit areas. Finally, the watershedding algorithm was performed using the Cellpose centroids as local minima, to retrieve individually labelled microglia (Appendix Fig. S3A). Individually labelled microglia were then digitally traced using the APP2 plugin for VAA3D and the convex area of the cell, as well as the total length of its processes were calculated to quantify their morphological changes (Peng et al, 2010; Xiao and Peng, 2013). For counting number of microglia associated to each plaque, 3d z-stacks were flattened to tiffs post microglia segmentation. X-34-positive plaques were thresholded and segmented as described above. Any microglia whose centroid was located within X-34-positive thresholded plaque area and up to 5 µm outside a plaque's edge was considered as plaque-associated microglia. For each microglia, the mean distance to its three nearest neighbours were also calculated as a measure of clustering.

## FACS isolation of microglia and astrocytes

For sample collection, 6-month-old mice were anaesthetized using $CO_2$ chamber and perfused with ice-cold 1x DPBS. Each mouse brain, excluding cerebellum and olfactory bulbs were dissected and placed in FACS buffer (1x DPBS, 2% FCS and 2 mM EDTA) +

5 µM Actinomycin D (ActD) for transcriptomics. Brains were mechanically and enzymatically dissociated using Miltenyi neural tissue dissociation kit P supplemented with 5 µM ActD. Next, samples were passed through a 70 µm strainer (BD2 Falcon), washed in 10 ml of ice-cold FACS buffer + 5 µM ActD and spun at 300 rcf for 15 min at 4 °C. Note that 5 µM ActD was kept during collection and enzymatic dissociation of the tissue to preserve transcriptional state and prevent artificial induction stress-response genes during tissue dissociation (Marsh et al, 2022; Perocchi et al, 2007). ActD was removed from the myelin removal step to prevent toxicity derived from long-term exposure. Following dissociation, myelin was removed by resuspending pelleted cells into 30% isotonic Percoll and centrifuging at 300 rcf for 15 min at 4 °C. Accumulating layers of myelin and cellular debris were discarded and Fc receptors were blocked in FcR blocking solution (1:10) in cold FACS buffer for 10 min at 4 °C. Next, cells were washed in 5 ml of FACS buffer and pelleted cells split into two suspensions (one each for astrocyte and microglia sorting), and incubated with the following antibodies: for microglia —PE-CD11b (1:50), BV421-CD45 (1:500); for astrocytes—APC-ACSA2 (1:200), BV421-CD11b (1:500), PE-O4 (1:50); in cold FACS buffer for 30 min at 4 °C. TotalSeq™-A cell hashing antibodies (1:500) and viability dye (eFluor 780) were also added to both fractions during the incubation step. After incubation, cells were washed, and the pellet was resuspended in 500 µl of FACS buffer and passed through 35 µm strainer prior sorting. For isolating microglia, cell suspension was loaded into the input chamber of MACSQuant Tyto Cartridge and cells were sorted on MACSQuant Tyto Cell Sorter at 4 °C, based on CD11b-positive and CD45-low populations. For isolating AAV-transduced astrocytes, cells were sorted on Aria Fusion, based on negative gating for O4 and CD11b populations, followed by positive gating for double-positive ACSA2 and mCherry populations. FACS data was analysed using FCS express software.

## Single-cell library preparation and sequencing

For single-cell RNA sequencing microglia and astrocytes, 20,000–30,000 cells of each type from each mouse were sorted as described above and diluted to a final concentration of 1000 cells/µl. All the samples were individually hashed using TotalSeq™-A cell hashing antibodies, 3000 cells/animal were pooled and loaded onto the Chromium Next GEM Chip G (PN #2000177). The DNA library preparations were generated following manufacturers' instructions (Chromium Next GEM Single Cell 3′ Reagent Kits v3.1). In parallel the hashtag oligo libraries were prepared according to manufacturers' instructions (BioLegend - TotalSeq™-A Antibodies and Cell Hashing with 10x Single Cell 3′ Reagent Kit v3.1 Protocol) using 16cycles for the index PCR. From the 4 experimental groups (APOEKO, APOE2, AOE3, and APOE4), a total of 8 libraries (APOEKO_astrocytes, APOE2_astrocytes, APOE3_astrocytes, APOE4_astrocytes, APOEKO_microglia, APOE2_microglia, APOE3_microglia, APOE4_microglia), containing 22 biological replicates ($n$(APOEKO) = 2, $n$(APOE2) = 3, $n$(APOE3) = 3, $n$(APOE4) = 3), were sequenced (service provided by BGI Genomics, China) targeting 90% mRNA and 10% hashtag oligo library (50,000 reads/cell) on a DNBSEQ-G400 (MGI Tech) platform with the recommended read lengths by 10X Genomics workflow.

## Analysis of single-cell RNA sequencing datasets

### Alignment and analysis pipeline

Raw BCL files were aligned to a modified GRCm39 mouse genome, which was appended with the human APOE gene. We used Bedtools 2.27.1 to extract the human *APOE* sequence and a thousand base buffer from GRCh38 (Quinlan and Hall, 2010). This was then appended to the murine GRCm39 genome in an artificial chromosome. The resulting genome was then filtered using the *mkgtf* function from Cellranger (6.1.2), in accordance with 10x guidelines (Zheng et al, 2017) and subsequently processed into a functional reference using cellranger *mkref*. Finally, each of the eight scRNAseq libraries were aligned to a custom genome using Cellranger's count function which matches each transcript to its corresponding gene using the STAR aligner (Dobin et al, 2013). Raw count matrices were imported in R (v4.3.3) for data analysis. Datasets were analysed using R packages: *Seurat (4.4)* (Hao et al, 2021), *scater (1.30.1)* (McCarthy et al, 2017), *clustree (0.5.1)* (Zappia and Oshlack, 2018), and *SuperExactTest* (1.1) (Wang et al, 2015). Visualizations were done using functions from *Seurat, SuperExactTest* or *ggplot2 (3.5)*.

### Quality control of cells and samples

In this overexpression model, since APOE is under a constitutive promoter based on viral transduction, and hence not a part of the normal transcriptional regulatory networks, we excluded human APOE gene expression from both astrocyte and microglia datasets prior to the following steps.

For each library of microglia, low-level quality control was done by first filtering out cells with <200 genes detected and genes expressed in less than 3 cells, from the raw counts matrix. Subsequent filtering involved three steps. Step 1: Cell hashing information was added to each library using *Seurat::HTODemux()* function to assign each cell as either singlets, doublets or negatives. Only singlets were retained, as doublets and singlets had abnormally high and low number of reads, respectively, indicative of poor-quality cells. Step 2: Libraries from the 4 experimental groups were merged using *Seurat::merge()* function, to have in total 23,495 cells (Appendix Fig. S6A). Cell clustering was performed using the pipeline described below. Cell type annotation was done using *seurat::AddModuleScore()* to assign cell scores based on previously identified signatures for different brain cell types (Zeisel et al, 2018). ~9% of cells had non-microglial identity with half of those having macrophage identity. Contaminating cell types were identified (Appendix Fig. S6A, B) as macrophages (microglia QC Cluster 4), endothelial cells (microglia QC Cluster 5), mix of astrocyte and microglial cells (microglia QC Cluster 6), monocytes (microglia QC Clusters 7 and 9), and oligodendrocytes (microglia QC Cluster 10). Out of the rest ~10% separated out as a cluster (microglia QC Cluster 2) with low number of reads and gene count (Appendix Fig. S6C). Step 3: After filtering out these clusters, each library was split into individual ones and quality control was refined with an additional step using *scater::isOutlier()* function. Here cells with number of reads or genes or percentage of mitochondrial genes (%mito), outside $n$ median absolute deviation ($n(\text{reads}) = +/- 4$, $n(\text{genes}) = -4$, $n(\%\text{mito}) = +5$) from library median, were removed. Finally, 18,569 microglia were retained for analysis as good-quality cells.

For astrocytes, the same steps as described above with few modifications were executed. Step 1: Cell hashing did not work efficiently for astrocytes, as a good proportion of cells identified as

negatives had read counts in the same range as Singlet cells. Thus, we used the other two steps to filter out low-quality cells. Step 2: Libraries from the 4 experimental groups were merged to have in total 32,982 cells (Appendix Fig. S6D). After cell type annotation as described above, ~7% of cells had non-astrocyte identity. Contaminating cell types were identified (Appendix Fig. S6D, E) as ependymal cells (astrocyte QC Cluster 5), mural cells (astrocyte QC Cluster 7), microglia (astrocyte QC Cluster 9), endothelial cells (astrocyte QC Cluster 10) and a population of undetermined identity (astrocyte QC Cluster 8). Out of the rest, ~31% separated out as clusters (astrocyte QC Clusters 1 and 6) with low number of reads and gene count (Appendix Fig. S6F). Step 3: After filtering out these clusters, each libraries were split into individual ones using *Seurat::SplitObject()* and quality control was refined with an additional step using *scater::isOutlier()* function. Here cells with number of reads or genes or percentage of mitochondrial genes (% mito), outside $n$ median absolute deviation ($n(\text{reads}) = +/- 2.5$, $n(\text{genes}) = -2.5$, $n(\%\text{mito}) = +5$) from library median, were removed. Finally, 18,667 astrocytes were retained for analysis as good-quality cells.

### Library integration and cell clustering

To remove batch effects between libraries while cell clustering, the Seurat Object was split into individual libraries using *Seurat::SplitObject()*. Each libraries were individually normalized (*Seurat::NormalizeData*) and variable genes identified (*Seurat::FindVariableFeatures*). Genes that are repeatedly variable across libraries were selected for integration (*Seurat:: SelectIntegrationFeatures()*). Integration anchors were identified (Seurat::*FindIntegrationAnchors()*) and batch correction was performed (Seurat::*IntegrateData()*). After scaling and centering genes (Seurat::*ScaleData()*) in the integrated dataset, Principal Component Analysis (PCA) was performed. $n$ dimensions from the PCA ($n(\text{astrocytes}) = 50$; $n(\text{microglia}) = 40$) were selected for identifying clusters (*Seurat::FindNeighbors()* and *Seurat::FindClusters()*). To avoid over-clustering or under-clustering, multiple resolutions were used for the *FindCluster()* function and *clustree::clustree()* was used for selecting an optimum resolution (astrocytes = 0.7; microglia = 0.4). For visualization of clusters, dimensionality reduction by Uniform Manifold Approximation and Projection (UMAP), was performed with the *Seurat::RunUMAP()* function. Cluster markers for astrocytes and microglia were identified using *Seurat::FindAllMarkers()* function which calculates differential expression for each cluster against the rest. Astrocyte clusters were scored using *seurat::AddModuleScore()*, for markers from previously published genesets of astrocyte cell states (Zamanian et al, 2012; Hasel et al, 2021), plaque-induced genes (PIGs) (Chen et al, 2020) and AD risk genes (Bellenguez et al, 2022).

### Differential expression

For both astrocytes (telencephalon subtype) and microglia, differential expression was performed between the 4 experimental groups (each of APOEKO, APOE2 and APOE4 compared to APOE3), using *Seurat::FindMarkers()*. Genes expressed in at least 5% of cells were calculated. For calculating *p*-values, MAST was used which is a GLM-framework that treats cellular detection rate as a covariate. Since there were differences in the gender ratio between the groups, gender identity was regressed out during differential expression using *latent.vars* parameter. Post-analysis, genes with their adjusted $P$ value < 0.05 (post hoc, Bonferroni

correction) and $|\log(\text{fold-change})| > 0.2$ were considered as significant. *SuperExactTest* (Wang et al, 2015) was used to calculate statistical significance (Datasets EV3, EV6) and visualize overlap of differentially expressed genes while comparing each group with APOE3. Size of background gene population ($n$) for SuperExactTest was collected from total number of genes in the Seurat object for astrocytes and microglia library ($n$(astrocytes) = 25,321, $n$(microglia) = 23,819). Volcano plots and quadrant plots using ggplot2::*ggplot()* and dot plots using *Seurat::DotPlot()* were used to visualize differentially expressed genes between the groups.

### Gene Ontology (GO) Enrichment Analysis

For astrocytes and microglia, cluster markers identified, as mentioned above, was used to perform functional enrichment analysis. This was done using g:Profiler web interface, where the cluster markers were queried against GO biological process database, with g:SCS multiple testing correction method applying significance threshold of 0.05 (Kolberg et al, 2023). Bar plots using ggplot2::*ggplot()* was used to visualize the top 10 significantly enriched terms for astrocyte clusters a0 and a6.

## Statistical analysis

Number of samples included in each experiment is included in the results section/figure legends. Sample size estimation was based on pilot experiments. All statistical comparisons were performed in Graphpad Prism (v10.0.2) or R (v4.3.3). Statistical test used for each experiment is reported in the figure legends. For analysis where individual plaques or microglia were plotted as data points, mixed effects models were used to account for the variability between individual mice and Q-Q plots were used to assess the normality of the residuals, using the lme4 package for mixed effects models and the "estimated marginal means" package in R. For comparison of ASM values in Figs. 2 and 5, the linear mixed-effects model was used with mouse ID and diameter of plaques as confounding factors. ASM response variable was also log-transformed to improve the normality of the residuals. Differential abundance of plaque-associated microglia around individual plaques in Fig. 4 was tested using binomial generalized linear mixed-effects model with mouse ID as confounding factor. Morphological differences between plaque-associated microglia and non-plaque-associated microglia in Fig. EV1 was tested using linear mixed-effects model with mouse ID as confounding factor. Differences in total number of differentially expressed genes between groups (compared to APOE3) was tested using Fisher's exact test with Bonferroni correction of $p$-values. Enrichment of plaque-induced genes (PIGs) (Chen et al, 2020) in intersection of differentially expressed genes in microglia dataset was tested using hypergeometric test with Bonferroni correction of $p$-values.

## Data availability

Raw data used for this research article is available upon request to the corresponding authors. Single cell RNA sequencing data generated in this study are available at Gene Expression Omnibus (GEO) database number GSE252454.

The source data of this paper are collected in the following database record: biostudies:S-SCDT-10_1038-S44321-024-00162-7.

### The paper explained

#### Problem

The APOE4 allele increases the risk of Alzheimer's disease (AD) fourfold compared to APOE3, while APOE2 reduces the risk by half. Numerous hypotheses have been proposed regarding the role of APOE in AD, reflecting its pleiotropic effects in different cell types. In AD mouse models, the primary effect of APOE deficiency is the prevention of amyloid plaque formation. We investigated whether expressing APOE isotypes in astrocytes alone could restore amyloid plaque formation and assess the involvement of microglia.

#### Results

Astrocyte-derived APOE is sufficient to induce fibrillar amyloid plaque formation in *Apoe*-deficient mouse brain. Surprisingly, both APOE2 and APOE3 induced plaque formation to similar extents. Astrocyte-derived APOE also reduced soluble Aβ aggregates compared to *Apoe*-deficient brains. Astrocytes expressing APOE showed altered transcriptional responses, with potential impairment of autophagy and proteostasis in those expressing APOE4. Notably, microglia in Apoe-deficient mice showed clustering around amyloid plaques and a DAM-like response once astrocyte APOE expression was induced.

#### Impact

APOE from astrocytes is the primary driver of amyloid plaque formation in this AD model. While APOE4 increases plaque formation, APOE2 does not reduce it compared to APOE3, suggesting that APOE2's protective effect is not simply the inverse of APOE4's effect. This study highlights the central role of astrocytes and underscores the significant impact of APOE4 on plaque formation. Although APOE is a critical therapeutic target for AD, this research shows that developing effective treatments is complex. Downregulating APOE expression overlaps partially with both APOE4 and APOE2 effects, and APOE2 does not counteract APOE4 in amyloid pathology in the early disease stages.

## Peer review information

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

## Acknowledgements

We thank Veronique Hendrickx and Amber Claes for help with the mouse colonies and Dries T'Syen, Katrien Horré and Katleen Craessaerts for technical assistance. Mouse experiments were supported by CBD Mouse Expertise Unit (KU Leuven and VIB). Confocal microscopy was performed in the VIB Bio Imaging Core (LiMoNe). Part of the figure schematics was created with BioRender (biorender.com). This project received funding from the European Research Council (ERC) under the European Union's Horizon 2020 Research and Innovation Programme (grant agreement no. ERC-834682 CELLPHASE_AD). This work was also supported by funding from UKRI and the Medical Research Council (MR/Y014847/1) via the Dementia Research Institute. Further support was given by the Flanders Institute for Biotechnology (VIB vzw), a Methusalem grant from KU Leuven and the Flemish Government, the Fonds voor Wetenschappelijk Onderzoek, KU Leuven, The Queen Elisabeth Medical Foundation for Neurosciences, the Opening the Future campaign of the Leuven Universitair Fonds, The Belgian Alzheimer Research Foundation (SAO-FRA) and the Alzheimer's Association USA. BDS holds the Bax-Vanluffelen Chair for Alzheimer's Disease. DK is funded by UK Dementia Research Institute, which receives its funding from DRI Ltd. funded by the UK Medical Research Council and the Royal Society. AMA receives funding from the Ministerio de Ciencia e Innovación under

grant no. MCIN/AEI/10.13039/501100011033 (PID2021-125443OB-100 also by FEDER, UE and RYC2020-029494-I by FSE invierte en tu futuro), the Alzheimer's Association (AARG-21-850389), and the Basque Government (PIBA-2020-1-0030).

## Author contributions

**Pranav Preman**: Conceptualization; Data curation; Formal analysis; Investigation; Visualization; Methodology; Writing—original draft; Writing— review and editing. **Daan Moechars**: Data curation; Software; Formal analysis; Visualization; Methodology; Writing—review and editing. **Emre Fertan**: Investigation; Writing—review and editing. **Leen Wolfs**: Investigation. **Lutgarde Serneels**: Investigation; Methodology; Writing—review and editing. **Disha Shah**: Investigation; Methodology. **Jochen Lamote**: Resources; Investigation. **Suresh Poovathingal**: Resources; Investigation. **An Snellinx**: Investigation. **Renzo Mancuso**: Investigation; Writing—review and editing. **Sriram Balusu**: Investigation; Writing—review and editing. **David Klenerman**: Supervision; Validation; Writing—review and editing. **Amaia M Arranz**: Conceptualization; Supervision; Funding acquisition; Validation; Writing—review and editing. **Mark Fiers**: Resources; Data curation; Software; Formal analysis; Supervision; Validation; Writing—review and editing. **Bart De Strooper**: Conceptualization; Resources; Supervision; Funding acquisition; Validation; Writing—original draft; Project administration; Writing—review and editing.

Source data underlying figure panels in this paper may have individual authorship assigned. Where available, figure panel/source data authorship is listed in the following database record: biostudies:S-SCDT-10_1038-S44321-024-00162-7.

## Disclosure and competing interests statement

BDS is or has been a consultant for Eli Lilly, Biogen, Janssen Pharmaceutica, Eisai, AbbVie and other companies. BDS is an Editorial Board Member for EMBO Molecular Medicine. BDS is also a scientific founder of Augustine Therapeutics and a scientific founder and stockholder of Muna Therapeutics.

# Expanded View Figures

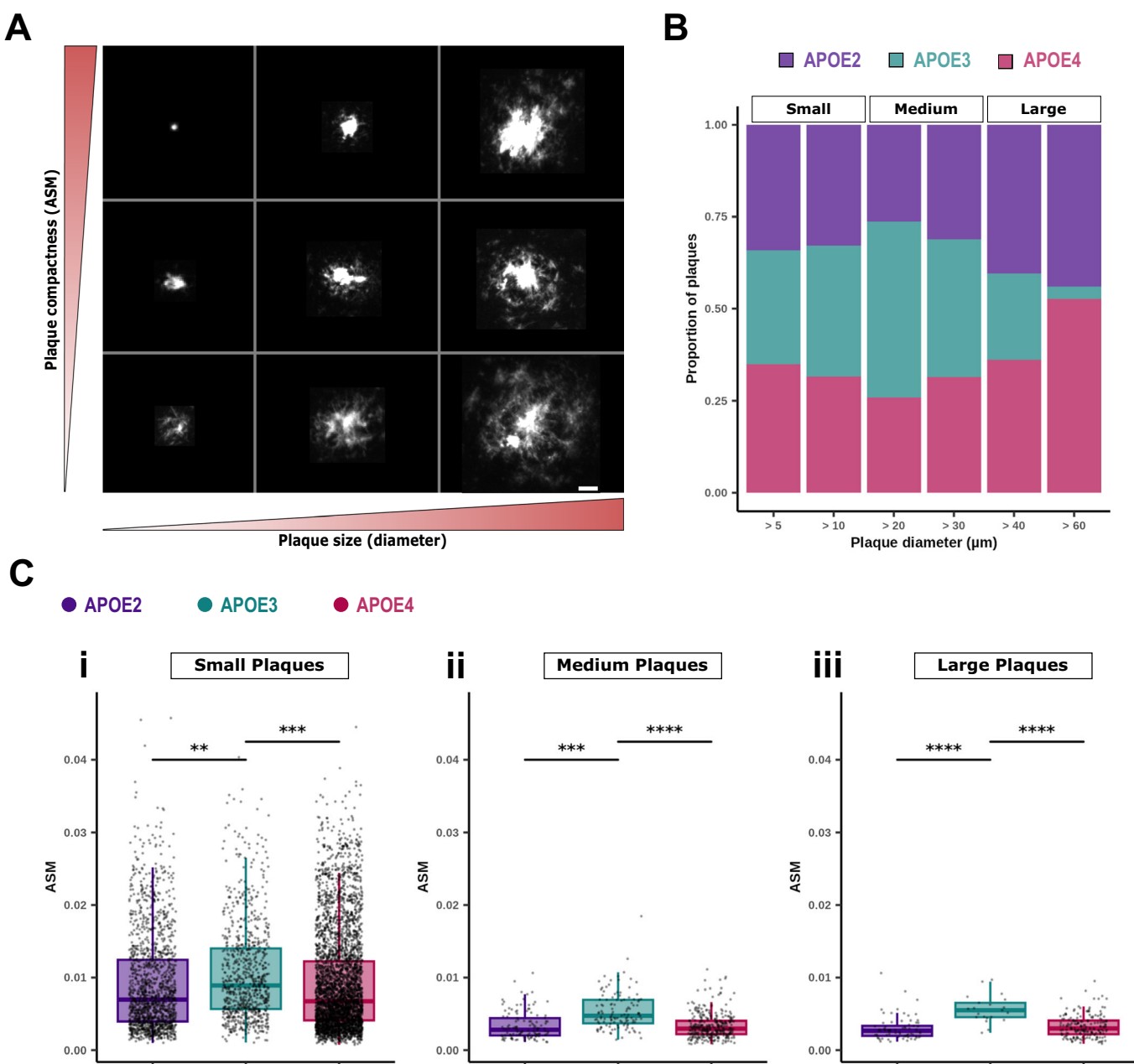

**Figure EV1. Astrocyte-derived APOE modulates the size and compactness of fibrillar Aβ plaques.**

(A) Composite tile showing individual X-34+ fibrillar plaques, with increasing plaque size (measured in diameter) along the x-axis and increasing compactness (measured in Angular Second Moment (ASM) value) along the y-axis. Higher ASM values correlate with higher compactness. Scale bar: 10 μm. (B) Stacked bar plot showing proportion of X-34+ fibrillar plaques from APOE2, APOE3 and APOE4 groups in different size categories (diameter in μm) along the x-axis. Lower limit for plaque size set to 5 μm in diameter. Size categories used: small (5–20 μm diameter), medium (20–40 μm diameter) and large (above 40 μm diameter). (C) Box plot showing ASM values as an estimate of plaque compactness in APOE2, APOE3 and APOE4 groups. Size categories used: (i) small (5–20 μm diameter), (ii) medium (20–40 μm diameter) and (iii) large (above 40 μm diameter). Data points show value for individual plaques ($n = 6$–10 mice per group. 3 FOV per mouse). In C(i), APOE2 vs APOE3 \*\*$p = 0.0093$; APOE2 vs APOE4 $p = 0.547$; APOE3 vs APOE4 \*\*\*$p = 0.0003$. In C(ii), APOE2 vs APOE3 \*\*\*$p = 0.0001$; APOE2 vs APOE4 $p = 0.9818$; APOE3 vs APOE4 \*\*\*\*$p < 0.0001$. In C(iii), APOE2 vs APOE3 \*\*\*\*$p < 0.0001$; APOE2 vs APOE4 $p = 0.471$; APOE3 vs APOE4 \*\*\*\*$p < 0.0001$. Statistical tests: Data presented as median and interquartile range ± values within 1.5 times the interquartile range (C). Minima and maxima for box plots in C(i) APOE2 (0.001, 0.0457), APOE3 (0.0011, 0.0501), APOE4 (0.0007, 0.0565); in C(ii) APOE2 (0.0011, 0.0106), APOE3 (0.0015, 0.0184), APOE4 (0.0008, 0.0111); in C(iii) APOE2 (0.0012, 0.0106), APOE3 (0.0024, 0.0097), APOE4 (0.0009, 0.0095). Linear mixed effects model with Tukey's HSD test in (C) done at mouse sample level.

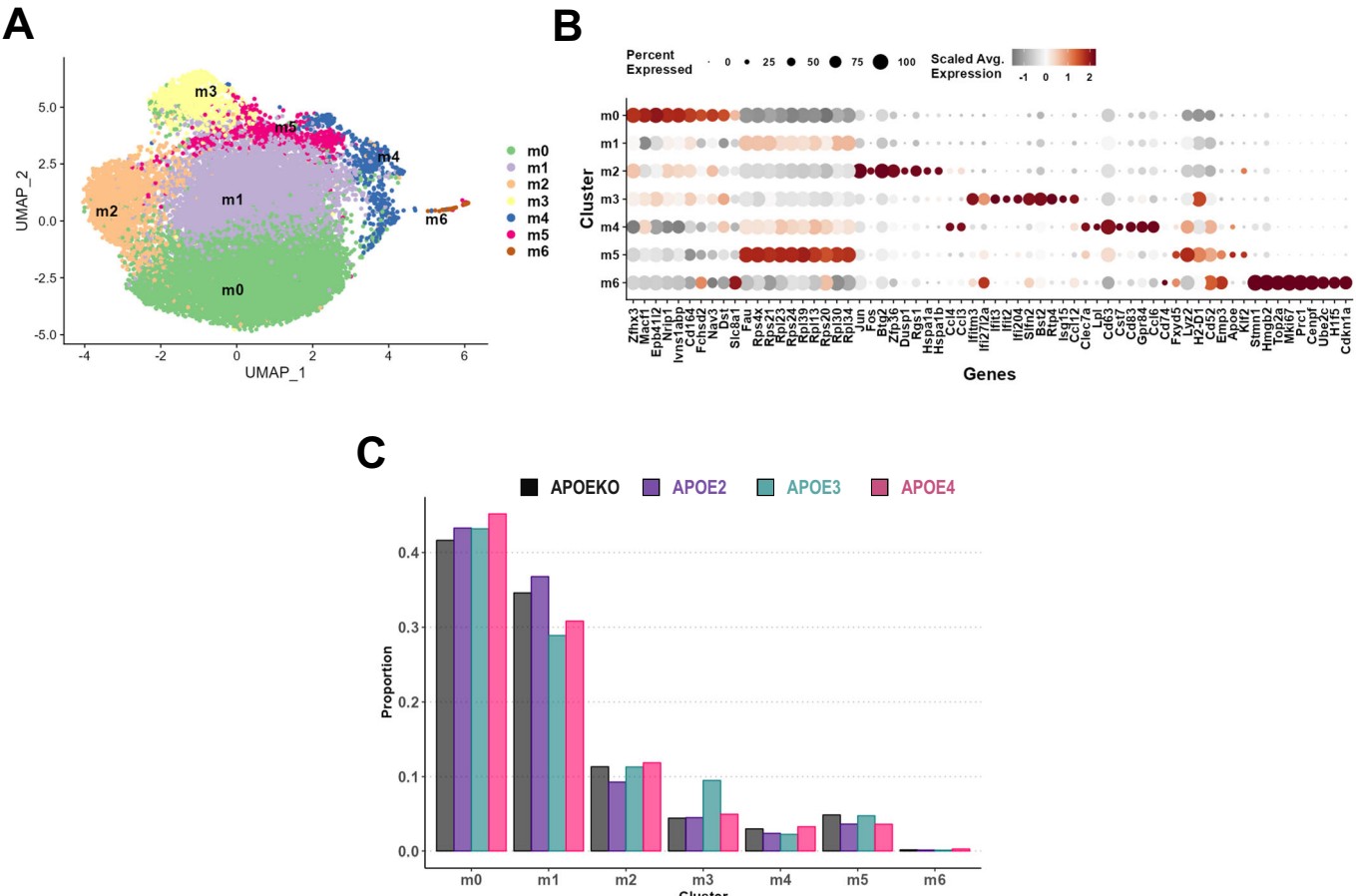

**Figure EV2. Astrocyte-derived APOE influences the expression of microglial cell state markers.**

(A) UMAP plot showing 18,569 APOE-deficient microglia (Cd11b+/Cd45-low) sorted from 6 months old mouse brains from the four experimental groups (n = 2 or 3 mice per group). Different sub-populations identified through unbiased clustering have been assigned cluster numbers. (B) Dot plot showing the top 10 differentially expressed genes in each cluster. Colour scale indicates normalized expression level, scaled per gene (z-score). Dot size indicates percentage of cells, in each cluster, expressing the gene. (C) Bar plot showing the proportion of microglia from each experimental group present in the different clusters.

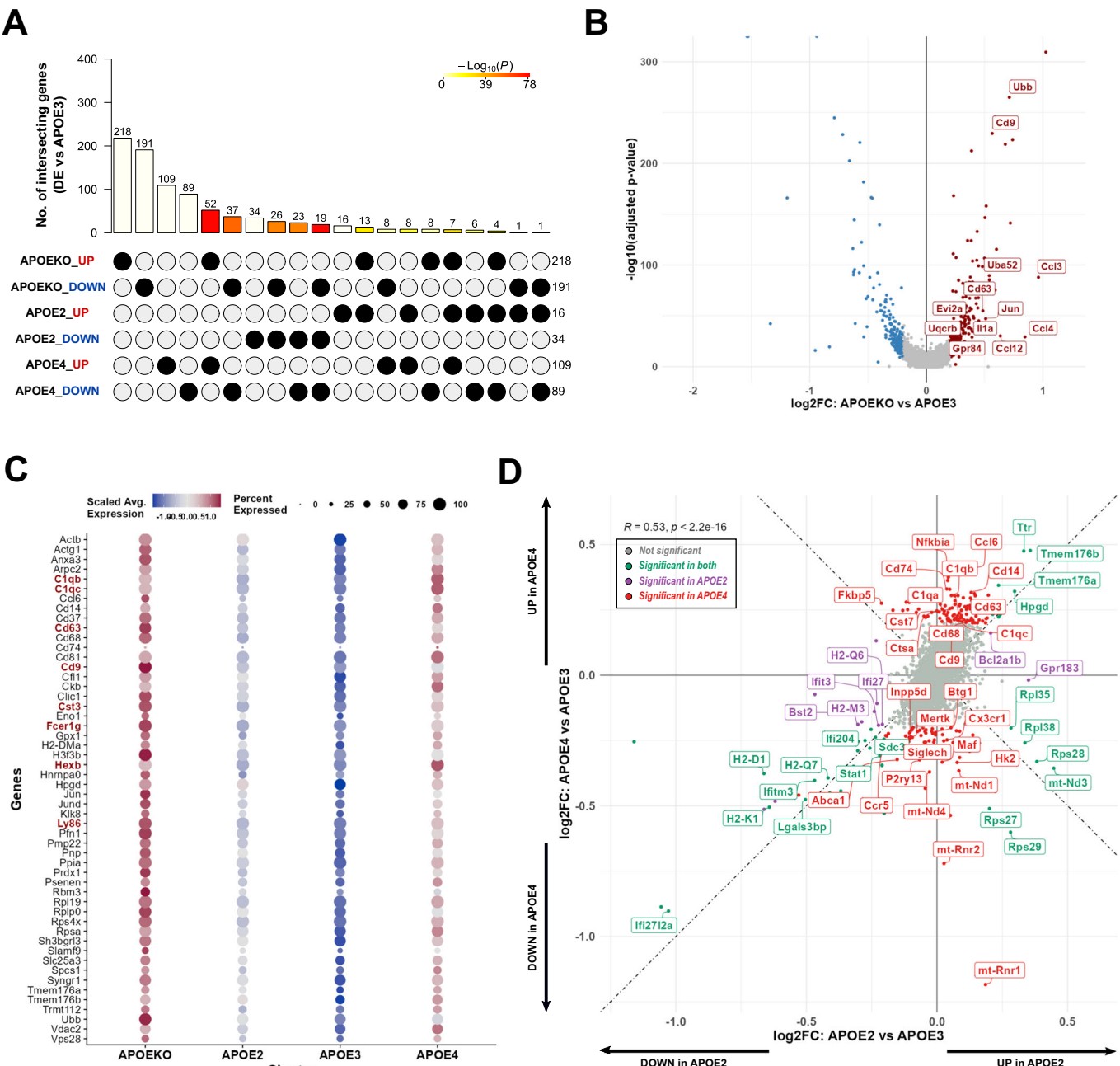

**Figure EV3. Astrocyte-derived APOE modulates gene expression in microglia.**

(A) Upset plot showing number of differentially expressed genes (UP or DOWN) in microglia from each experimental group, compared to APOE3 group. Bar plot shows the number of differentially expressed genes (DEGs) common between different DE analyses (overlapping sets indicated by black dots in column below the bars). Bars are coloured according to statistical significance of intersection of DEGs. (B) Volcano plot showing differentially expressed genes between APOEKO telencephalon astrocytes and APOE3 expressing telencephalon astrocytes. Data points for significant genes are coloured (Red for UP and blue for DOWN). Significance assigned based on |Log2(Fold Change)| > 0.2 and adjusted p-value < 0.05. Genes discussed in text are indicated in red. (C) Dot plot showing expression of the 52 genes (Fig. 2A) commonly upregulated in APOEKO and APOE4 microglia, split by experimental groups. Colour scale indicates normalized expression level, scaled per gene (z-score). Dot size indicates percentage of cells, in each group, expressing the gene. Plaque-induced genes (PIGs) (Chen et al, 2020) are coloured in red on y-axis. (D) Quadrant plot comparing differential expression of genes in microglia in APOE2 vs APOE3 (along x-axis) and in APOE4 vs APOE3 (along y-axis). Colours in legend key indicate statistical significance of genes up- or downregulated in APOE2 or APOE4 or both. Significance of differentially expressed genes based on |Log2(Fold Change)| > 0.2 and adjusted p-value < 0.05. Pearson's correlation, R = 0.52. Statistical tests: MAST differential expression test in (B), (C) and (D), p-values were adjusted with Bonferroni correction based on the total number of genes in the dataset.

