## [Peer Review File · EMBO Molecular Medicine]

APOE from astrocytes restores Alzheimer's A β -pathology and DAM-like responses in APOE deficient microglia

Pranav Preman, Daan Moechars, Emre Fertan, Leen Wolfs, Lutgarde Serneels, Disha Shah, Jochen Lamote, Suresh Poovathingal, An Snellinx, Renzo Mancuso, Sriram Balusu, David Klenerman, Amaia Arranz, Mark Fiers, and Bart De Strooper

Corresponding authors: Bart De Strooper (b.strooper@ucl.ac.uk) , Mark Fiers (mark.fiers@kuleuven.be)

Review Timeline:

Submission Date:	24th Apr 24
Editorial Decision:	22nd May 24
Revision Received:	14th Aug 24
Editorial Decision:	27th Sep 24
Revision Received:	15th Oct 24
Accepted:	16th Oct 24

Editor: Poonam Bheda

Transaction Report:

22nd May 2024

Dear Prof. De Strooper,

Thank you for the submission of your manuscript to EMBO Molecular Medicine. We have now received feedback from the three reviewers who agreed to evaluate your manuscript. As you will see from the reports below, the referees acknowledge the interest of the study and are overall supporting publication of your work pending appropriate revisions. In particular, all three reviewers have suggested improvements to the manuscript in regards to the animal model as follows: that you should determine how much overexpression of each APOE is achieved compared to endogenous levels, evaluate potential effects of APOE early overexpression on neuronal development via synaptic quantification, clearly state that the mice used in the study are Rag2-deficient, and check whether the astrocyte-limited expression observed at 2 months is altered at 6 months when the mice are analyzed.

Addressing the reviewers' concerns in full in a point-by-point response will be necessary for further considering the manuscript in our journal, and acceptance of the manuscript will entail a second round of review. EMBO Molecular Medicine encourages a single round of revision only and therefore, acceptance or rejection of the manuscript will depend on the completeness of your responses included in the next, final version of the manuscript. For this reason, and to save you from any frustrations in the end, I would strongly advise against returning an incomplete revision. If you would like to discuss further the points raised by the referees, I am available to do so via email or video. Let me know if you are interested in this option.

We are expecting your revised manuscript within three months, if you anticipate any delay, please contact us. When submitting your revised manuscript, please carefully review the instructions that follow below. We perform an initial quality control of all revised manuscripts before re-review; failure to include requested items will delay the evaluation of your revision.

We require:

4) A .docx formatted letter INCLUDING the reviewers' reports and your detailed point-by-point responses to their comments. As part of the EMBO Press transparent editorial process, the point-by-point response is part of the Review Process File (RPF), which will be published alongside your paper.

5) A complete author checklist, which you can download from our author guidelines (<https://www.embopress.org/page/journal/17574684/authorguide#submissionofrevisions>). Please insert information in the checklist that is also reflected in the manuscript. The completed author checklist will also be part of the RPF.

6) Please note that all corresponding authors are required to supply an ORCID ID for their name upon submission of a revised manuscript.

7) It is mandatory to include a 'Data Availability' section after the Materials and Methods. Before submitting your revision, primary datasets produced in this study need to be deposited in an appropriate public database, and the accession numbers and database listed under 'Data Availability'. Please remember to provide a reviewer password if the datasets are not yet public (see <https://www.embopress.org/page/journal/17574684/authorguide#dataavailability>).

In case you have no data that requires deposition in a public database, please state so in this section. Note that the Data Availability Section is restricted to new primary data that are part of this study. This study includes no data deposited in external repositories.

8) For data quantification: please specify the name of the statistical test used to generate error bars and P values, the number (n) of independent experiments (specify technical or biological replicates) underlying each data point and the test used to calculate p-values in each figure legend. The figure legends should contain a basic description of n, P and the test applied. Graphs must include a description of the bars and the error bars (s.d., s.e.m.). Please provide exact p values.

9) Our journal encourages inclusion of *data citations in the reference list* to directly cite datasets that were re-used and

obtained from public databases. Data citations in the article text are distinct from normal bibliographical citations and should directly link to the database records from which the data can be accessed. In the main text, data citations are formatted as follows: "Data ref: Smith et al, 2001" or "Data ref: NCBI Sequence Read Archive PRJNA342805, 2017". In the Reference list, data citations must be labeled with "[DATASET]". A data reference must provide the database name, accession number/identifiers and a resolvable link to the landing page from which the data can be accessed at the end of the reference. Further instructions are available at .

13) Author contributions: CRediT has replaced the traditional author contributions section because it offers a systematic machine readable author contributions format that allows for more effective research assessment. Please remove the Authors Contributions from the manuscript and use the free text boxes beneath each contributing author's name in our system to add specific details on the author's contribution. More information is available in our guide to authors.

Please also suggest a visual abstract to illustrate your article as a jpeg file 550 px wide x 300-600 px high.

Share synopsis text and image, as well as eTOC:

Please note that these would be the final versions and changes during proofing are usually not allowed

16) As part of the EMBO Publications transparent editorial process initiative (see our policy here:

https://www.embopress.org/transparent-process#Review_Process), EMBO Molecular Medicine will publish online a Peer Review File (PRF) to accompany accepted manuscripts.

In the event of acceptance, this file will be published in conjunction with your paper and will include the anonymous referee reports, your point-by-point response and all pertinent correspondence relating to the manuscript. Let us know whether you agree with the publication of the PRF and as here, if you want to remove or not any figures from it prior to publication.

I look forward to receiving your revised manuscript.

Yours sincerely,

Poonam Bheda

Poonam Bheda, PhD
Scientific Editor
EMBO Molecular Medicine

**** Reviewer's comments ****

Referee #1 (Comments on Novelty/Model System for Author):

This manuscript by Preman et al. addresses some of the most critical questions related to APOE genotype in Alzheimer's disease. First, they compared the impact of the various APOE genotypes (E2, E3, and E4) on APOE knock-out by overexpression of the different isoforms using AAV. Then, they ask how astrocytes expressing APOE impact hallmarks characteristic of Alzheimer's disease, such as microglia reactivity and amyloid deposition. Finally, they assessed the impact of astrocytic APOE on microglia. The authors found that astrocytic APOE is sufficient to alter amyloid deposition; however, with this set of experiments, I believe the role of astrocytic APOE on microglia remains. The comparison of APOE2, APOE3, and APOE4 highlights a complex mechanism of action of APOE2, and multiple groups have previously avoided a complete comparison to simplify their work and conclusions. I salute the efforts of the authors for including APOE2. Publishing more manuscripts like this one is essential to better understand APOE genotypes. Below are some concerns that should be addressed before publication

Development of the AAV model:

While we understand the authors are using an artificial overexpression model, it is essential to compare expression levels across the different genotypes at the RNA and protein levels and quantify how much overexpression is achieved compared to endogenous levels.

Based on the data shown, only a small fraction (14%) of the astrocytes are infected by the AAV. Therefore, it is essential to show where the infected cells are in all microscopy images.

Since 90% of the mCherry-positive cells are astrocytes, it is important to identify what are the other cell types that express APOE and discuss how this could impact the phenotypes observed.

Synaptic impairment: changes in synaptic density are essential to Alzheimer's disease pathogenesis in human and mouse models. However, it has not been evaluated in this study. It is even more critical as the AAV was injected early after birth, and APOE was shown to alter neuronal development. Moreover, one of the most significant DEG identified in the single-cell study is PDE10a, which, as the authors pointed out, can also impact synapse and memory formation. Therefore, synaptic quantification should be done in this model to validate the model itself didn't cause a developmental phenotype and to have a complete picture of the impact of APOE genotype of AD.

Single-cell analysis: All conclusions based on single-cell analysis need to be based on quantifiable methods. A hypergeometric test should be performed to evaluate the significance of common genes across genotypes (Fig 3a and S7a), and an enrichment score or pathway analysis should be used to label astrocyte and microglia transcriptional clusters.

Astrocyte-microglia interactions: While the impact of astrocytic APOE on amyloid plaque is well-characterized, the link between astrocytes and microglia is indirect. The microglia figures are solely based on the presence of amyloid plaque, and it needs to be clarified if and where APOE-expressing astrocytes are located in similar regions. Microglia are changing in response to amyloid load but that was known. Therefore, a better characterization of astrocyte-microglia direct or indirect interaction or identifying potential communication pathways using single-cell RNA sequencing data is needed. As is, the title is broad and misleading about the role of microglia and should be more specific.

Referee #1 (Remarks for Author):

This manuscript by Preman et al. addresses some of the most critical questions related to APOE genotype in Alzheimer's disease. First, they compared the impact of the various APOE genotypes (E2, E3, and E4) on APOE knock-out by overexpression of the different isoforms using AAV. Then, they ask how astrocytes expressing APOE impact hallmarks characteristic of Alzheimer's disease, such as microglia reactivity and amyloid deposition. Finally, they assessed the impact of astrocytic APOE on microglia. The authors found that astrocytic APOE is sufficient to alter amyloid deposition; however, with this set of experiments, I believe the role of astrocytic APOE on microglia remains. The comparison of APOE2, APOE3, and APOE4 highlights a complex mechanism of action of APOE2, and multiple groups have previously avoided a complete comparison to simplify their work and conclusions. I salute the efforts of the authors for including APOE2. Publishing more manuscripts like this one is essential to better understand APOE genotypes. Below are some concerns that should be addressed before publication

Development of the AAV model:

While we understand the authors are using an artificial overexpression model, it is essential to compare expression levels across the different genotypes at the RNA and protein levels and quantify how much overexpression is achieved compared to endogenous levels.

Based on the data shown, only a small fraction (14%) of the astrocytes are infected by the AAV. Therefore, it is essential to show where the infected cells are in all microscopy images.

Since 90% of the mCherry-positive cells are astrocytes, it is important to identify what are the other cell types that express APOE and discuss how this could impact the phenotypes observed.

Synaptic impairment: changes in synaptic density are essential to Alzheimer's disease pathogenesis in human and mouse models. However, it has not been evaluated in this study. It is even more critical as the AAV was injected early after birth, and APOE was shown to alter neuronal development. Moreover, one of the most significant DEG identified in the single-cell study is PDE10a, which, as the authors pointed out, can also impact synapse and memory formation. Therefore, synaptic quantification should be done in this model to validate the model itself didn't cause a developmental phenotype and to have a complete picture of the impact of APOE genotype of AD.

Single-cell analysis: All conclusions based on single-cell analysis need to be based on quantifiable methods. A hypergeometric test should be performed to evaluate the significance of common genes across genotypes (Fig 3a and S7a), and an enrichment score or pathway analysis should be used to label astrocyte and microglia transcriptional clusters.

Astrocyte-microglia interactions: While the impact of astrocytic APOE on amyloid plaque is well-characterized, the link between astrocytes and microglia is indirect. The microglia figures are solely based on the presence of amyloid plaque, and it needs to be clarified if and where APOE-expressing astrocytes are located in similar regions. Microglia are changing in response to amyloid load but that was known. Therefore, a better characterization of astrocyte-microglia direct or indirect interaction or identifying potential communication pathways using single-cell RNA sequencing data is needed. As is, the title is broad and misleading about the role of microglia and should be more specific.

Referee #2 (Comments on Novelty/Model System for Author):

This is a relevant and appropriate model system

Referee #2 (Remarks for Author):

The current manuscript by Preman et al explores the effects of astrocyte-selective expression of the three APOE alleles in an ApoE KO background in the APP-NLFG mice. This exciting study adds to a growing, but still incomplete, understanding of cell-autonomous and non-autonomous effects of ApoE during various stages of AD pathogenesis. Inclusion and purposeful analyses of the role of APOE2, which is typically excluded and quite understudied, is a major strength. Further, the finding that microglia, themselves lacking ApoE, can still be driven to a DAM-like state via secretion of astrocyte ApoE is an intriguing and important observation. The overall concept is not especially novel, as other groups have previously investigated the effects of astrocyte-derived (or astrocyte-selective knockout of) human ApoE. However, it should be noted that none of these prior studies included an investigation of the protective APOE2 allele, as done in the current study, which is also unique in its experimental approach. Thus, the impact and significance of the current study - which was well-designed and thoughtfully completed - is still quite high.

Other major comments:

The authors generally have done an excellent job in summarizing the literature in their Introduction section. However, the statement "AD mouse models fail to produce amyloid plaque pathology in the absence of ApoE" is not accurate; nor do the authors provide a reference for this statement. While the APPV717F+/- TG mouse model showed no amyloid deposition (Bales et al 1999), many other studies have showed much more modest/nuanced effects of ApoE deletion on amyloid pathology. The authors should be careful and precise in their wording, as this is a complicated area of research with many studies with sometimes seemingly conflicting findings, particularly in regards to the effects of mApoe, human APOE, and ApoE KO.

The authors show that there was no difference in total APOE protein amount across the three isoform groups. This is very different than other human APOE mouse models, where E2 mice have higher ApoE concentrations, as well as human E2+ tissue, which also shows higher protein levels. Not only differences in the isoform of human ApoE, but also the total amount of those isoforms, could very well contribute to or explain E4>E3>E2 variance in amyloid pathology and AD risk. This is a critical point that warrants a bit more discussion.

There are several instances where the references do not seem to match the statements. For example, the "Holtzman, Bales et al 2000" paper does not feature a GFAP-specific astrocyte expression as the authors state. While the statement made here is still correct, the authors should carefully check each reference associated with their claims for accuracy.

All the mice used in this study are Rag2 deficient. It is assumed that this is due to other ongoing studies in the lab where this

model is necessary. However, this is not mentioned until a passing description in the Methods. The authors should state this up front and discuss any potential issues/confounds with using this model in the current study.

The recap of the findings in the last paragraph of the introduction mention that E2 expression did not reduce amyloid more than E3 - but should also state that both E2 and E3 were reduced compared to E4. As an aside, these findings would seem to agree with the original papers describing the APOE x 5XFAD mice from Mary Jo LaDu's group and others - this could be a point worth mentioning in the Discussion section.

The Results section should state when the mice were injected (and how) with AAV (the Methods are not exceptionally clear, but it seems immediately after birth).

Can the authors confirm that clusters 10 and 11 do not have microglial/myeloid cell contamination?

APOE is one of the differentially expressed genes in cluster 5. However, cluster 5 does not differ between the 3 AAV groups and the APOE KO control group. How is this possible / why do the authors think this is?

The images in Fig 4 should be labeled with the experimental group they came from (I believe APOE4-AAV). Did the authors confirm this Cd68 and CtsD staining in all three hAPOE experimental groups? If so, these data/representative images would be helpful to include in the figure, or at least as supplemental data.

The quantification of plaque load PLX is not explicitly shown (only representative images in Fig 5B), but this would seem to be important data to show - and directly compare against the non-PLX group.

Are the data points for the non-PLX group the same mice as data shown in earlier figures? This is fine, but if so it should be stated.

The relevance of the data in Fig 5C was not entirely clear - could the authors please reword this section of the Results to more clearly detail their thoughts here?

Minor comments:

In the introduction, the phrasing of "the APOE2 variant was the first identified protective gene" suggests APOE2 is a gene (APOE is the gene), rather than a variant or allele

"We also explored [the] astrocyte transcriptome ..."

Fig 3D is a clever way to visualize both the E4 and E2 differences relative to E3, but adding additional labels or otherwise clarifying the directionality could be even more helpful to readers.

Could the authors provide a bit more perspective on how they think their findings inform the role of microglia-derived ApoE in AD pathology. In other words, how do their results fit with findings from the Liu and Yin et al 2023 papers where hAPOE was selectively induced or deleted in microglia (both suggesting less of a DAM response) versus other papers suggesting an E4-mediated increase in DAM (ex. Lee et al, 2023 and Serrano-Pozo 2021)?

I would suggest replacing "badly understood" with "poorly" or "incompletely"

Referee #3 (Comments on Novelty/Model System for Author):

This is an important study that aims at determining the role of astrocyte APOE isoform expression in the development of AD core features like Ab accumulation and microglia responses. We know that (1) APOE4 is the main genetic risk factor for AD, (2) APOE is necessary for Ab plaque formation, (3) APOE is one of the most highly expressed genes in astrocytes and very specific to these cells in the healthy brain, (4) microglia strongly upregulate APOE in AD. However, we don't know how these observations relate to each other to contribute to AD development. In this manuscript the authors make a careful and thoughtful attempt to address this open question. Certainly there are many remaining questions to answer in order understand the interaction of astrocytes and microglia in the context of APOE4 driven AD, but this is an enlightening step forward towards this goal.

Strengths of the manuscript are:

- Careful assessment of the state of the field.
- Rigorous analysis and statistical assessment of the experimental data.
- Thorough interpretation of the results in the context of current knowledge.
- Acknowledgment of the limitations and further research avenues.

- Investigation of the very important question on what is the part astrocyte and microglia APOE play on the formation of Ab plaques and how the expression of different APOE isoforms in astrocytes may influence microglia responses.

Recommendations to improve the manuscript:

- The authors show that their AAVs are astrocyte specific in the AppNLGF mice at 2 months old. However, all their experiments are performed at 6 months and it could be possible that the Ab pathology alters the cell specific expression of the AAV. Similar assessments at 6 months would reassure that APOE isoforms are still astrocyte specific at the point of analysis.
- When mentioning Cluster 8 (Crym) citing Chai and <https://www.nature.com/articles/s41586-024-07138-0> would be more appropriate than Chai and Soto.
- When discussing Figure 3E, it would be good to demonstrate if the differences in expression between phenotypes are significantly different. This could be done either in comparison to APOE3 or with using ANOVA across genes and conditions.
- The following two sentences sound contradictory, perhaps the first one can be rephrased to increase clarity. "we provide evidence that microglia lacking endogenous APOE expression, can be stirred in their response to amyloid pathology in a non-cell autonomous way by the APOE isoforms secreted by astrocytes." And "microglial responses to fibrillar plaques do not seem to depend on the isoform of APOE produced by astrocytes, but are likely mediated directly by the plaque load."
- Astrocyte APOE3 upregulates interferon response in both astrocytes and microglia. This is very interesting, and it would be worth "hearing" how the authors interpret this finding in the discussion.
- I would also be interested in learning how the authors conceal the fact that "it is possible that APOE-A β co-aggregates are taken up by microglia for clearance, depletion of which leads to increased levels of APOE in the mouse brain" while at the same time there is less Ab amyloid in the absence of microglia.

Referee #3 (Remarks for Author):

This is an important study that aims at determining the role of astrocyte APOE isoform expression in the development of AD core features like Ab accumulation and microglia responses. We know that (1) APOE4 is the main genetic risk factor for AD, (2) APOE is necessary for Ab plaque formation, (3) APOE is one of the most highly expressed genes in astrocytes and very specific to these cells in the healthy brain, (4) microglia strongly upregulate APOE in AD. However, we don't know how these observations relate to each other to contribute to AD development. In this manuscript the authors make a careful and thoughtful attempt to address this open question. Certainly there are many remaining questions to answer in order understand the interaction of astrocytes and microglia in the context of APOE4 driven AD, but this is an enlightening step forward towards this goal.

Strengths of the manuscript are:

- Careful assessment of the state of the field.
- Rigorous analysis and statistical assessment of the experimental data.
- Thorough interpretation of the results in the context of current knowledge.
- Acknowledgment of the limitations and further research avenues.
- Investigation of the very important question on what is the part astrocyte and microglia APOE play on the formation of Ab plaques and how the expression of different APOE isoforms in astrocytes may influence microglia responses.

Recommendations to improve the manuscript:

- The authors show that their AAVs are astrocyte specific in the AppNLGF mice at 2 months old. However, all their experiments are performed at 6 months and it could be possible that the Ab pathology alters the cell specific expression of the AAV. Similar assessments at 6 months would reassure that APOE isoforms are still astrocyte specific at the point of analysis.
- When mentioning Cluster 8 (Crym) citing Chai and <https://www.nature.com/articles/s41586-024-07138-0> would be more appropriate than Chai and Soto.
- When discussing Figure 3E, it would be good to demonstrate if the differences in expression between phenotypes are significantly different. This could be done either in comparison to APOE3 or with using ANOVA across genes and conditions.
- The following two sentences sound contradictory, perhaps the first one can be rephrased to increase clarity. "we provide evidence that microglia lacking endogenous APOE expression, can be stirred in their response to amyloid pathology in a non-cell autonomous way by the APOE isoforms secreted by astrocytes." And "microglial responses to fibrillar plaques do not seem to depend on the isoform of APOE produced by astrocytes, but are likely mediated directly by the plaque load."
- Astrocyte APOE3 upregulates interferon response in both astrocytes and microglia. This is very interesting, and it would be worth "hearing" how the authors interpret this finding in the discussion.
- I would also be interested in learning how the authors conceal the fact that "it is possible that APOE-A β co-aggregates are taken up by microglia for clearance, depletion of which leads to increased levels of APOE in the mouse brain" while at the same time there is less Ab amyloid in the absence of microglia.

Leuven 06/08/2024

Dear Editor,

Thank you for your input and suggestions to improve our manuscript and also for the very helpful discussion we had on 2/07/2024. Overall we are very pleased with the positive comments of the reviewers, who all indicated the strengths of the manuscript and support publication. We have addressed their comments in our rebuttal and added, where needed, further support with additional figures in the manuscript. We provide a point per point answer in the attached rebuttal letter. In your letter you indicated 4 questions that seemed most crucial to address. Please find below our answers, as also discussed at our meeting.

We look forward to your answer, hoping for a positive decision.

Bart De Strooper

1. - how much overexpression of each APOE is achieved compared to endogenous levels

In this study we have only used ApoE-KO animals and compared for the effects of APOE versus this base line. Crucially, human APOE is expressed under the Aldh1l1 promoter in the brain regions where the AAV virus is targeted too and therefore comparing its expression with ApoE expression from endogenous promoters is prone to misinterpretation as endogenous APOE will be more uniformly distributed over the brain. Constitutive expression of human APOE alleles under the endogenous mouse ApoE promoter (Liao et al., 2015; Liemisa et al., 2023) leads to global concentrations of human APOE in the range of 750-1000 ng per mg of homogenised brain tissue which is several folds higher than the range of 1-2 ng per mg tissue that we observed in our model, but this is not indicative of the increased local concentration in the transduced regions of our experiments. We therefore checked that the levels of expression were comparable across the experimental groups as shown in Fig 1D. We have now, as requested, additionally added data for human APOE mRNA levels across the experimental groups as an additional Appendix figure S1F, which confirms similar levels of expression between the three isoforms at the RNA level.

2.- evaluate potential effects of APOE early overexpression on neuronal development via synaptic quantification

The purpose of this study was to evaluate the effect of astrocytic-APOE isoforms on amyloid plaque pathology and associated glial reactivity, with focus on microglia. We agree that it would be highly interesting to look at the effect of APOE on additional pathological features of AD such as dystrophic neurites around plaques, synaptic density loss, behavioural defects etc.. As the reviewers correctly pointed out, it is also likely that APOE has non cell autonomous effects on neuronal development. Such question would need additional work in wild type mice in which the APOE is knocked out and the different human APOE isoforms are re-expressed. In order to keep the study design focused and feasible, we limited our work on the amyloid plaque inducing aspects of APOE and used only ApoE^{-/-}/App^{NL-G-F} mice in our study. That is the reason why we did not make other claims regarding additional aspects of the complex physiology of APOE. We discuss this limitation of the study now as follows:

“Future studies comparing the effect of APOE deletion in AD versus wildtype mouse models should help to confirm these findings. Moreover, studying the effect of APOE isoforms in wild type mice on neuronal development and synapse formation, will be of interest as APOE is known to influence synaptic density and to cause behavioural defects in AD mouse (Dumanis et al., 2009; Klein et al., 2010; Koffie et al., 2012).”

3.- clearly state that the mice used in the study are Rag2-deficient

We apologise for the confusion regarding the genetic background of our mouse model. All experiments shown in the main paper are mice that express normal Rag2. We only used Rag2 deficient mice for an initial pilot study to characterize AAV transduction efficiency and cell-type specificity, data which have been presented only in two supplementary figures, as mentioned in the Materials and Methods > Mice section, as well as in corresponding figure legends: “All the data presented in this study are from App^{NL-G-F} x ApoE^{-/-} mice. For a pilot study to check viral transduction efficiency and specificity, App^{NL-G-F} x ApoE^{-/-} x Rag2^{-/-} mice were used, and data from these are presented in Appendix figures: S1A, S1E, S1F, S2A and S2B.”

4.- check whether the astrocyte-limited expression observed at 2 months is altered at 6 months when the mice are analyzed

There is no known universal marker for astrocytes. We have used astrocyte-specific promoter Aldh1l1 for our expression system and in the paper, we confirm that more than 90% of mCherry-positive cells are ACSA2-positive astrocytes. The remaining 10% might represent Aldh1l1-positive astrocytes which are ACSA2-negative. Neither of these markers are expressed in 100% of all astrocytes in brain.

We have followed up on the concern that other cells than astrocytes are transduced by the vector. In the previous version we had shown absence of colocalization with neurons, microglia and oligodendrocytes with mCherry expression in Fig S1D, at 2 months of age. Also in line with reviewer 3's comments, we have now replaced these images with staining from samples at 6 months of age as now seen in novel Appendix Figure S1D, we have no evidence for mCherry expression in microglia (no colocalization with Iba1), oligodendrocytes (no colocalization with APC) or neurons (no colocalization with NeuN) at 6 months of age neither. So, we can conclude based on the FACS analysis at 6 months that more than 90% of the expression is confirmed to be in astrocytes, and from the immunohistochemistry we can deduce that other major cell types in the brain do not express the viral vector. In addition, the promoter we are using is astrocyte specific, which further strengthens the claim that a large portion of the 10% ACSA2-negative cells are likely to be astrocytes. While we cannot formally exclude that a small number of non-astrocytic cells was transduced, this is likely a very small number, and we do not see how this would affect any of the conclusions in our manuscript. But we have indicated this more clearly in the new version of the manuscript as follows:

“At 6 months of age, we used immunofluorescence to confirm mCherry expression in ALDH1L1+ astrocytes (Appendix Fig. S1B). The ALDH1L1 promoter is specifically expressed in astrocytes in mouse brain (Cahoy et al., 2008; Zhang et al., 2014) and has been extensively used to target gene expression in an astrocyte-specific setting (Clarke et al., 2018; Michalovicz et al., 2019; Hasel et al., 2021; Endo et al., 2022; Mahan et al., 2022). Using flow cytometry, we found that around 14% of ACSA2+ astrocytes expressed mCherry (Appendix Fig. S1Cii) and around 90% of the mCherry-positive cells were positive for astrocyte marker ACSA2 (Appendix Fig. S1Ciii). It is likely that remaining mCherry-positive cells are subpopulation of astrocytes that are ACSA2-negative. We do not exclude that a small number of non-astrocytic cells are transduced by the viral particles. However, we confirmed by immunofluorescence the lack of mCherry expression in NeuN+ neurons, Apc+

oligodendrocytes and Iba1+ microglia (Appendix Fig. S1D). We also confirmed APOE expression in mCherry+ cells (Appendix Fig. S1E), demonstrating that we reconstituted human APOE expression in astrocytes.”

***** Reviewer's comments *****

We thank all the reviewers for their constructive feedback and for supporting publication of our study, with specially mentioning the inclusion of APOE2 allele which, as rightly pointed out, is often missing in previously published studies. We paste below the full comments of the reviewers. We indicate the positive comments in red without further comments, while we address as well as possible the constructive criticism of the three reviewers with further explanation, clarification, or additional experimental evidence. We also indicate what has been changed in the manuscript.

Note to all reviewers: Addressing the concerns of Referee #2 regarding the upregulation of human APOE gene in Cluster 5 of astrocytes, we have re-done the single cell analysis to exclude any potential artifacts caused by including this gene because in the overexpression system, APOE is under a constitutive promoter based on viral transduction, and hence not a part of the normal transcriptional regulatory networks. We mention this exclusion in the Materials and Methods section. After the re-analysis, we have updated all the figures related to single cell analysis, namely Figures 2, 3, EV2 (previously S6), EV3 (previously S7) and S6. Please note that all the minor shifts in total number of cells or p-values in differential expression are negligible, and the overall results and conclusion remain unchanged. This further confirms that human APOE gene was unlikely to be driving the formation of Cluster 5. Please also note that in the re-analysis, to improve clarity for readers, the astrocyte and microglia clusters have been renamed with prefixes "a" and "m" respectively.

Referee #1 (Comments on Novelty/Model System for Author):

This manuscript by Preman et al. addresses some of the most critical questions related to APOE genotype in Alzheimer's disease. First, they compared the impact of the various APOE genotypes (E2, E3, and E4) on APOE knock-out by overexpression of the different isoforms using AAV. Then, they ask how astrocytes expressing APOE impact hallmarks characteristic of Alzheimer's disease, such as microglia reactivity and amyloid deposition. Finally, they assessed the impact of astrocytic APOE on microglia. The authors found that astrocytic APOE is sufficient to alter amyloid deposition; however, with this set of experiments, I believe the role of astrocytic APOE on microglia remains. The comparison of APOE2, APOE3, and APOE4 highlights a complex mechanism of action of APOE2, and multiple groups have previously avoided a complete comparison to simplify their work and conclusions. **I salute the efforts of the authors for including APOE2. Publishing more manuscripts like this one is essential to better understand APOE genotypes. Below are some concerns that should be addressed before publication**

Development of the AAV model:

While we understand the authors are using an artificial overexpression model, it is essential to compare expression levels across the different genotypes at the RNA and protein levels and quantify how much overexpression is achieved compared to endogenous levels.

We agree that artificial overexpression models have inherent limitations and we have discussed those better in the new version of the manuscript. We have addressed this concern to a large extent by analysing how levels of protein expression affect the results. Performing linear regression analysis, we determined that APOE has a dose-dependent effect on amyloid plaque formation and that this was isoform-specific with APOE4 and APOE2 levels having a positive association while APOE3 levels showed no association with amyloid levels (Fig. 1E)

In this study human APOE is only expressed (under Aldh1l1 promoter) in brain regions where the AAV virus is expressed. Therefore, comparing its expression with APOE expression from endogenous promoters is prone to misinterpretation as endogenous APOE will be more uniformly distributed over the brain. Constitutive expression of human APOE alleles under the endogenous mouse *ApoE* promoter (Liao et al., 2015; Liemisa et al., 2023) leads to global concentrations of human APOE in the range of 750-1000 ng per mg of homogenised brain tissue which is several folds higher than the range of 1-2 ng per mg tissue that we observe in our model. However, this is not indicative of the increased local concentration in the transduced regions of our experiments. We therefore checked that the levels of expression were comparable across the experimental groups as shown in Fig 1D. We have now, as requested, additionally added data for human APOE mRNA levels across the experimental groups as an additional supplementary Appendix Figure S1F, which confirms similar levels of expression between the three isoforms at the RNA level as well. By doing so we have provided sufficient controls to allow us to make comparisons of the effects of the different genotypes on the amyloid plaques and the microglia in the neighbourhoods of the APOE expressing astrocytes.

Appendix Figure S1. (F) Bar plot showing human APOE mRNA levels at 6 months of age in mouse brain homogenates measured by semiquantitative real-time PCR. Data points show mean value for 3 technical replicates per mouse (n= 3 mice per group). ns = non-significant.

Based on the data shown, only a small fraction (14%) of the astrocytes are infected by the AAV. Therefore, it is essential to show where the infected cells are in all microscopy images.

All the images were taken from AAV-transduced cortical regions since the amyloid plaque pathology is reconstituted only in these regions (Appendix Fig. S2A). For all the microscopy images, we now include a reference to these overview images of coronal brain sections (Appendix Fig. S2A), to make it clear for the readers where the transduced cells are imaged. However, we agree with the reviewer that it is important to provide transparency and we have now also added the mCherry images (indicating the cells that are transduced) in Figures 4A and 5B to show the transduced astrocytes.

Figure 4. (A) IF images of AAV transduced cortical regions (see Appendix Fig. S2A), at 6 months of age. mCherry (red) shows the transduced astrocytes. X-34 staining (white) shows the fibrillar plaque deposits. Co-staining with anti-Iba1 antibody shows microglial cells (green). Scale bar: 100 μ m.

Figure 5. (B) IF images of AAV transduced cortical regions (see Appendix Fig. S2A) at 6 months of age from control (top row) and PLX3397-treated (bottom row) groups. mCherry (red) shows the transduced astrocytes. X-34 staining (white) shows the fibrillar plaque deposits in mouse brain cortex. Co-staining with anti-Iba1 antibody shows microglial cells (green). Scale bar: 100 μ m.

Since 90% of the mCherry-positive cells are astrocytes, it is important to identify what are the other cell types that express APOE and discuss how this could impact the phenotypes observed.

We have used astrocyte-specific promoter Aldh1l1 for our expression system, and we can confirm that more than 90% of mCherry-positive cells are ACSA2-positive astrocytes. The remaining 10%

might represent *Aldh1l1*-positive astrocytes which are *ACSA2*-negative. Neither of these markers are expressed in 100% of all astrocytes in brain.

We have followed up on the concern that other cells than astrocytes are transduced by the vector. In the previous version we had shown absence of colocalization with neurons, microglia and oligodendrocytes with mCherry expression at 2 months of age in Fig. S1D. Also addressing similar comments by Referee #3, we have now replaced these images with analyses from samples at 6 months of age in Appendix Figure S1D (reproduced below). We find no evidence for mCherry expression in microglia (no colocalization with *Iba1*), oligodendrocytes (no colocalization with *APC*) or neurons (no colocalization with *NeuN*). So, we can conclude based on the FACS analysis that more than 90% of the expression is confirmed to be in astrocytes, and from the immunohistochemistry we can deduce that other major cell types in the brain do not express the viral vector. In addition, the promoter we are using is astrocyte specific, which further strengthens the claim that the 10% *ACSA2*-negative cells are likely to be astrocytes. While we cannot formally exclude that a small number of non-astrocytic cells was transduced, this is likely a very small number, and we do not see how this would affect any of the conclusions in our manuscript. But we have indicated this more clearly in the new version of the manuscript as follows:

“At 6 months of age, we used immunofluorescence to confirm mCherry expression in ALDH1L1+ astrocytes (Appendix Fig. S1B). The ALDH1L1 promoter is specifically expressed in astrocytes in mouse brain (Cahoy et al., 2008; Zhang et al., 2014) and has been extensively used to target gene expression in an astrocyte-specific setting (Clarke et al., 2018; Michalovicz et al., 2019; Hasel et al., 2021; Endo et al., 2022; Mahan et al., 2022). Using flow cytometry, we found that around 14% of *ACSA2*+ astrocytes expressed mCherry (Appendix Fig. S1Cii) and around 90% of the mCherry-positive cells were positive for astrocyte marker *ACSA2* (Appendix Fig. S1Ciii). It is likely that remaining mCherry-positive cells are a subpopulation of astrocytes that are *ACSA2*-negative. We do not exclude that a small number of non-astrocytic cells are transduced by the viral particles. However, we confirmed by immunofluorescence the lack of mCherry expression in *NeuN*+ neurons, *Apc*+ oligodendrocytes and *Iba1*+ microglia (Fig. Appendix S1D). We also confirmed *APOE* expression in mCherry+ cells (Appendix Fig. S1E), demonstrating that we reconstituted human *APOE* expression in astrocytes.”

Appendix Figure S1. (D) IF images of AAV transduced cortical region (see Appendix Fig. S2A) of 6 months old $App^{NL-G-F} \times ApoE^{-/-}$ mouse brain. mCherry (red) shows the transduced cells. Co-staining

with anti-Iba1 antibody showing microglia (cyan) (Note that this is the same image as S1B and co-staining was done with Aldh1l1); with anti-Apc antibody showing oligodendrocytes cells (green); with anti-NeuN antibody showing neurons (white) (Note that Apc and NeuN panels are shown from the same image as these markers were co-stained in the same slide). Images below show zoomed in (5x) views indicated by the inset box of white dashed-lines. Scale bar: 100 μ m.

Synaptic impairment: changes in synaptic density are essential to Alzheimer's disease pathogenesis in human and mouse models. However, it has not been evaluated in this study. It is even more critical as the AAV was injected early after birth, and APOE was shown to alter neuronal development. Moreover, one of the most significant DEG identified in the single-cell study is PDE10a, which, as the authors pointed out, can also impact synapse and memory formation. Therefore, synaptic quantification should be done in this model to validate the model itself did not cause a developmental phenotype and to have a complete picture of the impact of APOE genotype of AD.

The purpose of this study was to evaluate the effect of astrocytic-APOE isoforms on amyloid plaque pathology and associated glial reactivity, with focus on microglia. We agree that it would be highly interesting to look at the effect of APOE on additional pathological features of AD such as dystrophic neurites around plaques, synaptic density loss, behavioural defects etc. As the reviewers correctly pointed out, it is also likely that APOE has non cell autonomous effects on neuronal development. Such question would need additional work in wild type mice in which the APOE is knocked out and the different human APOE isoforms are re-expressed. In order to keep the study design focused and feasible, we limited our work on the amyloid plaque inducing aspects of APOE and used only $ApoE^{-/-}/App^{NL-G-F}$ mice in our study. That is the reason why we did not make other claims regarding additional aspects of the complex physiology of APOE. We discuss this limitation of the study now in the 7th paragraph of Discussion section as follows:

“Future studies comparing the effect of APOE deletion in AD versus wildtype mouse models should help to confirm these findings. Moreover, studying the effect of APOE isoforms in wild type mice on neuronal development and synapse formation, will be of interest as APOE is known to influence synaptic density and to cause behavioural defects in AD mouse (Dumanis et al., 2009; Klein et al., 2010; Koffie et al., 2012).” We hope that by clearly pointing out this limitation of the study, that the reviewer can agree that these questions can be addressed in follow up studies.

Single-cell analysis: All conclusions based on single-cell analysis need to be based on quantifiable methods. A hypergeometric test should be performed to evaluate the significance of common genes across genotypes (Fig 3a and S7a), and an enrichment score or pathway analysis should be used to label astrocyte and microglia transcriptional clusters.

*We thank the reviewer for these valuable suggestions. We have now calculated significance levels (p-values) indicating how likely it is to find a set of genes overlap in differential expression analyses between genotypes. For this we have replaced the upset-plots from **Figures 3A and EV3A (previously S7A)** using SuperExactTest algorithm (Wang et al., 2015), based on hypergeometric testing, developed for calculating statistical distribution for multiset intersections. We have also updated the **Figures 3A and EV3A (previously S7A)** to indicate the p-values by using colour-coding for the bars. The detailed results from the statistical test are present in supplementary tables Table EV3 (for Figure 3A) and Table EV6 (for Figure S7). Please note that using the SuperExactTest approach also resulted in a transition from an exclusive intersection count (akin to a Venn diagram) for the original upset plots*

(where the overlap of sets excluded elements present in additional sets), to an inclusive intersection count. Under this new approach, we calculate the total overlap irrespective of additional set memberships (e.g., an element in A&B is counted regardless of whether it is also in C). This results in slight changes in the values in figures 3A and EV3A (previously S7A), but does not change the conclusions.

Figure 3. (A) Upset plot showing number of differentially expressed genes (UP or DOWN) in telencephalon astrocytes from each experimental group, compared to APOE3 group. Bar plot shows the number of differentially expressed genes (DEGs) common between different DE analyses (overlapping sets indicated by black dots in column below the bars). Bars are colored according to statistical significance of intersection of DEGs.

Figure EV3. (A) Upset plot showing number of differentially expressed genes (UP or DOWN) in microglia from each experimental group, compared to APOE3 group. Bar plot shows the number of differentially expressed genes (DEGs) common between different DE analyses (overlapping sets indicated by black dots in column below the bars). Bars are colored according to statistical significance of intersection of DEGs.

For pathway analysis, we have also performed Gene Ontology (GO:BP) analysis to further characterize astrocyte and microglia clusters which is now included in new supplementary tables Table EV2 (astrocytes) and Table EV5 (microglia).

Astrocyte-microglia interactions: While the impact of astrocytic APOE on amyloid plaque is well-characterized, the link between astrocytes and microglia is indirect. The microglia figures are solely based on the presence of amyloid plaque, and it needs to be clarified if and where APOE-expressing astrocytes are located in similar regions. Microglia are changing in response to amyloid load but that was known. Therefore, a better characterization of astrocyte-microglia direct or indirect interaction or identifying potential communication pathways using single-cell RNA sequencing data is needed. As is, the title is broad and misleading about the role of microglia and should be more specific.

We are now providing the mCherry images in the main figure to show that the microglia and amyloid plaques are imaged in the AAV-transduced regions only and that the transduced astrocytes are close to the microglia (Figures 4A and 5B, as shown above). Also, we would like to point out that the amyloid plaque reconstitution occurs only in the AAV-transduced region as seen in Appendix Fig. S2A. The reviewer is right that we could not separate astrocyte-microglia direct or indirect interaction mediated in the presence of amyloid load. We understand that this is a limitation of our study and therefore we mentioned this in the Discussion section as- "APOE expression and A β pathology are closely associated in our model which makes it difficult to determine the factors contributing directly or indirectly to the DEG changes seen in microglia and astrocytes. Future studies to compare the effect of APOE deletion and isoforms in AD versus wildtype mouse models should help to confirm these findings."

To analyse communication pathways between astrocyte and microglia libraries, we have tried NicheNet analysis after merging the astrocyte and microglia single-cell libraries. This did not however yield any interpretable results. We believe that this might be due to methodological limitations of our study that only a part of the microglia are exposed to AAV transduced astrocytes. This has also impacted our ability to identify significant shifts in microglia cell states between the experimental groups (Figure EV2C), as discussed in the manuscript. Therefore, we believe it is not useful to delve further into this. We did recently publish a paper regarding astrocyte-microglia communication in the amyloid environment based on a spatial transcriptomics method (Mallach et al., 2024), and believe future studies using such approaches can give better insights about APOE allele specific role in these glial cell-cell communication.

Regarding the title, this is a fair criticism and we agree that the previous title was making too broad claims. We have adapted the title accordingly to take this criticism into full account. Previous title: "APOE from astrocytes is sufficient to facilitate microglial responses to Alzheimer's A β -pathology". The proposed new title: "APOE from astrocytes restore Alzheimer's A β -pathology and DAM-like responses in APOE deficient microglia"

Referee #2 (Comments on Novelty/Model System for Author):

This is a relevant and appropriate model system

Referee #2 (Remarks for Author):

The current manuscript by Preman et al explores the effects of astrocyte-selective expression of the three APOE alleles in an ApoE KO background in the APP-NLFG mice. **This exciting study adds to a growing, but still incomplete, understanding of cell-autonomous and non-autonomous effects of ApoE during various stages of AD pathogenesis. Inclusion and purposeful analyses of the role of APOE2, which is typically excluded and quite understudied, is a major strength.** Further, the finding that microglia, themselves lacking ApoE, can still be driven to a DAM-like state via secretion of astrocyte ApoE is an intriguing and important observation. The overall concept is not especially novel, as other groups have previously investigated the effects of astrocyte-derived (or astrocyte-selective knockout of) human ApoE. However, it should be noted that none of these prior studies included an investigation of the protective APOE2 allele, as done in the current study, which is also unique in its experimental approach. **Thus, the impact and significance of the current study - which was well-designed and thoughtfully completed - is still quite high.**

Other major comments:

The authors generally have done an excellent job in summarizing the literature in their Introduction section. However, the statement "AD mouse models fail to produce amyloid plaque pathology in the absence of ApoE" is not accurate; nor do the authors provide a reference for this statement. While the APPV717F+/- TG mouse model showed no amyloid deposition (Bales et al 1999), many other studies have showed much more modest/nuanced effects of ApoE deletion on amyloid pathology. The authors should be careful and precise in their wording, as this is a complicated area of research with many studies with sometimes seemingly conflicting findings, particularly in regards to the effects of mApoE, human APOE, and ApoE KO.

We apologise for the oversight, and we have modified the phrasing in the Introduction to reflect the above comment, with references added as follows:

"On the other hand, AD mouse models show limited amyloid plaque pathology in the absence of ApoE (Bales et al., 1999; Holtzman et al., 2000; Ulrich et al., 2018)."

The authors show that there was no difference in total APOE protein amount across the three isoform groups. This is very different than other human APOE mouse models, where E2 mice have higher ApoE concentrations, as well as human E2+ tissue, which also shows higher protein levels. Not only differences in the isoform of human ApoE, but also the total amount of those isoforms, could very well contribute to or explain E4>E3>E2 variance in amyloid pathology and AD risk. This is a critical point that warrants a bit more discussion.

We agree with the reviewer that this is a limitation of our overexpression model. We do not have fully control on the levels of expression of the constructs as this is determined stochastically by the number of astrocytes that are targeted in each experiment. We have tried to handle this issue by making quantitative correlations between APOE levels and amyloid levels in our model and gained important insight regarding APOE's dose-dependent as well as isoform-dependent effect on amyloid formation

(Fig 1E). We agree with the reviewer that we should discuss this issue more explicitly in the paper and have now added references in 4th paragraph of Discussion as follows:

“It has also been shown that APOE2 expression leads to increased APOE levels in mouse models and human brain (Bales et al., 2009; Conejero-Goldberg et al., 2014; Riddell et al., 2008), which can influence amyloid pathology. This is not recapitulated in our model due to inherent limitation of AAV-mediated overexpression.”

There are several instances where the references do not seem to match the statements. For example, the "Holtzman, Bales et al 2000" paper does not feature a GFAP-specific astrocyte expression as the authors state. While the statement made here is still correct, the authors should carefully check each reference associated with their claims for accuracy.

We rechecked the references for any unintended citation errors. However, in this particular case mentioned, the Holtzman, Bales et 2000 paper (<https://doi.org/10.1073/pnas.050004797>) clearly indicates (Materials and Methods > “Animals and Tissue Preparation” section as well as in Results> “Expression of Human apoE Results in Fibrillar A β Deposition and Neuritic Plaques” section) that the human APOE expression in their mouse model is driven by the GFAP promoter. The difference with our work is that in this case transgenic mouse were generated with GFAP driving the expression of human APOE3 or E4, while in our model ALDH1L1 promoter drives APOE expression.

All the mice used in this study are Rag2 deficient. It is assumed that this is due to other ongoing studies in the lab where this model is necessary. However, this is not mentioned until a passing description in the Methods. The authors should state this up front and discuss any potential issues/confounds with using this model in the current study.

We apologise for the confusion regarding the genetic background of our mouse model. All experiments shown in the main paper are mice that express normal Rag2. We only used Rag2 deficient mice for an initial pilot study to characterize AAV transduction efficiency and cell-type specificity, data which have been presented in two supplementary figures, as mentioned in the Materials and Methods > Mice section as well as in corresponding figure legends: “All the data presented in this study are from App^{NL-G-F} x ApoE^{-/-} mice. For a pilot study to check viral transduction efficiency and specificity, App^{NL-G-F} x ApoE^{-/-} x Rag2^{-/-} mice were used, and data from these are presented in Appendix figures: S1A, S1E, S1F, S2A and S2B.”

The recap of the findings in the last paragraph of the introduction mention that E2 expression did not reduce amyloid more than E3 - but should also state that both E2 and E3 were reduced compared to E4. As an aside, these findings would seem to agree with the original papers describing the APOE x 5XFAD mice from Mary Jo LaDu's group and others - this could be a point worth mentioning in the Discussion section.

We thank the reviewer for this suggestion to rephrase the Introduction section and have done so as follows:

“Interestingly, in our experimental paradigm, APOE2 expression only in astrocytes did not reduce the amyloid plaque formation compared to APOE3, even when APOE4 had significantly higher levels than both.”

Regarding literature, in our defence, we had already mentioned the suggested publications (including the ones from Mary Jo Ladu's group- Youmans et al 2012 and Rodriguez et al 2014) in the Discussion section 3rd paragraph : "Nevertheless other studies have also reported that APOE2 expression is associated with similar or even higher plaque load than APOE3 (Bales et al., 2009; Dodart et al., 2005; Rodriguez et al., 2014; Youmans et al., 2012)" as well as in the Introduction section 2nd paragraph: "Similarly, in mouse models, the effect of APOE2 on A β plaques remains controversial with some studies confirming decreased amyloid deposition compared to APOE3 (Castellano et al., 2011; Fagan et al., 2002; Hudry et al., 2013; Zhao et al., 2016), while others show similar or even increased deposits (Bales et al., 2009; Dodart et al., 2005; Rodriguez et al., 2014; Youmans et al., 2012)."

The Results section should state when the mice were injected (and how) with AAV (the Methods are not exceptionally clear, but it seems immediately after birth).

We apologise for this oversight and have now added this information in the Methods section (indicated in Results section as well), as follows:

Materials and Methods section:

"For intracerebroventricular (ICV) injections, on postnatal day 2/3, the mouse pups were anaesthetized by hypothermia and approximately 2×10^{10} infectious particles of AAV (1 μ l per site) were injected into the ventricles with Hamilton syringes bilaterally: halfway between lambda and bregma, 1 mm bilaterally from the midline and 3 mm from the pial surface (Kim et al., 2014)."

Results section:

*"In order to study the role of astrocyte-derived APOE on amyloid plaque pathology, we expressed the different human APOE isoforms (further referred to as APOE2, APOE3 and APOE4), along with reporter mCherry protein, specifically in astrocytes in an *ApoE*-deficient AD mouse model (*App*^{NL-G-F} x *ApoE*^{-/-}), using recombinant AAV2/8 particles (injected ICV at postnatal day 2/3, see Materials and Methods for details) and the astrocyte specific promoter ALDH1L1 (Fig. 1A)."*

Can the authors confirm that clusters 10 and 11 do not have microglial/myeloid cell contamination?

*We can confirm that all the astrocyte clusters presented in the figures are free from other cell type contamination. We provide additional data for this in the Materials and Methods > Analysis of single-cell RNA sequencing datasets > Quality control of cells and samples section, during the pre-processing stage of single cell transcriptome data, "After cell type annotation as described above, ~7% of cells had non-astrocyte identity." These non-astrocyte cells also included small proportion of microglial/myeloid cells (astrocyte QC Cluster 9) and hence were excluded from further analysis. We have now added new Appendix **Figure S6** to show the contaminating clusters that were excluded, and expanded the Quality control of cells and samples section in Materials and Methods as follows:*

"Step 2: Libraries from the 4 experimental groups were merged to have in total 32,982 cells (Appendix Fig. S6D). After cell type annotation as described above, ~7% of cells had non-astrocyte identity. Contaminating cell types were identified (Appendix Figs. S6D, E) as ependymal cells (astrocyte QC Cluster 5), mural cells (astrocyte QC Cluster 7), microglia (astrocyte QC Cluster 9), endothelial cells (astrocyte QC Cluster 10) and a population of undetermined identity (astrocyte QC

Cluster 8). Out of the rest, ~31% separated out as clusters (astrocyte QC Clusters 1 and 6) with low number of reads and gene count (Appendix Fig. S6F).’

Appendix **Figure S6**. **(D)** UMAP plot showing 32,982 astrocytes (mCherry+/Acsa2+) sorted from 6 months old mouse brains from the four experimental groups (n= 2 or 3 mice per group). **(E)** Dot plot showing the top 10 differentially expressed genes in each QC cluster of astrocyte library. Colour scale indicates normalized expression, scaled by gene. Dot size indicates percentage of cells, in each cluster, expressing the gene. Astrocyte QC Clusters 0, 1, 2, 3, 4 and 6 have astrocyte identity. Contaminating cell types identified are ependymal cells (Astrocyte QC Cluster 5), mural cells (Astrocyte QC Cluster 7), microglia (Astrocyte QC Cluster 9), endothelial cells (Astrocyte QC Cluster 10) and a population of undetermined identity (Astrocyte QC Cluster 8). **(F)** Violin plot showing (i) nUMI (total count of RNA transcripts captured), (ii) nGene (total number of genes), (iii) mito (percentage of mitochondrial genes), (iv) ribo (percentage of ribosomal genes), in each astrocyte QC cluster.

Additionally, Cluster 10 in Fig 2 especially upregulates several astrocyte markers such as *Gfap*, *s100a6*, *Vimentin* etc. Thus, we are confident that these clusters reflect cells with astrocyte identity.

APOE is one of the differentially expressed genes in cluster 5. However, cluster 5 does not differ between the 3 AAV groups and the APOE KO control group. How is this possible / why do the authors think this is?

We thank the reviewer for bringing this important observation to the discussion. It is unlikely that the Cluster 5 is driven based on APOE expression because, like the reviewer correctly points out, there is equal membership of the APOEKO group in this cluster. However, we agree that it was our mistake to

include human APOE gene in the single cell analysis pipeline since its expression is driven by variation in transduction in an overexpression system. Hence to avoid potential artifacts in the analysis, we have excluded the gene and re-done the single cell analysis to avoid confusion for the readers. We have also mentioned this in the Quality control of cells and samples section in Materials and Methods as follows:

“In this overexpression model, since APOE is under a constitutive promoter based on viral transduction, and hence not a part of the normal transcriptional regulatory networks, we excluded human APOE gene expression from both astrocyte and microglia datasets prior to the following steps.”

After the re-analysis, we have updated all the figures related to single cell analysis, namely Figures 2, 3, EV2 (previously S6), EV3 (previously S7) and S6. Please note that all the minor shifts in total number of cells or p-values in differential expression are negligible, and the overall results and conclusion remain unchanged. This further confirms that human APOE gene was unlikely to be driving the formation of Cluster 5.

Please also note that in the re-analysis, to improve clarity for readers, the astrocyte and microglia clusters have been renamed with prefixes “a” and “m” respectively.

The images in Fig 4 should be labeled with the experimental group they came from (I believe APOE4-AAV). Did the authors confirm this Cd68 and CtsD staining in all three hAPOE experimental groups? If so, these data/representative images would be helpful to include in the figure, or at least as supplemental data.

We thank the reviewer for this suggestion. We have added the label for APOE4 to the figure panel in Figure 4A. We also have staining for Cd68 and CtsD for all three APOE groups and we have included the APOE2 and APOE3 groups now in Appendix Figures S4G-H.

Appendix **Figure S4**: **(G)** IF images of AAV-APOE2 and AAV-APOE3 transduced cortical regions (see Appendix Fig. S2A) at 6 months of age. X-34 staining (white) shows the fibrillar plaque deposits. Co-staining with anti-Iba1 antibody shows microglial cells (green) and anti-Cd68 antibody shows phagocytic structures (red) inside microglia surrounding plaques. Scale bar: 100 μ m. **(H)** IF images of AAV-APOE2 and AAV-APOE3 transduced cortical regions (see Appendix Fig. S2A) at 6 months of age. X-34 staining (white) shows the fibrillar plaque deposits. Co-staining with anti-Iba1 antibody shows microglia (green) and anti-CtsD antibody shows lysosomal structures (red) inside clustered microglia surrounding plaques. Scale bar: 100 μ m.

The quantification of plaque load PLX is not explicitly shown (only representative images in Fig 5B), but this would seem to be important data to show - and directly compare against the non-PLX group.

*We thank the reviewer for this suggestion and have included the quantification of plaque load from the images for the PLX group in **Figure 5C**. We see a trend of decrease in plaque load in all 3 groups, but this reaches statistical significance only in the APOE4 group.*

Figure 5. (C) Bar plots showing number of X-34+ plaques per field of view (FOV, 20x magnification) and area of X-34+ plaques (as fraction of total area in FOV, 20x magnification), in the AAV-transduced cortical region for (i) APOE2, (ii) APOE3 and (iii) APOE4 groups and their respective PLX3397 treated groups. Data points show mean value of 3 fields of view per mouse (n= 4-10 mice per group). Data points for control non-treated groups are the same as shown in Fig. 1C

Are the data points for the non-PLX group the same mice as data shown in earlier figures? This is fine, but if so it should be stated.

Yes, the data-points for the non-PLX group in IHC quantification (Fig. 5C) and ELISA quantification (Fig. 5D) are the same mice as shown in earlier figures- Fig. 1C and 1E respectively. However, we have used different mice groups for IHC and ELISA experiments. We have now added this information to the figure legends to make this clear.

The relevance of the data in Fig 5C was not entirely clear - could the authors please reword this section of the Results to more clearly detail their thoughts here?

Here in Figure 5, we are checking whether PLX treatment can also influence amyloid load. We have two factors (APOE level and PLX treatment) that can influence amyloid plaque load. Comparison of plaque load from IHC images or comparison of guanidine- $A\beta_{42}$ between treated and non-treated groups, does not take into account the confounding effect of APOE levels on plaque load. Therefore, we used multiple regression analysis on the biochemical quantification of amyloid level, correcting for

APOE level, to determine the effect of PLX treatment on plaque load. We have rephrased the presentation of this data in the Results section to make this clearer as follow:

“Since we have earlier seen that APOE levels also influence A β plaque deposition (Fig. 1E), we used multiple regression analysis to account for the confounding effect of APOE levels while comparing guanidine-soluble A β_{42} levels between PLX3397-treated and untreated animals in each experimental group (Fig. 5C).”

Minor comments:

In the introduction, the phrasing of "the APOE2 variant was the first identified protective gene" suggests APOE2 is a gene (APOE is the gene), rather than a variant or allele

Corrected

"We also explored [the] astrocyte transcriptome ..."

Corrected

Fig 3D is a clever way to visualize both the E4 and E2 differences relative to E3, but adding additional labels or otherwise clarifying the directionality could be even more helpful to readers.

*We thank the reviewer for this suggestion. In order to avoid clutter and confusion regarding directionality, we have reduced the colours used. Now the colours represent significantly dysregulated genes in each group (APOE2 or APOE4 or both). Additionally, we have provided text along the axis to denote directionality of dysregulation of the genes. See adapted **Figure 3D** below*

Figure 3. (D) Quadrant plot comparing differential expression of genes in telencephalon astrocytes in APOE2 vs APOE3 (along x-axis) and in APOE4 vs APOE3 (along y-axis). Colours in legend key indicate significance of genes up- or downregulated in APOE2 or APOE4 or both. Significance of differentially expressed genes based on $|\text{Log}_2(\text{Fold Change})| > 0.2$ and adjusted p -value < 0.05 . Pearson's correlation, $R = 0.5$.

Could the authors provide a bit more perspective on how they think their findings inform the role of microglia-derived ApoE in AD pathology. In other words, how do their results fit with findings from the Liu and Yin et al 2023 papers where hAPOE was selectively induced or deleted in microglia (both suggesting less of a DAM response) versus other papers suggesting an E4-mediated increase in DAM (ex. Lee et al, 2023 and Serrano-Pozo 2021)?

We thank the reviewer for this interesting point of discussion. It is indeed true that recent evidence suggests that microglia-specific knockout of APOE4 can lead to increased clearing of amyloid plaques, with APOE4 expression failing to induce DAM response, compared to APOE3 (Liu et al., 2023; Yin et al., 2023b). In contrast, we have presented non-cell autonomous effects of different human APOE isoforms on the microglial response to plaques. It is possible that when these APOE isoforms are expressed by microglia, they can have varying effect on microglia transcriptomics. We mention this in 7th paragraph of Discussion as follows:

“Recent data also suggests a cell autonomous role of microglia-specific APOE4 expression in blocking the DAM response, compared to APOE3 (Liu et al., 2023; Yin et al., 2023b). In our model, we have

presented a non-cell autonomous effect of different astrocyte-specific APOE isoforms on microglial response to amyloid pathology.”

I would suggest replacing "badly understood" with "poorly" or "incompletely"

Corrected

Referee #3 (Comments on Novelty/Model System for Author):

This is an important study that aims at determining the role of astrocyte APOE isoform expression in the development of AD core features like Ab accumulation and microglia responses. We know that (1) APOE4 is the main genetic risk factor for AD, (2) APOE is necessary for Ab plaque formation, (3) APOE is one of the most highly expressed genes in astrocytes and very specific to these cells in the healthy brain, (4) microglia strongly upregulate APOE in AD. However, we don't know how these observations relate to each other to contribute to AD development. In this manuscript the authors make a careful and thoughtful attempt to address this open question. **Certainly there are many remaining questions to answer in order understand the interaction of astrocytes and microglia in the context of APOE4 driven AD, but this is an enlightening step forward towards this goal.**

Strengths of the manuscript are:

- Careful assessment of the state of the field.
- Rigorous analysis and statistical assessment of the experimental data.
- Thorough interpretation of the results in the context of current knowledge.
- Acknowledgment of the limitations and further research avenues.
- Investigation of the very important question on what is the part astrocyte and microglia APOE play on the formation of Ab plaques and how the expression of different APOE isoforms in astrocytes may influence microglia responses.

Recommendations to improve the manuscript:

- The authors show that their AAVs are astrocyte specific in the AppNLGF mice at 2 months old. However, all their experiments are performed at 6 months and it could be possible that the Ab pathology alters the cell specific expression of the AAV. Similar assessments at 6 months would reassure that APOE isoforms are still astrocyte specific at the point of analysis.

We apologise for the confusion regarding the data. The imaging for astrocyte specificity of mCherry expression (S1B) was done at 2 months of age. But, we wanted to make sure the astrocyte-specificity was maintained at 6 months of age and therefore quantification for the same, as shown in Fig S1C, was done using flow cytometry at 6 months age. We have rephrased the text to make this explicitly

clear. We have now added in addition images for cell-type markers (Appendix Figure S1D, below) to show specificity for astrocyte transduction at 6 months of age instead of 2 months of age.

Appendix Figure S1. **(D)** IF images of AAV transduced cortical region (see Appendix Fig. S2A) of 6 months old $App^{NL-G-F} \times ApoE^{-/-}$ mouse brain. mCherry (red) shows the transduced cells. Co-staining with anti-Iba1 antibody showing microglia (cyan) (Note that this is the same image as S1B and co-staining was done with Aldh1l1); with anti-Apc antibody showing oligodendrocytes cells (green); with anti-NeuN antibody showing neurons (white) (Note that Apc and NeuN panels are shown from the same image as these markers were co-stained in the same slide). Images below show zoomed in (5x) views indicated by the inset box of white dashed-lines. Scale bar: 100 μ m.

- When mentioning Cluster 8 (Crym) citing Chai and <https://www.nature.com/articles/s41586-024-07138-0> would be more appropriate than Chai and Soto.

We thank the reviewer for the suggested reference and have included this in the updated version.

- When discussing Figure 3E, it would be good to demonstrate if the differences in expression between phenotypes are significantly different. This could be done either in comparison to APOE3 or with using ANOVA across genes and conditions.

We thank the reviewer for this suggestion. We have now added statistical significance to the plot based on differential expression against APOE3 (Fig. 3E). Squares around dots denote statistical significance based on adjusted p-value < 0.05. We noticed that several of the significant genes do not cross the threshold of Log2(Fold Change) above 0.2 or below -0.2 that we have used for significance in our manuscript. These are indicated with red squares. Genes that have $|\text{Log}_2(\text{Fold Change})| > 0.2$ are indicated with green squares. As many of the statistically significant genes do not show log fold changes above 0.2 we have decided to move the figure to supplemental and is now presented as Appendix Figure S3E.

Appendix **Figure S3.** (E) Dot plot showing expression of previously identified AD risk genes, split by experimental groups. Colour scale indicates normalized expression level, scaled per gene (z-score). Dot size indicates percentage of cells, in each group, expressing the gene. Squares around dots mark statistically significant genes (expressed in more than 25% of cells in each group and with adjusted p-value < 0.05) based on differential expression against APOE3 group. Green squares indicate genes with $|\text{Log}_2(\text{Fold Change})| > 0.2$. Red squares indicate genes with $|\text{Log}_2(\text{Fold Change})| < 0.2$.

- The following two sentences sound contradictory, perhaps the first one can be rephrased to increase clarity. "we provide evidence that microglia lacking endogenous APOE expression, can be stirred in their response to amyloid pathology in a non-cell autonomous way by the APOE isoforms secreted by astrocytes." And "microglial responses to fibrillar plaques do not seem to depend on the isoform of APOE produced by astrocytes, but are likely mediated directly by the plaque load."

We apologise for the confusion regarding the text. The second sentence refers to there being no difference, between APOE groups, in morphological changes in clustered microglia around amyloid plaques. Since the first sentence was regarding APOE-deficient microglia being able to cluster around plaques irrespective of the APOE isoforms coming from astrocytes, we have now corrected the first sentence in the Introduction section to avoid confusion:

"We provide evidence that microglia lacking endogenous APOE expression, can be stirred in their response to amyloid pathology by APOE secreted by astrocytes"

- Astrocyte APOE3 upregulates interferon response in both astrocytes and microglia. This is very interesting, and it would be worth "hearing" how the authors interpret this finding in the discussion.

We thank the reviewer for this suggestion, and we have now added this to the 7th paragraph of Discussion section as follows:

"In both astrocytes and microglia, there was an enrichment of interferon response and antigen-presentation genes in the APOE3 dataset. This was accompanied with limited plaque formation, compact plaques and low levels of oligomeric A β in the mice brain. Recently it was reported that conditional knockout of microRNA miR155 in microglia in APP/PS1 mice leads to activation of interferon-mediated signaling, which was associated with increased phagocytosis and plaque compaction (Yin et al., 2023a)."

- I would also be interested in learning how the authors conceal the fact that "it is possible that APOE-A β co-aggregates are taken up by microglia for clearance, depletion of which leads to increased levels of APOE in the mouse brain" while at the same time there is less Ab amyloid in the absence of microglia.

We apologise for the confusion in the text. In this point of discussion, we have tried to provide a possible explanation for the increased APOE levels in the microglia-depleted mouse brain. While the microglia-depleted mouse brains have decreased deposition of β -amyloid plaques, when we are talking about APOE-A β co-aggregates here, we are referring to the soluble fraction of β -amyloid (Xia et al., 2024). Hence in the absence of clearance of these co-aggregates due to microglia depletion, it is possible that APOE levels can go up. At the same time the soluble fraction of amyloid can possibly increase too. However, kinetic studies of co-aggregate levels will need to be performed in order to confirm these speculations. To avoid the confusion regarding soluble and plaque forms of β -amyloid, we have now modified the text as follows:

*"We also observed increased levels of APOE in the microglia depleted mouse brains, irrespective of the isoform. APOE co-aggregates with soluble A β species *in vitro* (Xia et al., 2024). Thus, it is possible that APOE-A β co-aggregates are taken up by microglia for clearance, and when microglia is depleted, this leads to increased levels of APOE in the mouse brain."*

Bibliography

- Bales, K.R., Liu, F., Wu, S., Lin, S., Koger, D., DeLong, C., Hansen, J.C., Sullivan, P.M., Paul, S.M., 2009. Human APOE Isoform-Dependent Effects on Brain β -Amyloid Levels in PDAPP Transgenic Mice. *J. Neurosci.* 29, 6771–6779. <https://doi.org/10.1523/JNEUROSCI.0887-09.2009>
- Bales, K.R., Verina, T., Cummins, D.J., Du, Y., Dodel, R.C., Saura, J., Fishman, C.E., DeLong, C.A., Piccardo, P., Petegnief, V., Ghetti, B., Paul, S.M., 1999. Apolipoprotein E is essential for amyloid deposition in the APPV717F transgenic mouse model of Alzheimer's disease. *Proc. Natl. Acad. Sci.* 96, 15233–15238. <https://doi.org/10.1073/pnas.96.26.15233>
- Castellano, J.M., Kim, J., Stewart, F.R., Jiang, H., DeMattos, R.B., Patterson, B.W., Fagan, A.M., Morris, J.C., Mawuenyega, K.G., Cruchaga, C., Goate, A.M., Bales, K.R., Paul, S.M., Bateman, R.J., Holtzman, D.M., 2011. Human apoE Isoforms Differentially Regulate Brain Amyloid- β Peptide Clearance. *Sci. Transl. Med.* 3, 89ra57-89ra57. <https://doi.org/10.1126/scitranslmed.3002156>
- Conejero-Goldberg, C., Gomar, J.J., Bobes-Bascaran, T., Hyde, T.M., Kleinman, J.E., Herman, M.M., Chen, S., Davies, P., Goldberg, T.E., 2014. APOE2 enhances neuroprotection against Alzheimer's disease through multiple molecular mechanisms. *Mol. Psychiatry* 19, 1243–1250. <https://doi.org/10.1038/mp.2013.194>
- Dodart, J.-C., Marr, R.A., Koistinaho, M., Gregersen, B.M., Malkani, S., Verma, I.M., Paul, S.M., 2005. Gene delivery of human apolipoprotein E alters brain A β burden in a mouse model of Alzheimer's disease. *Proc. Natl. Acad. Sci.* 102, 1211–1216. <https://doi.org/10.1073/pnas.0409072102>
- Dumanis, S.B., Tesoriero, J.A., Babus, L.W., Nguyen, M.T., Trotter, J.H., Ladu, M.J., Weeber, E.J., Turner, R.S., Xu, B., Rebeck, G.W., Hoe, H.-S., 2009. ApoE4 Decreases Spine Density and Dendritic Complexity in Cortical Neurons *In Vivo*. *J. Neurosci.* 29, 15317–15322. <https://doi.org/10.1523/JNEUROSCI.4026-09.2009>
- Fagan, A.M., Watson, M., Parsadanian, M., Bales, K.R., Paul, S.M., Holtzman, D.M., 2002. Human and murine ApoE markedly alters A beta metabolism before and after plaque formation in a mouse model of Alzheimer's disease. *Neurobiol. Dis.* 9, 305–318. <https://doi.org/10.1006/nbdi.2002.0483>

- Holtzman, D.M., Fagan, A.M., Mackey, B., Tenkova, T., Sartorius, L., Paul, S.M., Bales, K., Hsiao Ashe, K., Irizarry, M.C., Hyman, B.T., 2000. Apolipoprotein E facilitates neuritic and cerebrovascular plaque formation in an Alzheimer's disease model. *Ann. Neurol.* 47, 739–747. [https://doi.org/10.1002/1531-8249\(200006\)47:6<739::AID-ANA6>3.0.CO;2-8](https://doi.org/10.1002/1531-8249(200006)47:6<739::AID-ANA6>3.0.CO;2-8)
- Hudry, E., Dashkoff, J., Roe, A.D., Takeda, S., Koffie, R.M., Hashimoto, T., Scheel, M., Spires-Jones, T., Arbel-Ornath, M., Betensky, R., Davidson, B.L., Hyman, B.T., 2013. Gene Transfer of Human ApoE Isoforms Results in Differential Modulation of Amyloid Deposition and Neurotoxicity in Mouse Brain. *Sci. Transl. Med.* 5, 212ra161. <https://doi.org/10.1126/scitranslmed.3007000>
- Kim, J.-Y., Grunke, S.D., Levites, Y., Golde, T.E., Jankowsky, J.L., 2014. Intracerebroventricular viral injection of the neonatal mouse brain for persistent and widespread neuronal transduction. *J. Vis. Exp. JoVE* 51863. <https://doi.org/10.3791/51863>
- Klein, R.C., Mace, B.E., Moore, S.D., Sullivan, P.M., 2010. Progressive loss of synaptic integrity in human apoE4 targeted replacement mice and attenuation by apoE2. *Neuroscience* 171, 1265–1272. <https://doi.org/10.1016/j.neuroscience.2010.10.027>
- Koffie, R.M., Hashimoto, T., Tai, H.-C., Kay, K.R., Serrano-Pozo, A., Joyner, D., Hou, S., Kopeikina, K.J., Frosch, M.P., Lee, V.M., Holtzman, D.M., Hyman, B.T., Spires-Jones, T.L., 2012. Apolipoprotein E4 effects in Alzheimer's disease are mediated by synaptotoxic oligomeric amyloid- β . *Brain* 135, 2155–2168. <https://doi.org/10.1093/brain/aws127>
- Liao, F., Zhang, T.J., Jiang, H., Lefton, K.B., Robinson, G.O., Vassar, R., Sullivan, P.M., Holtzman, D.M., 2015. Murine versus human apolipoprotein E4: differential facilitation of and co-localization in cerebral amyloid angiopathy and amyloid plaques in APP transgenic mouse models. *Acta Neuropathol. Commun.* 3, 70. <https://doi.org/10.1186/s40478-015-0250-y>
- Liemisa, B., Newbury, S.F., Novy, M.J., Pasato, J.A., Morales-Corraliza, J., Peng, K.Y., Mathews, P.M., 2023. Brain apolipoprotein E levels in mice challenged by a Western diet increase in an allele-dependent manner. *Aging Brain* 4, 100102. <https://doi.org/10.1016/j.nbas.2023.100102>
- Liu, C.-C., Wang, N., Chen, Yuanxin, Inoue, Y., Shue, F., Ren, Y., Wang, M., Qiao, W., Ikezu, T.C., Li, Z., Zhao, J., Martens, Y., Doss, S.V., Rosenberg, C.L., Jeevaratnam, S., Jia, L., Raulin, A.-C., Qi, F., Zhu, Y., Alnobani, A., Knight, J., Chen, Yixing, Linares, C., Kurti, A., Fryer, J.D., Zhang, B., Wu, L.-J., Kim, B.Y.S., Bu, G., 2023. Cell-autonomous effects of APOE4 in restricting microglial response in brain homeostasis and Alzheimer's disease. *Nat. Immunol.* 24, 1854–1866. <https://doi.org/10.1038/s41590-023-01640-9>
- Mallach, A., Zielonka, M., Lieshout, V. van, An, Y., Khoo, J.H., Vanheusden, M., Chen, W.-T., Moechars, D., Arancibia-Carcamo, I.L., Fiers, M., Strooper, B.D., 2024. Microglia-astrocyte crosstalk in the amyloid plaque niche of an Alzheimer's disease mouse model, as revealed by spatial transcriptomics. *Cell Rep.* 43. <https://doi.org/10.1016/j.celrep.2024.114216>
- Riddell, D.R., Zhou, H., Atchison, K., Warwick, H.K., Atkinson, P.J., Jefferson, J., Xu, L., Aschmies, S., Kirksey, Y., Hu, Y., Wagner, E., Parratt, A., Xu, J., Li, Z., Zaleska, M.M., Jacobsen, J.S., Pangalos, M.N., Reinhart, P.H., 2008. Impact of Apolipoprotein E (ApoE) Polymorphism on Brain ApoE Levels. *J. Neurosci.* 28, 11445–11453. <https://doi.org/10.1523/JNEUROSCI.1972-08.2008>
- Rodriguez, G.A., Tai, L.M., LaDu, M.J., Rebeck, G.W., 2014. Human APOE4 increases microglia reactivity at A β plaques in a mouse model of A β deposition. *J. Neuroinflammation* 11, 111. <https://doi.org/10.1186/1742-2094-11-111>
- Ulrich, J.D., Ulland, T.K., Mahan, T.E., Nyström, S., Nilsson, K.P., Song, W.M., Zhou, Y., Reinartz, M., Choi, S., Jiang, H., Stewart, F.R., Anderson, E., Wang, Y., Colonna, M., Holtzman, D.M., 2018. ApoE facilitates the microglial response to amyloid plaque pathology. *J. Exp. Med.* 215, 1047–1058. <https://doi.org/10.1084/jem.20171265>
- Wang, M., Zhao, Y., Zhang, B., 2015. Efficient Test and Visualization of Multi-Set Intersections. *Sci. Rep.* 5, 16923. <https://doi.org/10.1038/srep16923>
- Xia, Z., Prescott, E.E., Urbanek, A., Wareing, H.E., King, M.C., Olerinyova, A., Dakin, H., Leah, T., Barnes, K.A., Matuszyk, M.M., Dimou, E., Hidari, E., Zhang, Y.P., Lam, J.Y.L., Danial, J.S.H., Strickland, M.R., Jiang, H., Thornton, P., Crowther, D.C., Ohtonen, S., Gómez-Budia, M., Bell,

- S.M., Ferraiuolo, L., Mortiboys, H., Higginbottom, A., Wharton, S.B., Holtzman, D.M., Malm, T., Ranasinghe, R.T., Klenerman, D., De, S., 2024. Co-aggregation with Apolipoprotein E modulates the function of Amyloid- β in Alzheimer's disease. *Nat. Commun.* 15, 4695. <https://doi.org/10.1038/s41467-024-49028-z>
- Yin, Z., Herron, S., Silveira, S., Kleemann, K., Gauthier, C., Mallah, D., Cheng, Y., Margeta, M.A., Pitts, K.M., Barry, J.-L., Subramanian, A., Shorey, H., Brandao, W., Durao, A., Delpech, J.-C., Madore, C., Jedrychowski, M., Ajay, A.K., Murugaiyan, G., Hersh, S.W., Ikezu, S., Ikezu, T., Butovsky, O., 2023a. Identification of a protective microglial state mediated by miR-155 and interferon- γ signaling in a mouse model of Alzheimer's disease. *Nat. Neurosci.* 26, 1196–1207. <https://doi.org/10.1038/s41593-023-01355-y>
- Yin, Z., Rosenzweig, N., Kleemann, K.L., Zhang, X., Brandão, W., Margeta, M.A., Schroeder, C., Sivanathan, K.N., Silveira, S., Gauthier, C., Mallah, D., Pitts, K.M., Durao, A., Herron, S., Shorey, H., Cheng, Y., Barry, J.-L., Krishnan, R.K., Wakelin, S., Rhee, J., Yung, A., Aronchik, M., Wang, C., Jain, N., Bao, X., Gerrits, E., Brouwer, N., Deik, A., Tenen, D.G., Ikezu, T., Santander, N.G., McKinsey, G.L., Baufeld, C., Sheppard, D., Krasemann, S., Nowarski, R., Eggen, B.J.L., Clish, C., Tanzi, R.E., Madore, C., Arnold, T.D., Holtzman, D.M., Butovsky, O., 2023b. APOE4 impairs the microglial response in Alzheimer's disease by inducing TGF β -mediated checkpoints. *Nat. Immunol.* 24, 1839–1853. <https://doi.org/10.1038/s41590-023-01627-6>
- Youmans, K.L., Tai, L.M., Nwabuisi-Heath, E., Jungbauer, L., Kanekiyo, T., Gan, M., Kim, J., Eimer, W.A., Estus, S., Rebeck, G.W., Weeber, E.J., Bu, G., Yu, C., LaDu, M.J., 2012. APOE4-specific Changes in A β Accumulation in a New Transgenic Mouse Model of Alzheimer Disease*. *J. Biol. Chem.* 287, 41774–41786. <https://doi.org/10.1074/jbc.M112.407957>
- Zhao, L., Gottesdiener, A.J., Parmar, M., Li, M., Kaminsky, S.M., Chiuchiolo, M.J., Sondhi, D., Sullivan, P.M., Holtzman, D.M., Crystal, R.G., Paul, S.M., 2016. Intracerebral adeno-associated virus gene delivery of apolipoprotein E2 markedly reduces brain amyloid pathology in Alzheimer's disease mouse models. *Neurobiol. Aging* 44, 159–172. <https://doi.org/10.1016/j.neurobiolaging.2016.04.020>

27th Sep 2024

Dear Prof. De Strooper,

Thank you for the submission of your revised manuscript to EMBO Molecular Medicine. We have now received the enclosed reports from the referees that were asked to re-assess it. As you will see the reviewers are now globally supportive and I am pleased to inform you that we will be able to accept your manuscript pending the following final amendments:

- 1) Please check the "Author Checklist" carefully and complete all relevant questions. Please update the section on DNA and RNA sequences to indicate that they were used (for genotyping and RT-qPCR).
- 2) The email address for Leen Wolfs (leen.wolfs@kuleuven.be) does not seem to function properly as emails to this address bounce back to our system. Please provide an updated email address.
- 3) Please include a link to the scRNA-seq data in GEO in the Data availability section and ensure that the data are now released and publicly accessible.
- 4) Please rename "Conflict of Interest" to "Disclosure and competing interests statement". Please also disclose in this section that Bart de Strooper is an Editorial Board Member for EMBO Molecular Medicine. We updated our journal's competing interests policy in January 2022 and request authors to consider both actual and perceived competing interests. Please review the policy <https://www.embopress.org/competing-interests> and update your competing interests if necessary.
- 5) References: Please correct the reference citation in the reference list. Where there are more than 10 authors on a paper, note that only 10 should be listed, followed by "et al.". Please also remove the DOIs. Please check "Author Guidelines" for more information.

<https://www.embopress.org/page/journal/17574684/authorguide#referencesformat>

- 6) Please rename "Materials and Methods" to "Methods".

7) All materials and methods need to be described in the main text using our 'Structured Methods' format, which is required for all research articles. According to this format, the Methods section includes a Reagents and Tools Table (listing key reagents, experimental models, software and relevant equipment and including their sources and relevant identifiers) followed by a Methods and Protocols section describing the methods using a step-by-step protocol format. The aim is to facilitate adoption of the methodologies across labs. More information on how to adhere to this format as well as a downloadable template (.docx) for the Reagents and Tools Table can be found in our author guidelines:

<https://www.embopress.org/page/journal/17574684/authorguide#structuredmethods>

8) Please place individual sections of the manuscript in the following order: Title page - Abstract & Keywords - Introduction - Results - Discussion - Methods - Data Availability - Acknowledgements - Disclosure and Competing Interests Statement - The Paper Explained - References - Figure Legends - Expanded View Figure Legends.

9) For the figures and figure legends, please take care of the following:

- Please note that the exact p values are not provided in the legends of figures 1c(i-ii)-d, g; 4d(i); EV 1c(i-iii).
- Please indicate the statistical test used for data analysis in the legends of figures 2d; 3b; EV 3b.
- Please note that in figures 1c(i-ii)-d, g; 4d(i); EV 1c(i-iii); there is a mismatch between the annotated p values in the figure legend and the annotated p values in the figure file that should be corrected.
- Please note that the box plots need to be defined in terms of minima, maxima, in the legends of figures EV 1c(i-iii).
- Please note that information related to n is missing in the legend of figure 3c.
- Please note that the error bars are not defined in the legend of figure 4d(i).
- Please note that the scale bar needs to be defined for figure EV 1a.

10) Tables: Please rename the Table EV1-6 to Dataset EV1-6. Each dataset will need its legend removed from the manuscript and added to the corresponding file in a separate tab. Please update their callouts in main manuscript text.

11) Appendix file: Please add page numbers to the Table of Contents.

12) Funding: Please note that funding information should be given in the "Acknowledgements" section (not in its own separate section).

13) Synopsis:

- Synopsis image: Please edit your synopsis image so that at the width of 550 pixels it is between 250-400 pixels high. Currently the image is too high (550x600).

- Synopsis text: Please shorten the second bullet point to a maximum of 30 words. Please also upload this text separately from The Paper Explained (the latter of which should be provided in the main manuscript file.)

14) The Paper Explained: Please add "The Paper Explained" to the main manuscript text (not separately uploaded) to just after the "Disclosure and Competing Interests Statement".

15) As part of the EMBO Publications transparent editorial process initiative (see our policy here:

https://www.embopress.org/transparent-process#Review_Process), EMBO Molecular Medicine will publish online a Peer Review File (PRF) to accompany accepted manuscripts. This file will be published in conjunction with your paper and will include the

anonymous referee reports, your point-by-point response and all pertinent correspondence relating to the manuscript. Let us know whether you agree with the publication of the PRF and as here, if you want to remove or not any figures from it prior to publication. Please note that the Authors checklist will be published at the end of the PRF.

16) Please provide a point-by-point letter INCLUDING my comments as well as the reviewer's reports and your detailed responses (as Word file).

I look forward to reading a new revised version of your manuscript as soon as possible.

Yours sincerely,

Poonam Bheda

Poonam Bheda, PhD
Scientific Editor
EMBO Molecular Medicine

***** Reviewer's comments *****

Referee #1 (Comments on Novelty/Model System for Author):

The authors have responded adequately to all the concerns raised in the previous round of comments.

Referee #1 (Remarks for Author):

The authors have responded adequately to all the concerns raised in the previous round of comments.

Referee #2 (Comments on Novelty/Model System for Author):

n/a

Referee #2 (Remarks for Author):

The authors have carefully addressed all of my comments. They should be commended on an exciting and well written manuscript that will be of great interest to the APOE and AD research community.

Referee #3 (Remarks for Author):

I thank the authors for addressing my comments. I'd be very happy for this manuscript to be accepted for publication.

We thank all the reviewers for evaluating our response to their previous feedback and supporting the publication of our study. We have made the necessary amendments suggested by the editorial team.

16th Oct 2024

Dear Prof. De Strooper,

Congratulations on an excellent manuscript, I am pleased to inform you that your manuscript has been accepted for publication in the EMBO Molecular Medicine. Thank you for your comprehensive response to referee concerns and for providing detailed source data. It has been a pleasure to work with you to get this to the acceptance stage.

Yours sincerely,

Poonam Bheda, PhD
Scientific Editor
EMBO Molecular Medicine
